# BE AWARE OF THE NEIGHBORHOOD EFFECT: MODELING SELECTION BIAS UNDER INTERFERENCE

**Haoxuan Li**[1]     **Chunyuan Zheng**[1]     **Sihao Ding**[2]     **Peng Wu**[3,*]     **Zhi Geng**[3]
**Fuli Feng**[2]     **Xiangnan He**[2]
[1]Peking University     [2]University of Science and Technology of China
[3]Beijing Technology and Business University
hxli@stu.pku.edu.cn     dsihao@mail.ustc.edu.cn
{zhengchunyuan99, fulifeng93, xiangnanhe}@gmail.com
{pengwu, zhigeng}@btbu.edu.cn

## ABSTRACT

Selection bias in recommender system arises from the recommendation process of system filtering and the interactive process of user selection. Many previous studies have focused on addressing selection bias to achieve unbiased learning of the prediction model, but ignore the fact that potential outcomes for a given user-item pair may vary with the treatments assigned to other user-item pairs, named neighborhood effect. To fill the gap, this paper formally formulates the neighborhood effect as an interference problem from the perspective of causal inference and introduces a treatment representation to capture the neighborhood effect. On this basis, we propose a novel ideal loss that can be used to deal with selection bias in the presence of neighborhood effect. We further develop two new estimators for estimating the proposed ideal loss. We theoretically establish the connection between the proposed and previous debiasing methods ignoring the neighborhood effect, showing that the proposed methods can achieve unbiased learning when both selection bias and neighborhood effect are present, while the existing methods are biased. Extensive semi-synthetic and real-world experiments are conducted to demonstrate the effectiveness of the proposed methods.

## 1 INTRODUCTION

Selection bias is widespread in recommender system (RS) and challenges the prediction of users' true preferences (Wu et al., 2022; Chen et al., 2023), which arises from the recommendation process of system filtering and the interactive process of user selection (Marlin and Zemel, 2009; Huang et al., 2022). For example, in the rating prediction task, selection bias happens in explicit feedback data as users are free to choose which items to rate, so that the observed ratings are not a representative sample of all ratings (Steck, 2010; Wang et al., 2023c). In the post-click conversion rate (CVR) prediction task, selection bias happens due to conventional CVR models are trained with samples of clicked impressions while utilized to make inference on the entire space with samples of all impressions (Ma et al., 2018; Zhang et al., 2020; Wang et al., 2022a; Li et al., 2023f).

Inspired by the causal inference literature (Imbens and Rubin, 2015), many studies have proposed unbiased estimators for eliminating the selection bias, such as inverse propensity scoring (IPS) (Schnabel et al., 2016), self-normalized IPS (SNIPS) (Swaminathan and Joachims, 2015), and doubly robust (DR) methods (Wang et al., 2019; Chen et al., 2021a; Dai et al., 2022; Li et al., 2023d;e). Given the features of a user-item pair, these methods first estimate the probability of observing that user rating or clicking on the item, called propensity. Then the inverse of the propensity is used to weight the observed samples to achieve unbiased estimates of the ideal loss.

However, the theoretical guarantees of the previous methods are all established under the Stable Unit Treatment Values Assumption (SUTVA) (Rubin, 1980), which requires that the potential outcomes for one user-item pair do not vary with the treatments assigned to other user-item pairs (also known

---

*Corresponding author.

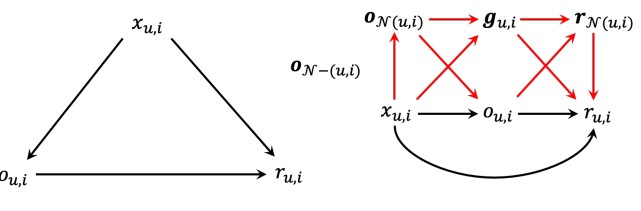

Figure 1: Causal diagrams of the existing debiasing methods under no interference assumption (left), and the proposed method taking into account the presence of interference (right), where $x_{u,i}$, $o_{u,i}$, and $r_{u,i}$ denote the confounder, treatment, and outcome of user-item pair $(u, i)$, respectively. In the presence of interference, $\mathcal{N}_{(u,i)}$ and $\mathcal{N}_{-(u,i)}$ denote the other user-item pairs affecting and not affecting $(u, i)$, respectively, and $g_{u,i}$ denotes the treatment representation to capture the interference.

as no interference or no neighborhood effect), as shown in Figure 1(a). In fact, such an assumption can hardly be satisfied in real-world scenarios. For example, a user's rating on an item can be easily influenced by other users' ratings on that item, as well as a user's clicking on an item might facilitate other users' clicking and purchasing of that item (Chen et al., 2021b; Zheng et al., 2021). Figure 1(b) shows a general causal diagram in the presence of interference in debiased recommendation.

To fill this gap, in this paper, we first formulate the debias problem in Figure 1(b) from the perspective of causal inference and extend the definition of potential outcomes to be compatible in the presence of interference, then introduce a learnable treatment representation to capture such interference. Based on the extended potential outcome and treatment representation, we propose a novel ideal loss that can effectively evaluate the performance of the prediction model when both selection bias and neighborhood effect are present. We then propose two new estimators for estimating the proposed ideal loss, named neighborhood inverse propensity score (N-IPS) and neighborhood doubly robust (N-DR), respectively. Theoretical analysis shows that the proposed N-IPS and N-DR estimators can achieve unbiased learning in the presence of both selection bias and neighborhood effect, while the previous debiasing estimators cannot result in unbiased learning without imposing extra strong assumptions. Extensive semi-synthetic and real-world experiments are conducted to demonstrate the effectiveness of the proposed methods for eliminating the selection bias under interference.

## 2 PRELIMINARIES: PREVIOUS SELECTION BIAS FORMULATION

Let $u \in \mathcal{U}$ and $i \in \mathcal{I}$ be a user and an item, $x_{u,i}$, $o_{u,i}$, and $r_{u,i}$ be the feature, treatment (e.g., exposure), and feedback (e.g., conversion) of the user-item pair $(u, i)$, where $o_{u,i}$ equals 1 or 0 represents whether the item $i$ is exposed to user $u$ or not. Let $\mathcal{D} = \{(u, i) | u \in \mathcal{U}, i \in \mathcal{I}\}$ be the set of all user-item pairs. Using the potential outcome framework (Rubin, 1974; Neyman, 1990), let $r_{u,i}(1)$ be the potential feedback that would be observed if item $i$ had been exposed to user $u$ (i.e., $o_{u,i}$ had been set to 1). The potential feedback $r_{u,i}(1)$ is observed only when $o_{u,i} = 1$, otherwise it is missing. Then ignoring the missing $r_{u,i}(1)$ and training the prediction model directly with the exposed data suffers from selection bias, since the exposure is not random and is affected by various factors.

In the absence of neighborhood effect, the potential feedback $r_{u,i}(1)$ represents the user's preference by making intervention $o_{u,i} = 1$. To predict $r_{u,i}(1)$ for all $(u, i) \in \mathcal{D}$, let $\hat{r}_{u,i} \triangleq f_\theta(x_{u,i})$ be a prediction model parameterized with $\theta$. Denote $\hat{\mathbf{R}} \in \mathbb{R}^{|\mathcal{U}| \times |\mathcal{I}|}$ as the predicted potential feedback matrix with each element being $\hat{r}_{u,i}$. If all the potential feedback $\{r_{u,i}(1) | (u, i) \in \mathcal{D}\}$ were observed, the ideal loss for training the prediction model $\hat{r}_{u,i}$ is formally defined as

$$\mathcal{L}_{\text{ideal}}(\hat{\mathbf{R}}) = |\mathcal{D}|^{-1} \sum_{(u,i) \in \mathcal{D}} \delta(\hat{r}_{u,i}, r_{u,i}(1)),$$

where $\delta(\cdot, \cdot)$ is a pre-defined loss function, e.g., the squared loss $(r_{u,i}(1) - \hat{r}_{u,i})^2$. However, since $r_{u,i}(1)$ is missing when $o_{u,i} = 0$, the ideal loss cannot be computed directly from observational data. To tackle this problem, many debiasing methods are developed to address the selection bias by establishing unbiased estimators of $\mathcal{L}_{\text{ideal}}(\hat{\mathbf{R}})$, such as error imputation based (EIB) method (Hernández-Lobato et al., 2014), inverse propensity scoring (IPS) method (Schnabel et al., 2016), self-normalized

IPS (SNIPS) method (Swaminathan and Joachims, 2015), and doubly robust (DR) methods (Wang et al., 2019; Chen et al., 2021a; Dai et al., 2022; Li et al., 2023e). We summarize the causal parameter of interest and the corresponding estimation methods in the previous studies as follows.

- For the causal parameter of interest, previous studies assume the targeted user preference $r_{u,i}(o_{u,i} = 1)$ depends only on the treatment status $o_{u,i} = 1$. Then the ideal loss is defined using the sample average of $\delta(\hat{r}_{u,i}, r_{u,i}(o_{u,i} = 1))$.
- For the methods of estimating the causal parameter of interest, previous studies have made extensive efforts to estimate the probability $\mathbb{P}(o_{u,i} = 1 \mid x_{u,i})$, called propensity, i.e., the probability of item $i$ exposed to user $u$ given the features $x_{u,i}$. Then the existing IPS and DR methods use the inverse of the propensity for weighting the observed samples.

Nevertheless, we argue that both the causal parameter and the corresponding estimation methods in the previous studies lead to the failure when eliminating the selection bias under interference.

- (Section 3) For the causal parameter of interest, as shown in Figure 1(b), in the presence of interference, both the treatment status $o_{u,i}$ and the treatment statuses $o_{\mathcal{N}_{(u,i)}}$ would affect the targeted user preference $r_{u,i}(o_{u,i}, o_{\mathcal{N}_{(u,i)}})$, instead of $r_{u,i}(o_{u,i})$ in the previous studies.
- (Section 4) For the estimation methods of the causal parameter of interest, as shown in Figure 1(b), when performing propensity-based reweighting methods, both $o_{u,i}$ and $o_{\mathcal{N}_{(u,i)}}$ from its neighbors should be considered as treatments of user-item pair $(u, i)$. Therefore, the propensity should be modeled as $\mathbb{P}(o_{u,i} = 1, o_{\mathcal{N}_{(u,i)}} \mid x_{u,i})$ instead of $\mathbb{P}(o_{u,i} = 1 \mid x_{u,i})$ in previous studies, which motivates us to design new IPS and DR estimators under interference.

## 3 Modeling Selection Bias under Neighborhood Effect

In this section, we take the neighborhood effect in RS as an interference problem in causal inference area, and then introduce a treatment representation to capture the neighborhood effect. Lastly, we propose a novel ideal loss when both selection bias and neighborhood effect are present.

### 3.1 Beyond "No Interference" Assumption in Previous Studies

In the presence of neighborhood effect, the value of $r_{u,i}(1)$ depends on not only the user's preference but also the neighborhood effect, therefore we cannot distinguish the influence of user preference and the neighborhood effect, even if all the potential outcomes $\{r_{u,i}(1) : (u, i) \in \mathcal{D}\}$ are known. Conceptually, the neighborhood effect will cause the value of $r_{u,i}(1)$ relying on the exposure status $o_{u',i'}$ and the feedback $r_{u',i'}$ for some other user-item pairs $(u', i') \neq (u, i)$. Formally, we say that interference exists when a treatment on one unit has an effect on the outcome of another unit (Ogburn and VanderWeele, 2014; Forastiere et al., 2021; Sävje et al., 2021), due to the social or physical interaction among units. Previous debiasing methods rely on the "no interference" assumption, which requires the potential outcomes of a unit are not affected by the treatment status of the other units. Nevertheless, such an assumption can hardly be satisfied in real-world recommendation scenarios.

### 3.2 Proposed Causal Parameter of Interest under Interference

Let $o = (o_{1,1}, ..., o_{|\mathcal{U}|,|\mathcal{I}|})$ be the vector of exposures of all user-item pairs. For each $(u, i) \in \mathcal{D}$, we define a partition of $o = (o_{u,i}, o_{\mathcal{N}_{(u,i)}}, o_{\mathcal{N}_{-(u,i)}})$, where $\mathcal{N}_{(u,i)}$ is all the user-item pairs affecting $(u, i)$, called the *neighbors* of $(u, i)$, and $\mathcal{N}_{-(u,i)}$ is all the user-item pairs not affecting $(u, i)$. When the feedback $r_{u,i}$ is further influenced by the neighborhood exposures $o_{\mathcal{N}_{(u,i)}}$, then the potential feedback of $(u, i)$ should be defined as $r_{u,i}(o_{u,i}, o_{\mathcal{N}_{(u,i)}})$ to account for the neighbourhood effect.

However, if we take $(o_{u,i}, o_{\mathcal{N}_{(u,i)}})$ as the new treatment directly, it would be a high-dimensional sparse vector when the dimension of $o_{\mathcal{N}_{(u,i)}}$ is high and the number of exposed neighbors is limited. To address this problem and capture the neighborhood effect effectively, we make an assumption on the interference mechanism leveraging the idea of representation learning (Johansson et al., 2016).

**Assumption 1** (Neighborhood Treatment Representation). *There exists a representation vector* $\phi : \{0,1\}^{|\mathcal{N}_{(u,i)}|} \to \mathcal{G}$, *if* $\phi(o_{\mathcal{N}_{(u,i)}}) = \phi(o'_{\mathcal{N}_{(u,i)}})$, *then* $r_{u,i}(o_{u,i}, o_{\mathcal{N}_{(u,i)}}) = r_{u,i}(o_{u,i}, o'_{\mathcal{N}_{(u,i)}})$.

The above assumption implies that the value of $r_{u,i}(o_{u,i}, \boldsymbol{o}_{\mathcal{N}_{(u,i)}})$ depends on $\boldsymbol{o}_{\mathcal{N}_{(u,i)}}$ through a specific treatment representation $\phi(\cdot)$ that summarizes the neighborhood effect. Denote $\boldsymbol{g}_{u,i} = \phi(\boldsymbol{o}_{\mathcal{N}_{(u,i)}})$, then we have $r_{u,i}(o_{u,i}, \boldsymbol{o}_{\mathcal{N}_{(u,i)}}) = r_{u,i}(o_{u,i}, \boldsymbol{g}_{u,i})$ under Assumption 1, i.e., the feedback of $(u,i)$ under individual exposure $o_{u,i}$ and treatment representation $\boldsymbol{g}_{u,i}$.

We now propose ideal loss under neighborhood effect with treatment representation level $\boldsymbol{g} \in \mathcal{G}$ as

$$\mathcal{L}_{\mathrm{ideal}}^{\mathrm{N}}(\hat{\mathbf{R}}|\boldsymbol{g}) = |\mathcal{D}|^{-1} \sum_{(u,i) \in \mathcal{D}} \delta(\hat{r}_{u,i}, r_{u,i}(o_{u,i} = 1, \boldsymbol{g})),$$

and the final ideal loss summarizes various neighborhood effects $\boldsymbol{g} \in \mathcal{G}$ as

$$\mathcal{L}_{\mathrm{ideal}}^{\mathrm{N}}(\hat{\mathbf{R}}) = \int \mathcal{L}_{\mathrm{ideal}}^{\mathrm{N}}(\hat{\mathbf{R}}|\boldsymbol{g})\pi(\boldsymbol{g})d\boldsymbol{g},$$

where $\pi(\boldsymbol{g})$ is a pre-specified probability density function of $\boldsymbol{g}$.

The proposed $\mathcal{L}_{\mathrm{ideal}}^{\mathrm{N}}(\hat{\mathbf{R}})$ forces the prediction model $\hat{r}_{u,i}$ to perform well across varying treatment representation levels $\boldsymbol{g} \in \mathcal{G}$. Thus, $\mathcal{L}_{\mathrm{ideal}}^{\mathrm{N}}(\hat{\mathbf{R}})$ is expected to control the extra bias that arises from the neighborhood effect. In comparison, the self interest and neighborhood effect are intertwined in previously used $\mathcal{L}_{\mathrm{ideal}}(\hat{\mathbf{R}})$, whereas our proposed $\mathcal{L}_{\mathrm{ideal}}^{\mathrm{N}}(\hat{\mathbf{R}})$ is very flexible due to the free choice of $\pi(\boldsymbol{g})$. The choice of $\pi(\boldsymbol{g})$ depends on the target population that we want to make predictions on. Consider an extreme case of no neighborhood effect, this corresponds to $\boldsymbol{g}_{u,i} = 0$ for all user-item pairs. In such a case, we can write $r_{u,i}(1,0)$ as $r_{u,i}(1)$ and $\mathcal{L}_{\mathrm{ideal}}^{\mathrm{N}}(\hat{\mathbf{R}})$ would reduce to $\mathcal{L}_{\mathrm{ideal}}(\hat{\mathbf{R}})$.

## 4 UNBIASED ESTIMATION AND LEARNING UNDER INTERFERENCE

In this section, we first discuss the consequence of ignoring the neighborhood effect, and then propose two novel estimators for estimating the ideal loss $\mathcal{L}_{\mathrm{ideal}}^{\mathrm{N}}(\hat{\mathbf{R}})$. Moreover, we theoretically analyze the bias, variance, optimal bandwidth, and generalization error bounds of the proposed estimators.

Before presenting the proposed debiasing methods under interference, we briefly discuss the identifiability of the ideal loss $\mathcal{L}_{\mathrm{ideal}}^{\mathrm{N}}(\hat{\mathbf{R}})$. A causal estimand is said to be identifiable if it can be written as a series of quantities that can be estimated from observed data.

**Assumption 2** (Consistency under Interference). $r_{u,i} = r_{u,i}(1, \boldsymbol{g})$ if $o_{u,i} = 1$ and $\boldsymbol{g}_{u,i} = \boldsymbol{g}$.

**Assumption 3** (Unconfoundedness under Interference). $r_{u,i}(1, \boldsymbol{g}) \perp\!\!\!\perp (o_{u,i}, \boldsymbol{G}_{u,i}) \mid x_{u,i}$.

These assumptions are common in causal inference to ensure the identifiability of causal effects. Specifically, Assumption 2 implies that $r_{u,i}(1, \boldsymbol{g})$ is observed only when $o_{u,i} = 1$ and $\boldsymbol{g}_{u,i} = \boldsymbol{g}$. Assumption 3 indicates that there is no unmeasured confounder that affects both $r_{u,i}$ and $(o_{u,i}, \boldsymbol{g}_{u,i})$.

**Theorem 1** (Identifiability). *Under Assumptions 1–3, $\mathcal{L}_{\mathrm{ideal}}^{\mathrm{N}}(\hat{\mathbf{R}}|\boldsymbol{g})$ and $\mathcal{L}_{\mathrm{ideal}}^{\mathrm{N}}(\hat{\mathbf{R}})$ are identifiable.*

Theorem 1 ensures the identifiability of the proposed ideal loss $\mathcal{L}_{\mathrm{ideal}}^{\mathrm{N}}(\hat{\mathbf{R}})$. Let $\mathbb{E}$ denote the expectation on the target population $\mathcal{D}$, and $p(\cdot)$ denotes the probability density function of $\mathbb{P}$.

### 4.1 EFFECT OF IGNORING INTERFERENCE

The widely used ideal loss $\mathcal{L}_{\mathrm{ideal}}(\hat{\mathbf{R}})$ under no neighborhood effect is generally different from the proposed ideal loss $\mathcal{L}_{\mathrm{ideal}}^{\mathrm{N}}(\hat{\mathbf{R}})$ in the presence of neighborhood effect. Next, we establish the connection between these two loss functions, to deepen the understanding of the methods of considering/ignoring neighborhood effect. For brevity, we let $\delta_{u,i}(\boldsymbol{g}) = \delta(\hat{r}_{u,i}, r_{u,i}(1, \boldsymbol{g}))$ hereafter.

**Theorem 2** (Link to Selection Bias). *Under Assumptions 1–3,*

*(a) if $\boldsymbol{g}_{u,i} \perp\!\!\!\perp o_{u,i} \mid x_{u,i}$, $\mathcal{L}_{\mathrm{ideal}}^{\mathrm{N}}(\hat{\mathbf{R}}) = \mathcal{L}_{\mathrm{ideal}}(\hat{\mathbf{R}})$;*

*(b) if $\boldsymbol{g}_{u,i} \not\perp\!\!\!\perp o_{u,i} \mid x_{u,i}$, $\mathcal{L}_{\mathrm{ideal}}^{\mathrm{N}}(\hat{\mathbf{R}}) - \mathcal{L}_{\mathrm{ideal}}(\hat{\mathbf{R}})$ is equal to*

$$\int \mathbb{E}\Big[\mathbb{E}\{\delta_{u,i}(\boldsymbol{g})|x_{u,i}\} \cdot \Big\{p(\boldsymbol{g}_{u,i} = \boldsymbol{g}|x_{u,i}) - p(\boldsymbol{g}_{u,i} = \boldsymbol{g}|x_{u,i}, o_{u,i} = 1)\Big\}\Big]\pi(\boldsymbol{g})d\boldsymbol{g}.$$

From Theorem 2(a), if the individual and neighborhood exposures are independent conditional on $x_{u,i}$, then $\mathcal{L}^{\mathrm{N}}_{\mathrm{ideal}}(\hat{\mathbf{R}})$ is equal to $\mathcal{L}_{\mathrm{ideal}}(\hat{\mathbf{R}})$, which indicates that the existing debiasing methods neglecting neighborhood effect are also unbiased estimator of $\mathcal{L}^{\mathrm{N}}_{\mathrm{ideal}}(\hat{\mathbf{R}})$. This is intuitively reasonable since in such a case, the neighborhood effect randomly influences $o_{u,i}$ conditional on $x_{u,i}$, and the effect of neighbors would be smoothed out in an average sense. Theorem 2(b) shows that a bias would arise when $\boldsymbol{g}_{u,i} \not\perp o_{u,i} \mid x_{u,i}$, and the bias mainly depends on the association between $o_{u,i}$ and $\boldsymbol{g}_{u,i}$ conditional on $x_{u,i}$, i.e., $p(\boldsymbol{g}_{u,i} = \boldsymbol{g}|x_{u,i} = x) - p(\boldsymbol{g}_{u,i} = \boldsymbol{g}|x_{u,i} = x, o_{u,i} = 1)$.

## 4.2 PROPOSED UNBIASED ESTIMATORS

To derive an unbiased estimator of $\mathcal{L}^{\mathrm{N}}_{\mathrm{ideal}}(\hat{\mathbf{R}})$, it suffices to find an unbiased estimator of $\mathcal{L}^{\mathrm{N}}_{\mathrm{ideal}}(\hat{\mathbf{R}}|\boldsymbol{g})$. Motivated by Schnabel et al. (2016), an intuitive solution is to take $(o_{u,i}, \boldsymbol{g}_{u,i})$ as a joint treatment, then the IPS estimator of $\mathcal{L}^{\mathrm{N}}_{\mathrm{ideal}}(\hat{\mathbf{R}}|\boldsymbol{g})$ should be $|\mathcal{D}|^{-1} \sum_{(u,i)\in\mathcal{D}} \mathbb{I}\{o_{u,i} = 1, \boldsymbol{g}_{u,i} = \boldsymbol{g}\} \cdot \delta_{u,i}(\boldsymbol{g})/p_{u,i}(\boldsymbol{g})$, where $\mathbb{I}(\cdot)$ is an indicator function, $p_{u,i}(\boldsymbol{g}) = p(o_{u,i} = 1, \boldsymbol{g}_{u,i} = \boldsymbol{g}|x_{u,i})$ is the propensity score. Clearly, this strategy works if $\boldsymbol{g}_{u,i}$ is a binary or multi-valued random variable. However, if $\boldsymbol{g}_{u,i}$ has a continuous probability density, the above estimator is numerically infeasible even if theoretically feasible, since almost all $\mathbb{I}\{o_{u,i} = 1, \boldsymbol{g}_{u,i} = \boldsymbol{g}\}$ will be zero in such a case.

To tackle this problem, we propose a novel kernel-smoothing based neighborhood IPS (N-IPS) estimator of $\mathcal{L}^{\mathrm{N}}_{\mathrm{ideal}}(\hat{\mathbf{R}}|\boldsymbol{g})$, which is given as

$$\mathcal{L}^{\mathrm{N}}_{\mathrm{IPS}}(\hat{\mathbf{R}}|\boldsymbol{g}) = |\mathcal{D}|^{-1} \sum_{(u,i)\in\mathcal{D}} \frac{\mathbb{I}(o_{u,i} = 1) \cdot K\left((\boldsymbol{g}_{u,i} - \boldsymbol{g})/h\right) \cdot \delta_{u,i}(\boldsymbol{g})}{h \cdot p_{u,i}(\boldsymbol{g})},$$

where $h$ is a bandwidth (smoothing parameter) and $K(\cdot)$ is a symmetric kernel function (Fan and Gijbels, 1996; Li and Racine, 2023; Wu et al., 2024) that satisfies $\int K(t)dt = 1$ and $\int tK(t)dt = 1$. For example, Epanechnikov kernel $K(t) = 3(1 - t^2)\mathbb{I}\{|t| \leq 1\}/4$ and Gaussian kernel $K(t) = \exp(-t^2/2)/\sqrt{2\pi}$ for $t \in \mathbb{R}$. For ease of presentation, we state the results for a scalar $\boldsymbol{g}$. Similar conclusions can be derived for multi-dimensional $\boldsymbol{g}$ and we put them in Appendix C.

Similarly, the kernel-smoothing based neighborhood DR (N-DR) estimator can be constructed by

$$\mathcal{L}^{\mathrm{N}}_{\mathrm{DR}}(\hat{\mathbf{R}}|\boldsymbol{g}) = |\mathcal{D}|^{-1} \sum_{(u,i)\in\mathcal{D}} \left[\hat{\delta}_{u,i}(\boldsymbol{g}) + \frac{\mathbb{I}(o_{u,i} = 1) \cdot K\left((\boldsymbol{g}_{u,i} - \boldsymbol{g})/h\right) \cdot \{\delta_{u,i}(\boldsymbol{g}) - \hat{\delta}_{u,i}(\boldsymbol{g})\}}{h \cdot p_{u,i}(\boldsymbol{g})}\right],$$

where $\hat{\delta}_{u,i}(\boldsymbol{g}) = \delta(\hat{r}_{u,i}, m(x_{u,i}, \phi_{\boldsymbol{g}}))$ is the imputed error of $\delta_{u,i}(\boldsymbol{g})$, and $m(x_{u,i}, \phi_{\boldsymbol{g}})$ is an imputation model of $r_{u,i}(1, \boldsymbol{g})$. The imputation model is trained by minimizing the training loss

$$\mathcal{L}^{\mathrm{N-DR}}_e(\hat{\mathbf{R}}) = \int |\mathcal{D}|^{-1} \sum_{(u,i)\in\mathcal{D}} \frac{\mathbb{I}(o_{u,i} = 1) \cdot K\left((\boldsymbol{g}_{u,i} - \boldsymbol{g})/h\right) \cdot (\delta_{u,i}(\boldsymbol{g}) - \hat{\delta}_{u,i}(\boldsymbol{g}))^2}{h \cdot p_{u,i}(\boldsymbol{g})} \pi(\boldsymbol{g})d\boldsymbol{g}.$$

Then, the corresponding N-IPS and N-DR estimators of $\mathcal{L}^{\mathrm{N}}_{\mathrm{ideal}}(\hat{\mathbf{R}})$ are given as

$$\mathcal{L}^{\mathrm{N}}_{\mathrm{IPS}}(\hat{\mathbf{R}}) = \int \mathcal{L}^{\mathrm{N}}_{\mathrm{IPS}}(\hat{\mathbf{R}}|\boldsymbol{g})\pi(\boldsymbol{g})d\boldsymbol{g}, \quad \mathcal{L}^{\mathrm{N}}_{\mathrm{DR}}(\hat{\mathbf{R}}) = \int \mathcal{L}^{\mathrm{N}}_{\mathrm{DR}}(\hat{\mathbf{R}}|\boldsymbol{g})\pi(\boldsymbol{g})d\boldsymbol{g}. \tag{1}$$

Next, we show the bias and variance of the proposed N-IPS and N-DR estimators, which rely on a standard assumption in kernel-smoothing estimation (Härdle et al., 2004; Li and Racine, 2023).

**Assumption 4** (Regularity Conditions for Kernel Smoothing). *(a) $h \to 0$ as $|\mathcal{D}| \to \infty$; (b) $|\mathcal{D}|h \to \infty$ as $|\mathcal{D}| \to \infty$; (c) $p(o_{u,i} = 1, \boldsymbol{g}_{u,i} = \boldsymbol{g} \mid x_{u,i})$ is twice differentiable with respect to $\boldsymbol{g}$.*

**Theorem 3** (Bias and Variance of N-IPS and N-DR). *Under Assumptions 1–4,*

*(a) the bias of the N-DR estimator is given as*

$$Bias(\mathcal{L}^{\mathrm{N}}_{\mathrm{DR}}(\hat{\mathbf{R}})) = \frac{1}{2}\mu_2 \int \mathbb{E}\left[\frac{\partial^2 p(o_{u,i} = 1, \boldsymbol{g}_{u,i} = \boldsymbol{g}|x_{u,i})}{\partial \boldsymbol{g}^2} \cdot \{\delta_{u,i}(\boldsymbol{g}) - \hat{\delta}_{u,i}(\boldsymbol{g})\}\right]\pi(\boldsymbol{g})d\boldsymbol{g} \cdot h^2 + o(h^2),$$

*where $\mu_2 = \int K(t)t^2dt$. The bias of N-IPS is provided in Appendix B.2;*

*(b) the variance of the N-DR estimator is given as*

$$Var(\mathcal{L}_{\mathrm{DR}}^{\mathrm{N}}(\hat{\mathbf{R}})) = \frac{1}{|\mathcal{D}|h} \int \psi(\boldsymbol{g})\pi(\boldsymbol{g})d\boldsymbol{g} + o(\frac{1}{|\mathcal{D}|h}),$$

*where $\psi(\boldsymbol{g}) = \int \frac{1}{p_{u,i}(\boldsymbol{g}')} \cdot \bar{K}(\frac{\boldsymbol{g}-\boldsymbol{g}'}{h}) \cdot \{\delta_{u,i}(\boldsymbol{g}) - \hat{\delta}_{u,i}(\boldsymbol{g})\}\{\delta_{u,i}(\boldsymbol{g}') - \hat{\delta}_{u,i}(\boldsymbol{g}')\}\pi(\boldsymbol{g}')d\boldsymbol{g}'$ is a bounded function of $\boldsymbol{g}$, $\bar{K}(\cdot) = \int K(t)K(\cdot+t)\,dt$. The variance of N-IPS is provided in Appendix B.2.*

From Theorem 3(a), the kernel-smoothing based N-DR estimator has a small bias of order $O(h^2)$, which converges to 0 as $|\mathcal{D}| \to \infty$ by Assumption 4(a). Theorem 3(b) shows that the variance of the N-DR estimator has a convergence rate of order $O(1/|\mathcal{D}|h)$. Notably, the bandwidth $h$ plays a key role in the bias-variance trade-off of the N-DR estimator: the larger the $h$, the larger the bias and the smaller the variance. The following Theorem 4 gives the optimal bandwidth for N-IPS and N-DR.

**Theorem 4** (Optimal Bandwidth of N-IPS and N-DR). *Under Assumptions 1–4, the optimal bandwidth for the N-DR estimator in terms of the asymptotic mean-squared error is*

$$h_{\mathrm{N-DR}}^* = \left[\frac{\int \psi(\boldsymbol{g})\pi(\boldsymbol{g})d\boldsymbol{g}}{4|\mathcal{D}|\left(\frac{1}{2}\mu_2 \int \mathbb{E}\left[\frac{\partial^2 p(o_{u,i}=1,\boldsymbol{g}_{u,i}=\boldsymbol{g}|x_{u,i})}{\partial \boldsymbol{g}^2} \cdot \{\delta_{u,i}(\boldsymbol{g}) - \hat{\delta}_{u,i}(\boldsymbol{g})\}\right]\pi(\boldsymbol{g})d\boldsymbol{g}\right)^2}\right]^{1/5},$$

*where $\psi(\boldsymbol{g})$ is defined in Theorem 3. The optimal bandwidth for N-IPS is provided in Appendix B.3.*

Theorem 4 shows that the optimal bandwidth of N-DR is of order $O(|\mathcal{D}|^{-1/5})$. In such a case,

$$\left[\mathrm{Bias}(\mathcal{L}_{\mathrm{DR}}^{\mathrm{N}}(\hat{\mathbf{R}}))\right]^2 = O(h^4) = O(|\mathcal{D}|^{-4/5}), \quad Var(\mathcal{L}_{\mathrm{DR}}^{\mathrm{N}}(\hat{\mathbf{R}})) = O(\frac{1}{|\mathcal{D}|h}) = O(|\mathcal{D}|^{-4/5}),$$

that is, the square of the bias has the same convergence rate as the variance.

### 4.3 PROPENSITY ESTIMATION METHOD

Different from previous debiasing methods in RS, in the presence of neighborhood effect, the propensity is defined for joint treatment that includes a binary variable $o$ and a continuous variable $\boldsymbol{g}$. To fill this gap, we consider a novel method for propensity estimation. Let $\mathbb{P}^u(\boldsymbol{g} \mid o = 1, \boldsymbol{x})$ be a uniform distribution on $\mathcal{G}$ and equals $1/c$ for all feature $\boldsymbol{x}$. Note that

$$\frac{1}{p_{u,i}(\boldsymbol{g})} = \frac{1}{\mathbb{P}(o=1 \mid \boldsymbol{x})\mathbb{P}(\boldsymbol{g} \mid o=1, \boldsymbol{x})} = \frac{c}{\mathbb{P}(o=1 \mid \boldsymbol{x})} \cdot \frac{\mathbb{P}^u(\boldsymbol{g} \mid o=1, \boldsymbol{x})}{\mathbb{P}(\boldsymbol{g} \mid o=1, \boldsymbol{x})},$$

where $\mathbb{P}(o = 1 \mid \boldsymbol{x})$ can be estimated by using the existing methods such as naive Bayes or logistic regression with or without a few unbiased ratings, respectively (Schnabel et al., 2016). To estimate the density ratio $\mathbb{P}^u(\boldsymbol{g} \mid o = 1, \boldsymbol{x})/\mathbb{P}(\boldsymbol{g} \mid o = 1, \boldsymbol{x})$, we first label the samples in the exposed data $\{(\boldsymbol{x}_{u,i}, \boldsymbol{g}_{u,i})\}_{\{(u,i):o_{u,i}=1\}}$ as positive samples ($L = 1$), then uniformly sample treatments $\boldsymbol{g}'_{u,i} \in \mathcal{G}$ to generate samples $\{(\boldsymbol{x}_{u,i}, \boldsymbol{g}'_{u,i})\}_{\{(u,i):o_{u,i}=1\}}$ with negative labels ($L = 0$). Since the data generating process ensures that $\mathbb{P}^u(\boldsymbol{x} \mid o = 1) = \mathbb{P}(\boldsymbol{x} \mid o = 1)$, we have

$$\frac{\mathbb{P}^u(\boldsymbol{g} \mid o=1, \boldsymbol{x})}{\mathbb{P}(\boldsymbol{g} \mid o=1, \boldsymbol{x})} = \frac{\mathbb{P}^u(\boldsymbol{x}, \boldsymbol{g} \mid o=1)}{\mathbb{P}(\boldsymbol{x}, \boldsymbol{g} \mid o=1)} = \frac{\mathbb{P}(\boldsymbol{x}, \boldsymbol{g} \mid L=0)}{\mathbb{P}(\boldsymbol{x}, \boldsymbol{g} \mid L=1)} = \frac{\mathbb{P}(L=1)}{\mathbb{P}(L=0)} \cdot \frac{\mathbb{P}(L=0 \mid \boldsymbol{x}, \boldsymbol{g})}{\mathbb{P}(L=1 \mid \boldsymbol{x}, \boldsymbol{g})},$$

where $\mathbb{P}(L = l \mid \boldsymbol{x}, \boldsymbol{g})$ for $l = 0$ or $1$ can be obtained by modeling $L$ with $(\boldsymbol{x}, \boldsymbol{g})$.

### 4.4 FURTHER THEORETICAL ANALYSIS

We further theoretically analyze the generalization error bounds of the proposed N-IPS and N-DR estimators. Letting $\mathcal{F}$ be the hypothesis space of prediction matrices $\hat{\mathbf{R}}$ (or prediction model $f_\theta$), we define the Rademacher complexity

$$\mathcal{R}(\mathcal{F}) = \mathbb{E}_{\boldsymbol{\sigma} \sim \{-1,+1\}^{|\mathcal{D}|}} \sup_{f_\theta \in \mathcal{F}} \left[\frac{1}{|\mathcal{D}|} \sum_{(u,i) \in \mathcal{D}} \sigma_{u,i}\delta_{u,i}(\boldsymbol{g})\right],$$

where $\boldsymbol{\sigma} = \{\sigma_{u,i} : (u,i) \in \mathcal{D}\}$ is a Rademacher sequence (Mohri et al., 2018).

**Assumption 5** (Boundedness). $1/p_{u,i}(\boldsymbol{g}) \leq M_p$, $\delta_{u,i}(\boldsymbol{g}) \leq M_\delta$, and $|\delta_{u,i}(\boldsymbol{g}) - \hat{\delta}_{u,i}(\boldsymbol{g})| \leq M_{|\delta-\hat{\delta}|}$.

Theorem 5 gives the generalization error bounds of the prediction model trained by minimizing our proposed N-IPS and N-DR estimators.

**Theorem 5** (Generalization Error Bounds of N-IPS and N-DR). *Under Assumptions 1–5 and suppose that $K(t) \leq M_K$, we have with probability at least $1 - \eta$,*

$$\mathcal{L}_{\mathrm{ideal}}^{\mathrm{N}}(\hat{\mathbf{R}}^\dagger) \leq \min_{\hat{\mathbf{R}} \in \mathcal{F}} \mathcal{L}_{\mathrm{ideal}}^{\mathrm{N}}(\hat{\mathbf{R}}) + \mu_2 M_{|\delta-\hat{\delta}|} \Big| \int \mathbb{E}\Big[\frac{\partial^2 p(o_{u,i}=1, \boldsymbol{g}_{u,i}=\boldsymbol{g}|x_{u,i})}{\partial \boldsymbol{g}^2}\Big] \pi(\boldsymbol{g}) d\boldsymbol{g} \Big| \cdot h^2$$

$$+ \frac{4 M_p M_K}{h} \mathcal{R}(\mathcal{F}) + \frac{5 M_p M_K M_{|\delta-\hat{\delta}|}}{h} \sqrt{\frac{2}{|\mathcal{D}|} \log(\frac{4}{\eta})} + o(h^2),$$

*where $\hat{\mathbf{R}}^\dagger = \arg\min_{\hat{\mathbf{R}} \in \mathcal{F}} \mathcal{L}_{\mathrm{DR}}^{\mathrm{N}}(\hat{\mathbf{R}})$ is the learned prediction model by minimizing the N-DR estimator. The generalization error bounds of the N-IPS estimator is provided in Appendix B.4.*

## 5 SEMI-SYNTHETIC EXPERIMENTS

We conduct semi-synthetic experiments using the MovieLens 100K[1] (**ML-100K**) dataset, focusing on the following two research questions (RQs): **RQ1.** Do the proposed estimators result in more accurate estimation for ideal loss compared to the previous estimators in the presence of neighborhood effect? **RQ2.** How does the neighborhood effect strength affect the estimation accuracy?

**Experimental Setup**[2]. The **ML-100K** dataset contains 100,000 missing-not-at-random (MNAR) ratings from 943 users to 1,682 movies. Following the previous studies (Schnabel et al., 2016; Wang et al., 2019; Guo et al., 2021), we first complete the full rating matrix $\mathbf{R}$ by Matrix Factorization (MF) (Koren et al., 2009), resulting in $r_{u,i} \in \{1,2,3,4,5\}$, and then set propensity $p_{u,i} = p\alpha^{\max(0,4-r_{u,i})}$ with $\alpha = 0.5$ to model MNAR effect (Wang et al., 2019; Guo et al., 2021). Next, to model the neighborhood effect, we compute $g_{u,i} = \mathbb{I}(\sum_{(u',i')\in\mathcal{N}_{(u,i)}} o_{u',i'} \geq c)$ with varying $c$ for 100,000 observed MNAR ratings, where $\mathcal{N}_{(u,i)} = \{(u',i') \neq (u,i) \mid u' = u \text{ or } i' = i\}$. In our experiment, $c$ is chosen to be the median of all $\sum_{(u',i')\in\mathcal{N}_{(u,i)}} o_{u',i'}$ for $(u,i) \in \mathcal{D}$. Then we complete two full rating matrices $\mathbf{R}^{g=0}$ and $\mathbf{R}^{g=1}$ with $r_{u,i}(1,g) \in \{1,2,3,4,5\}$ by MF, using $\{(u,i) \mid o_{u,i} = 1, g_{u,i} = 0\}$ and $\{(u,i) \mid o_{u,i} = 1, g_{u,i} = 1\}$ respectively.

**Experimental Details**. The computation of the ideal loss needs both a ground-truth rating matrix and a predicted rating matrix. Therefore, we generate the following six predicted matrices $\hat{\mathbf{R}}$:

- **ONE**: The predicted rating matrix $\hat{\mathbf{R}}$ is identical to the true rating matrix, except that $|\{(u,i) \mid r_{u,i} = 5\}|$ randomly selected true ratings of 1 are flipped to 5. This means half of the predicted fives are true five, and half are true one.
- **THREE**: Same as **ONE**, but flipping true rating of 3.
- **FOUR**: Same as **ONE**, but flipping true rating of 4.
- **ROTATE**: For each predicted rating $\hat{r}_{u,i} = r_{u,i} - 1$ when $r_{u,i} \geq 2$, and $\hat{r}_{u,i} = 5$ when $r_{u,i} = 1$.
- **SKEW**: Predicted $\hat{r}_{u,i}$ are sampled from the Gaussian distribution $\mathcal{N}(\mu = r_{u,i}, \sigma = (6 - r_{u,i})/2)$, and clipped to the interval $[1,5]$.
- **CRS**: Set $\hat{r}_{u,i} = 2$ if $r_{u,i} \leq 3$, otherwise, set $\hat{r}_{u,i} = 4$.

To consider the neighborhood effect, we assume that each user-item pair in the uniform data has an equal probability of having $g_{u,i} = 0$ and $g_{u,i} = 1$, that is, $\pi(g) = 0.5$ for $g \in \{0,1\}$. Thus, $\mathcal{L}_{\mathrm{ideal}}^{\mathrm{N}}(\hat{\mathbf{R}}) = |\mathcal{D}|^{-1} \sum_{(u,i)\in\mathcal{D}} \{\delta(\hat{r}_{u,i}, r_{u,i}(1, g = 0)) + \delta(\hat{r}_{u,i}, r_{u,i}(1, g = 1))\}/2$, where $\delta(\cdot,\cdot)$ is the mean absolute error (MAE). We follow the previous studies (Guo et al., 2021; Li et al., 2023b) to adopt relative absolute error (RE) to measure the estimation accuracy, which is defined as $\mathrm{RE}(\mathcal{L}_{\mathrm{est}}) = |\mathcal{L}_{\mathrm{ideal}}^{\mathrm{N}}(\hat{\mathbf{R}}) - \mathcal{L}_{\mathrm{est}}(\hat{\mathbf{R}})|/\mathcal{L}_{\mathrm{ideal}}^{\mathrm{N}}(\hat{\mathbf{R}})$, where $\mathcal{L}_{\mathrm{est}}$ denotes the ideal loss estimation by the estimator. The smaller the RE, the higher the estimation accuracy (see Appendix E for more details).

**Performance Analysis**. We take three propensity-based estimators: IPS, DR, and MRDR as baselines (see Section 6 for baselines introduction). The results are shown in Table 1. First, the REs of our

---

[1]https://grouplens.org/datasets/movielens/100k/

[2]Our codes and datasets are available at https://github.com/haoxuanli-pku/ICLR24-Interference.

Table 1: Relative error on six prediction metrics. The best results are bolded.

|  | ONE | THREE | FOUR | ROTATE | SKEW | CRS |
|---|---|---|---|---|---|---|
| Naive | 0.8612 ± 0.0068 | 1.0011 ± 0.0075 | 1.0471 ± 0.0077 | 0.2781 ± 0.0019 | 0.3538 ± 0.0038 | 0.3419 ± 0.0030 |
| IPS | 0.4766 ± 0.0060 | 0.5501 ± 0.0056 | 0.5731 ± 0.0057 | 0.1434 ± 0.0040 | 0.1969 ± 0.0046 | 0.1885 ± 0.0028 |
| N-IPS | **0.2383 ± 0.0066** | **0.2670 ± 0.0069** | **0.2829 ± 0.0062** | **0.0417 ± 0.0043** | **0.1024 ± 0.0051** | **0.0966 ± 0.0029** |
| DR | 0.4247 ± 0.0088 | 0.4637 ± 0.0093 | 0.4661 ± 0.0096 | 0.0571 ± 0.0021 | 0.1938 ± 0.0043 | 0.0565 ± 0.0020 |
| N-DR | **0.3089 ± 0.0088** | **0.3533 ± 0.0091** | **0.3577 ± 0.0092** | **0.0339 ± 0.0031** | **0.1219 ± 0.0039** | **0.0511 ± 0.0026** |
| MRDR | 0.2578 ± 0.0070 | 0.2639 ± 0.0071 | 0.2611 ± 0.0073 | 0.1001 ± 0.0025 | 0.1538 ± 0.0038 | 0.0156 ± 0.0021 |
| N-MRDR | **0.0622 ± 0.0065** | **0.0520 ± 0.0065** | **0.0503 ± 0.0064** | **0.0456 ± 0.0037** | **0.0672 ± 0.0038** | **0.0042 ± 0.0022** |

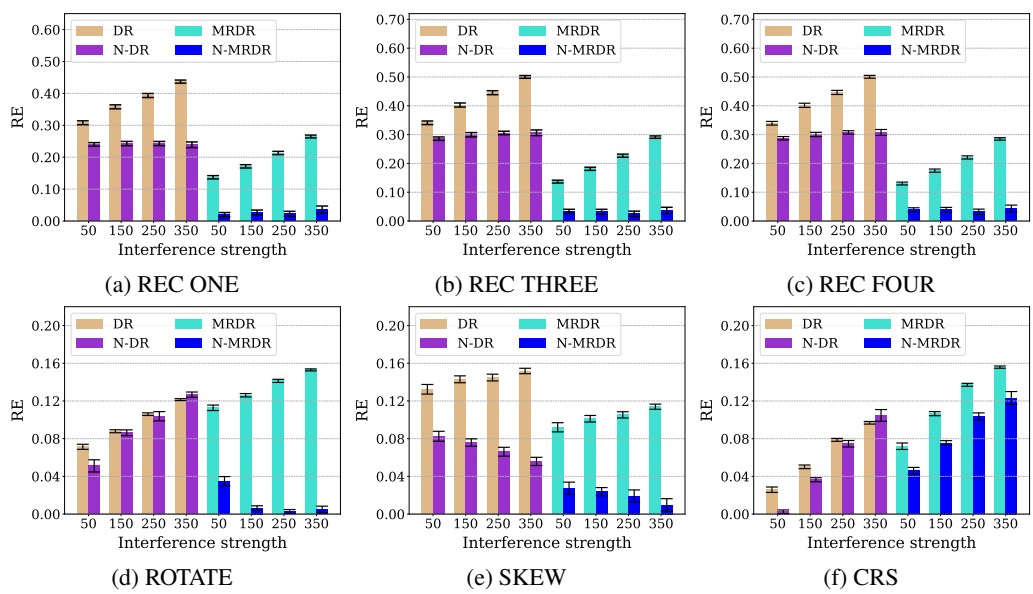

Figure 2: The effect of mask numbers as interference strength on RE on six prediction matrices.

estimators are significantly lower compared to the corresponding previous estimators, which indicates that our estimators are able to estimate the ideal loss accurately in the presence of neighborhood effect. In addition, to investigate how the neighborhood effect affects the estimation error, we randomly mask some user rows and item columns before sampling $o_{u,i}$, which results in $p_{u,i} = 0$ for the masked user-item pairs. For unmasked user-item pairs, we raise their propensities such that the expected total number of observed samples remains the same, which increases the proportion of observed samples with $g_{u,i} = 1$ to strengthen the neighborhood effect. Figure 2 shows the RE of the estimators with varying neighborhood effects. Our methods stably outperform the previous methods in all scenarios, which verifies that our methods are robust to the increased neighborhood effect.

## 6 REAL-WORLD EXPERIMENTS

**Dataset and Experiment Details.** We verify the effectiveness of the proposed estimators on three real-world datasets: **Coat** contains 6,960 MNAR ratings and 4,640 missing-at-random (MAR) ratings. **Yahoo! R3** contains 311,704 MNAR ratings and 54,000 MAR ratings. **KuaiRec** contains 4,676,570 video watching ratio records from 1,411 users for 3,327 videos. We pre-specify three neighborhood choices for a user-item pair in MNAR data: (1) using user historical behavior, (2) using the purchase history of an item, and (3) using the interaction of users and items, and let $g_{u,i}$ be the neighborhood number of the user-item pair, which is a multi-valued representation. We report the best result of our methods among the three neighborhood choices using MSE, AUC, and NDCG@$K$ as the evaluation protocols, where $K = 5$ for **Coat** and **Yahoo! R3** and $K = 50$ for **KuaiRec**. We adopt both the Gaussian kernel and Epanechnikov kernel as the kernel function for implementing our proposed N-IPS, N-DR-JL, and N-MRDR (see Appendix F for more details).

**Baselines.** We take **Matrix Factorization (MF)** (Koren et al., 2009) as the base model and consider the following debiasing baselines: **IPS** (Schnabel et al., 2016; Saito et al., 2020), **SNIPS** (Schnabel

Table 2: Performance of MSE, AUC, and NDCG@5 on three real-world datasets. The best six results are bolded, and the best baseline is underlined.

| Dataset | Coat | | | Yahoo! R3 | | | KuaiRec | | |
|---|---|---|---|---|---|---|---|---|---|
| Method | MSE ↓ | AUC ↑ | N@5 ↑ | MSE ↓ | AUC ↑ | N@5 ↑ | MSE ↓ | AUC ↑ | N@50 ↑ |
| Base model (Koren et al., 2009) | 0.238 | 0.710 | 0.616 | 0.249 | 0.682 | 0.634 | 0.137 | 0.754 | 0.553 |
| + CVIB (Wang et al., 2020) | 0.222 | 0.722 | 0.635 | 0.257 | 0.683 | 0.645 | 0.103 | 0.769 | 0.563 |
| + DIB (Liu et al., 2021) | 0.242 | 0.726 | 0.629 | 0.248 | 0.687 | 0.641 | 0.142 | 0.754 | 0.556 |
| + SNIPS (Schnabel et al., 2016) | 0.208 | 0.737 | 0.636 | 0.245 | 0.687 | 0.656 | 0.048 | **0.788** | **0.576** |
| + ASIPS (Saito, 2020a) | **0.205** | 0.722 | 0.621 | 0.230 | 0.678 | 0.643 | 0.097 | 0.753 | 0.554 |
| + DAMF (Saito and Nomura, 2022) | 0.218 | 0.734 | 0.643 | 0.245 | **0.697** | 0.656 | 0.097 | 0.775 | 0.572 |
| + DR (Saito, 2020b) | 0.208 | 0.726 | 0.634 | 0.216 | 0.684 | 0.658 | **0.046** | 0.773 | 0.564 |
| + DR-BIAS (Dai et al., 2022) | 0.223 | 0.717 | 0.631 | 0.220 | 0.689 | 0.654 | **0.046** | 0.771 | 0.552 |
| + DR-MSE (Dai et al., 2022) | 0.214 | 0.720 | 0.630 | 0.222 | 0.689 | 0.657 | 0.047 | 0.769 | 0.547 |
| + MR (Li et al., 2023a) | 0.210 | 0.730 | 0.643 | 0.247 | 0.693 | 0.651 | 0.114 | 0.780 | 0.573 |
| + TDR (Li et al., 2023b) | 0.229 | 0.710 | 0.634 | 0.234 | 0.674 | 0.662 | 0.134 | 0.769 | 0.573 |
| + TDR-JL (Li et al., 2023b) | 0.216 | 0.734 | 0.639 | 0.248 | 0.684 | 0.654 | 0.121 | 0.771 | 0.560 |
| + SDR (Li et al., 2023e) | **0.208** | 0.736 | 0.642 | 0.210 | 0.690 | 0.655 | 0.116 | 0.775 | 0.574 |
| + IPS (Schnabel et al., 2016) | 0.214 | 0.718 | 0.626 | 0.221 | 0.681 | 0.644 | 0.097 | 0.752 | 0.554 |
| + N-IPS [LR, Gaussian] | 0.212 | 0.742 | **0.678** | 0.226 | 0.693 | **0.664** | 0.092 | **0.796** | **0.585** |
| + N-IPS [LR, Epanechnikov] | 0.224 | **0.746** | 0.645 | 0.242 | **0.703** | **0.673** | 0.094 | **0.794** | **0.582** |
| + N-IPS [NB, Gaussian] | **0.206** | **0.744** | 0.648 | **0.196** | 0.693 | 0.658 | 0.049 | 0.785 | **0.579** |
| + N-IPS [NB, Epanechnikov] | 0.210 | **0.753** | 0.646 | **0.197** | 0.685 | 0.653 | **0.047** | 0.755 | 0.562 |
| + DR-JL (Wang et al., 2019) | 0.211 | 0.721 | 0.620 | 0.224 | 0.682 | 0.646 | 0.050 | 0.764 | 0.526 |
| + N-DR-JL [LR, Gaussian] | 0.231 | 0.731 | **0.651** | 0.247 | **0.698** | **0.664** | 0.113 | 0.779 | 0.537 |
| + N-DR-JL [LR, Epanechnikov] | 0.235 | 0.741 | **0.655** | 0.251 | 0.693 | **0.663** | 0.108 | 0.784 | 0.552 |
| + N-DR-JL [NB, Gaussian] | **0.204** | **0.748** | 0.650 | **0.198** | 0.691 | 0.653 | 0.049 | 0.778 | 0.574 |
| + N-DR-JL [NB, Epanechnikov] | 0.209 | **0.744** | 0.648 | **0.191** | 0.681 | 0.637 | **0.046** | 0.786 | 0.570 |
| + MRDR-JL (Guo et al., 2021) | 0.214 | 0.721 | 0.631 | 0.215 | 0.686 | 0.650 | 0.047 | 0.777 | 0.554 |
| + N-MRDR-JL [LR, Gaussian] | 0.217 | 0.728 | **0.662** | 0.252 | **0.697** | **0.666** | 0.107 | 0.785 | 0.539 |
| + N-MRDR-JL [LR, Epanechnikov] | 0.233 | 0.734 | **0.656** | 0.253 | **0.695** | **0.666** | 0.097 | **0.791** | 0.560 |
| + N-MRDR-JL [NB, Gaussian] | **0.208** | 0.742 | **0.651** | **0.206** | **0.694** | 0.663 | **0.045** | **0.793** | **0.583** |
| + N-MRDR-JL [NB, Epanechnikov] | **0.207** | **0.756** | 0.635 | **0.194** | 0.690 | 0.644 | **0.044** | **0.802** | **0.587** |

et al., 2016), **DR-JL** (Wang et al., 2019), **ASIPS** (Saito, 2020a), **CVIB** (Wang et al., 2020), **DR** (Saito, 2020b), **MRDR-JL** (Guo et al., 2021), **DIB** (Liu et al., 2021), **DAMF** (Saito and Nomura, 2022), **DR-BIAS** (Dai et al., 2022), **DR-MSE** (Dai et al., 2022), **MR** (Li et al., 2023a), **Stable-DR** (Li et al., 2023e), **TDR** (Li et al., 2023b), and **TDR-JL** (Li et al., 2023b). Following previous studies (Schnabel et al., 2016; Wang et al., 2019), for all baseline methods requiring propensity estimation, we adopt naive Bayes (NB) method using 5% MAR ratings for training the propensity model. For our proposed methods, we also adopt logistic regression (LR) to estimate the propensities without MAR ratings.

**Real-World Debiasing Performance.** Table 2 shows the performance of the baselines and our methods on three datasets. Compared with the base model, the debiasing methods achieve better performance. Notably, the proposed methods can stably outperform the baseline methods in all metrics, showing that our methods can effectively take the neighbor effect into account. This also provides empirical evidence of the existence of the neighborhood effect in real-world datasets. The proposed methods show competitive performance whether the MAR data are accessible (NB) or not (LR), and perform similarly in the case of adopting the Gaussian kernel or Epanechnikov kernel.

## 7 CONCLUSION

In this paper, we study the problem of selection bias in the presence of neighborhood effect. First, we formulate the neighborhood effect in RS as an interference problem in causal inference. Next, a neighborhood treatment representation vector is introduced to reduce the dimension and sparsity of the neighborhood treatments. Based on it, we reformulate the potential feedback and propose a novel ideal loss that can be used to deal with selection bias in the presence of neighborhood effect. Then, we propose two novel kernel-smoothing based neighborhood estimators for the ideal loss, which allows the neighborhood treatment representation vector to have continuous probability density. We systematically analyze the properties of the proposed estimators, including the bias, variance, optimal bandwidth, and generalization error bounds. In addition, we also theoretically establish the connection between the debiasing methods considering and ignoring the neighborhood effect. Extensive experiments are conducted on semi-synthetic and real-world data to demonstrate the effectiveness of our approaches. A limitation of this work is that the hypothesis space $\mathcal{G}$ of $g$ relies on prior knowledge, and it is not obvious to choose it in practice. We leave it for our future work.

## 8    ACKNOWLEDGEMENT

This work was supported in part by National Natural Science Foundation of China (No. 623B2002, 12301370).

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

## A    RELATED WORK

**Selection Bias in Recommender System.** Recommender system (RS) plays an important role in the current era of information explosion (Huang et al., 2023; Lv et al., 2023; 2024). However, due to the self-selection by the users, the data collected by RS will often contain selection bias, which will result in the distribution of training data (observed population) different from that of test data (target population), leading to the challenges of achieving unbiased learning (Wang et al., 2022b; 2023b; Zou et al., 2023; Wang et al., 2023a; 2024). If we learn the model directly on the training data without debiasing, it will result in a sub-optimal performance on the test data (Wang et al., 2023c; Zhang et al., 2023; Bai et al., 2024; Zhang et al., 2024). Many methods have been developed to address selection bias (Wu et al., 2022; Wang et al., 2023d; Luo et al., 2024), for instance, Marlin and Zemel (2009); Steck (2010) discussed the error-imputation-based methods, Schnabel et al. (2016) recommended using the inverse propensity scoring (IPS) method for unbiased learning. Wang et al. (2019) proposed a doubly robust (DR) joint learning method and achieved superior performance. Subsequently, various novel model structures and algorithms are designed to enhance the DR method, such as Guo et al. (2021); Dai et al. (2022); Li et al. (2024), which proposed new DR methods by further reducing the bias or variance of the DR estimator. Zhang et al. (2020) proposed multi-task learning through parameter sharing between the propensity and prediction model, Chen et al. (2021a); Wang et al. (2021); Li et al. (2023c) proposed using a small uniform dataset to enhance the performance of prediction models, and Ding et al. (2022) proposed an adversarial learning-based framework to address the unmeasured confounders. However, a user's feedback on an item may receive influence from the other user-item pairs (Zheng et al., 2021). To fill this gap, Chen et al. (2021b) focuses on the task of learning to rank (LTR), addressing position bias using implicit feedback data. They consider other user-item interactions as confounders from a counterfactual perspective and use embedding as a proxy confounder to capture the influence of other user-item interactions. Different from Chen et al. (2021b), our paper focuses on the selection bias in the context of rating prediction, and regards other user-item interactions as a new treatment from the perspective of interference in causal inference. We formulate the influence of other user-item interactions as an interference problem in causal inference and introduce a treatment representation to capture the influence. On this basis, we propose a novel ideal loss that can be used to deal with selection bias in the presence of interference.

**Interference in Causal Inference.** Interference is a common problem in observational studies, and the effects of interference are also called spillover effects in economics or peer effects in social sciences (Forastiere et al., 2021). Early literature focuses on the case of partial interference (Hong and Raudenbush, 2006; Sobel, 2006; Hudgens and Halloran, 2008; Tchetgen and VanderWeele, 2012; Ferracci et al., 2014), i.e., the sample can be divided into multiple groups, with interference between units in the same group, while units between groups are independent. Recent works have attempted to further relax the partial interference assumption by allowing for a wide variety of interference patterns (Ogburn and VanderWeele, 2014; Aronow and Samii, 2017), such as direct interference (Forastiere et al., 2021), interference by contagion (Ogburn and VanderWeele, 2017), allocational interference (Ogburn and VanderWeele, 2014), or their hybrids (Tchetgen et al., 2021). These studies differ from ours in several important ways: (1) their goal is to estimate the main effect and neighborhood effect, while our goal is to achieve unbiased learning of the prediction model that is more challenging and need to carefully design the loss as well as the training algorithm to mitigate the neighborhood effect; (2) they do not use treatment representation and the estimation does not take into account the possible cases of continuous and multi-dimensional representations.

## B    PROOFS

### B.1    PROOFS OF THEOREMS 1 AND 2

Recall that $p_{u,i}(\boldsymbol{g}) = \mathbb{P}(o_{u,i} = 1, \boldsymbol{g}_{u,i} = \boldsymbol{g}|x_{u,i})$ and $\hat{r}_{u,i} = f_\theta(x_{u,i})$ are functions of $x_{u,i}$, $\delta_{u,i}(\boldsymbol{g}) = \delta(\hat{r}_{u,i}, r_{u,i}(1, \boldsymbol{g}))$, $\mathbb{P}$ and $\mathbb{E}$ denote the distribution and expectation on the target population $\mathcal{D}$, and $p(\cdot)$ denotes the probability density function of $\mathbb{P}$.

**Theorem 1** (Identifiability). *Under Assumptions 1–3, $\mathcal{L}_{\text{ideal}}^{\text{N}}(\hat{\mathbf{R}}|\boldsymbol{g})$ and $\mathcal{L}_{\text{ideal}}^{\text{N}}(\hat{\mathbf{R}})$ are identifiable.*

*Proof of Theorem 1.* Since

$$\mathcal{L}_{\text{ideal}}^{\text{N}}(\hat{\mathbf{R}}) = \int \mathcal{L}_{\text{ideal}}^{\text{N}}(\hat{\mathbf{R}}|\boldsymbol{g})\pi(\boldsymbol{g})d\boldsymbol{g},$$

it suffices to show that $\mathcal{L}_{\text{ideal}}^{\text{N}}(\hat{\mathbf{R}}|\boldsymbol{g})$ is identifiable. This follows immediately from the following equations

$$
\begin{aligned}
\mathcal{L}_{\text{ideal}}^{\text{N}}(\hat{\mathbf{R}}|\boldsymbol{g}) &= \mathbb{E}[\delta(\hat{r}_{u,i}, r_{u,i}(1, \boldsymbol{g}))] \\
&= \mathbb{E}\Big[\mathbb{E}\Big\{\delta(\hat{r}_{u,i}, r_{u,i}(1, \boldsymbol{g})) \mid x_{u,i}\Big\}\Big] \quad \text{(the law of iterated expectations)} \\
&= \mathbb{E}\Big[\mathbb{E}\Big\{\delta(\hat{r}_{u,i}, r_{u,i}(1, \boldsymbol{g})) \mid x_{u,i}, o_{u,i} = 1, \boldsymbol{g}_{u,i} = \boldsymbol{g}\Big\}\Big] \quad \text{(Assumption 3)} \\
&= \mathbb{E}\Big[\mathbb{E}\Big\{\delta(\hat{r}_{u,i}, r_{u,i}) \mid x_{u,i}, o_{u,i} = 1, \boldsymbol{g}_{u,i} = \boldsymbol{g}\Big\}\Big] \quad \text{(Assumption 2)} \\
&= \int\int \delta(\hat{r}_{u,i}, r_{u,i})p(r_{u,i}|x_{u,i}, o_{u,i} = 1, \boldsymbol{g}_{u,i} = \boldsymbol{g})p(x_{u,i})dr_{u,i}dx_{u,i}.
\end{aligned}
$$

$\square$

**Theorem 2** (Link to Selection Bias). *Under Assumptions 1–3,*

*(a) if $\boldsymbol{g}_{u,i} \perp\!\!\!\perp o_{u,i} \mid x_{u,i}$, $\mathcal{L}_{\text{ideal}}^{\text{N}}(\hat{\mathbf{R}}) = \mathcal{L}_{\text{ideal}}(\hat{\mathbf{R}})$;*

*(b) if $\boldsymbol{g}_{u,i} \not\!\perp\!\!\!\perp o_{u,i} \mid x_{u,i}$, $\mathcal{L}_{\text{ideal}}^{\text{N}}(\hat{\mathbf{R}}) - \mathcal{L}_{\text{ideal}}(\hat{\mathbf{R}})$ equals*

$$\int \mathbb{E}\Big[\mathbb{E}\{\delta_{u,i}(\boldsymbol{g})|x_{u,i}\} \cdot \Big\{p(\boldsymbol{g}_{u,i} = \boldsymbol{g}|x_{u,i}) - p(\boldsymbol{g}_{u,i} = \boldsymbol{g}|x_{u,i}, o_{u,i} = 1)\Big\}\Big]\pi(\boldsymbol{g})d\boldsymbol{g}.$$

*Proof of Theorem 2.* For previous methods addressing selection bias without taking into account interference, the ideal loss $\mathcal{L}_{\text{ideal}}(\hat{\mathbf{R}})$ is

$$
\begin{aligned}
\mathcal{L}_{\text{ideal}}(\hat{\mathbf{R}}) &= \mathbb{E}[\delta(\hat{r}_{u,i}, r_{u,i}(1))] \\
&= \mathbb{E}[\mathbb{E}\{\delta(\hat{r}_{u,i}, r_{u,i}(1))|x_{u,i}\}] \\
&= \mathbb{E}[\mathbb{E}\{\delta(\hat{r}_{u,i}, r_{u,i})|x_{u,i}, o_{u,i} = 1\}] \\
&= \mathbb{E}\Big[\mathbb{E}\Big\{\delta(\hat{r}_{u,i}, r_{u,i})|x_{u,i}, o_{u,i} = 1, \boldsymbol{g}_{u,i} = \boldsymbol{g}\Big\} \cdot p(\boldsymbol{g}_{u,i} = \boldsymbol{g}|x_{u,i}, o_{u,i} = 1)\Big] \\
&= \mathbb{E}\Big[\mathbb{E}\Big\{\delta_{u,i}(\boldsymbol{g})|x_{u,i}, o_{u,i} = 1, \boldsymbol{g}_{u,i} = \boldsymbol{g}\Big\} \cdot p(\boldsymbol{g}_{u,i} = \boldsymbol{g}|x_{u,i}, o_{u,i} = 1)\Big] \\
&= \mathbb{E}\Big[\mathbb{E}\Big\{\delta_{u,i}(\boldsymbol{g})|x_{u,i}\Big\} \cdot p(\boldsymbol{g}_{u,i} = \boldsymbol{g}|x_{u,i}, o_{u,i} = 1)\Big].
\end{aligned}
$$

For the proposed method addressing selection bias under interference, our newly defined ideal loss $\mathcal{L}_{\text{ideal}}^{\text{N}}(\hat{\mathbf{R}})$ is

$$
\begin{aligned}
\mathcal{L}_{\text{ideal}}^{\text{N}}(\hat{\mathbf{R}}) &= \int \mathbb{E}[\delta(\hat{r}_{u,i}, r_{u,i}(1, \boldsymbol{g}))]\pi(\boldsymbol{g})d\boldsymbol{g} \\
&= \int \mathbb{E}[\mathbb{E}\{\delta_{u,i}(\boldsymbol{g})|x_{u,i}\}]\pi(\boldsymbol{g})d\boldsymbol{g} \\
&= \int \mathbb{E}\Big[\mathbb{E}\Big\{\delta_{u,i}(\boldsymbol{g})|x_{u,i}, \boldsymbol{g}_{u,i} = \boldsymbol{g}\Big\} \cdot p(\boldsymbol{g}_{u,i} = \boldsymbol{g} \mid x_{u,i})\Big]\pi(\boldsymbol{g})d\boldsymbol{g} \\
&= \int \mathbb{E}\Big[\mathbb{E}\Big\{\delta_{u,i}(\boldsymbol{g})|x_{u,i}\Big\} \cdot p(\boldsymbol{g}_{u,i} = \boldsymbol{g} \mid x_{u,i})\Big]\pi(\boldsymbol{g})d\boldsymbol{g}.
\end{aligned}
$$

Theorem 2(b) follows immediately from these two rewritten equations. When $\boldsymbol{g}_{u,i} \perp\!\!\!\perp o_{u,i} \mid x_{u,i}$, we have $p(\boldsymbol{g}_{u,i} = \boldsymbol{g} \mid x_{u,i} = x, o_{u,i} = 1) = p(\boldsymbol{g}_{u,i} = \boldsymbol{g} \mid x_{u,i} = x)$, which leads to $\mathcal{L}_{\text{ideal}}^{\text{N}}(\hat{\mathbf{R}}) = \mathcal{L}_{\text{ideal}}(\hat{\mathbf{R}})$. This completes the proof of Theorem 2(a).

$\square$

### B.2  PROOF OF THEOREM 3

**Theorem 3** (Bias and Variance of N-IPS and N-DR). *Under Assumptions 1–4,*

*(a) the bias of the N-DR estimator is given as*

$$Bias(\mathcal{L}_{\mathrm{DR}}^{\mathrm{N}}(\hat{\mathbf{R}})) = \frac{1}{2}\mu_2 \int \mathbb{E}\Big[\frac{\partial^2 p(o_{u,i} = 1, \boldsymbol{g}_{u,i} = \boldsymbol{g}|x_{u,i})}{\partial \boldsymbol{g}^2} \cdot \{\delta_{u,i}(\boldsymbol{g}) - \hat{\delta}_{u,i}(\boldsymbol{g})\}\Big]\pi(\boldsymbol{g})d\boldsymbol{g}\cdot h^2 + o(h^2),$$

*where $\mu_2 = \int K(t)t^2 dt$. The bias of the N-IPS estimator is given as*

$$Bias(\mathcal{L}_{\mathrm{IPS}}^{\mathrm{N}}(\hat{\mathbf{R}})) = \frac{1}{2}\mu_2 \int \mathbb{E}\Big[\frac{\partial^2 p(o_{u,i} = 1, \boldsymbol{g}_{u,i} = \boldsymbol{g}|x_{u,i})}{\partial \boldsymbol{g}^2} \cdot \delta_{u,i}(\boldsymbol{g})\Big]\pi(\boldsymbol{g})d\boldsymbol{g} \cdot h^2 + o(h^2);$$

*(b) the variance of the N-DR estimator is given as*

$$Var(\mathcal{L}_{\mathrm{DR}}^{\mathrm{N}}(\hat{\mathbf{R}})) = \frac{1}{|\mathcal{D}|h} \int \psi(\boldsymbol{g})\pi(\boldsymbol{g})d\boldsymbol{g} + o(\frac{1}{|\mathcal{D}|h}),$$

*where*

$$\psi(\boldsymbol{g}) = \int \frac{1}{p_{u,i}(\boldsymbol{g}')} \cdot \bar{K}(\frac{\boldsymbol{g} - \boldsymbol{g}'}{h}) \cdot \{\delta_{u,i}(\boldsymbol{g}) - \hat{\delta}_{u,i}(\boldsymbol{g})\}\{\delta_{u,i}(\boldsymbol{g}') - \hat{\delta}_{u,i}(\boldsymbol{g}')\}\pi(\boldsymbol{g}')d\boldsymbol{g}'$$

*is a bounded function of $\boldsymbol{g}$, $\bar{K}(\cdot) = \int K(t)K(\cdot + t)dt$. The variance of the N-IPS estimator is given as*

$$Var(\mathcal{L}_{\mathrm{IPS}}^{\mathrm{N}}(\hat{\mathbf{R}})) = \frac{1}{|\mathcal{D}|h} \int \varphi(\boldsymbol{g})\pi(\boldsymbol{g})d\boldsymbol{g} + o(\frac{1}{|\mathcal{D}|h}),$$

*where*

$$\varphi(\boldsymbol{g}) = \int \frac{1}{p_{u,i}(\boldsymbol{g}')} \cdot \bar{K}(\frac{\boldsymbol{g} - \boldsymbol{g}'}{h}) \cdot \delta_{u,i}(\boldsymbol{g})\delta_{u,i}(\boldsymbol{g}')\pi(\boldsymbol{g}')d\boldsymbol{g}'$$

*is a bounded function of $\boldsymbol{g}$.*

*Proof of Theorem 3.* (a) We first show the bias of the N-IPS estimator, and the bias of the N-DR estimator can be shown similarly. For a given $\boldsymbol{g}$, we have

$$
\begin{aligned}
\mathbb{E}[\mathcal{L}_{\mathrm{IPS}}^{\mathrm{N}}(\hat{\mathbf{R}}|\boldsymbol{g})] &= \mathbb{E}\left[\frac{\mathbb{I}(o_{u,i} = 1) \cdot K\left((\boldsymbol{g}_{u,i} - \boldsymbol{g})/h\right) \cdot \delta_{u,i}(\boldsymbol{g})}{h \cdot p_{u,i}(\boldsymbol{g})}\right] \\
&= \mathbb{E}\left[\frac{1}{p_{u,i}(\boldsymbol{g})}\mathbb{E}\Big\{o_{u,i} \cdot \frac{1}{h}K\left(\frac{\boldsymbol{g}_{u,i} - \boldsymbol{g}}{h}\right)\Big|x_{u,i}\Big\} \cdot \mathbb{E}\{\delta_{u,i}(\boldsymbol{g})|x_{u,i}\}\right] \\
&= \mathbb{E}\left[\frac{1}{p_{u,i}(\boldsymbol{g})}\int\int o_{u,i}\frac{1}{h}K(\frac{\boldsymbol{g}_{u,i} - \boldsymbol{g}}{h})p(o_{u,i}, \boldsymbol{g}_{u,i}|x_{u,i})do_{u,i}d\boldsymbol{g}_{u,i} \cdot \mathbb{E}\{\delta_{u,i}(\boldsymbol{g})|x_{u,i}\}\right] \\
&= \mathbb{E}\left[\frac{1}{p_{u,i}(\boldsymbol{g})}\int \frac{1}{h}K(\frac{\boldsymbol{g}_{u,i} - \boldsymbol{g}}{h})p(o_{u,i} = 1, \boldsymbol{g}_{u,i}|x_{u,i})d\boldsymbol{g}_{u,i} \cdot \mathbb{E}\{\delta_{u,i}(\boldsymbol{g})|x_{u,i}\}\right],
\end{aligned}
$$

where the second equation follows from Assumption 3. Letting $t = (\boldsymbol{g}_{u,i} - \boldsymbol{g})/h$, we have $\boldsymbol{g}_{u,i} = \boldsymbol{g} + ht$ and $d\boldsymbol{g}_{u,i} = hdt$, and then

$$\mathbb{E}\left[\frac{1}{p_{u,i}(\boldsymbol{g})}\int\frac{1}{h}K(\frac{\boldsymbol{g}_{u,i} - \boldsymbol{g}}{h})p(o_{u,i} = 1, \boldsymbol{g}_{u,i}|x_{u,i})d\boldsymbol{g}_{u,i}\cdot\mathbb{E}\{\delta_{u,i}(\boldsymbol{g})|x_{u,i}\}\right]$$

$$= \mathbb{E}\left[\frac{1}{p_{u,i}(\boldsymbol{g})}\int K(t)p(o_{u,i} = 1, \boldsymbol{g} + ht|x_{u,i})dt\cdot\mathbb{E}\{\delta_{u,i}(\boldsymbol{g})|x_{u,i}\}\right]$$

$$= \mathbb{E}\left[\frac{1}{p_{u,i}(\boldsymbol{g})}\int K(t)\left\{p(o_{u,i} = 1, \boldsymbol{g}|x_{u,i}) + \frac{\partial p(o_{u,i} = 1, \boldsymbol{g}|x_{u,i})}{\partial\boldsymbol{g}}ht\right.\right.$$

$$\left.\left. + \frac{\partial^2 p(o_{u,i} = 1, \boldsymbol{g}|x_{u,i})}{\partial\boldsymbol{g}^2}\frac{h^2 t^2}{2} + o(h^2)\right\}dt\cdot\mathbb{E}\{\delta_{u,i}(\boldsymbol{g})|x_{u,i}\}\right]$$

$$= \mathbb{E}\left[\frac{1}{p_{u,i}(\boldsymbol{g})}\int K(t)dt\cdot p(o_{u,i} = 1, \boldsymbol{g}|x_{u,i})\cdot\mathbb{E}\{\delta_{u,i}(\boldsymbol{g})|x_{u,i}\}\right]$$

$$+ \mathbb{E}\left[\frac{1}{p_{u,i}(\boldsymbol{g})}\int K(t)tdt\cdot\frac{\partial p(o_{u,i} = 1, \boldsymbol{g}|x_{u,i})}{\partial\boldsymbol{g}}h\cdot\mathbb{E}\{\delta_{u,i}(\boldsymbol{g})|x_{u,i}\}\right]$$

$$+ \mathbb{E}\left[\frac{1}{p_{u,i}(\boldsymbol{g})}\int K(t)t^2 dt\cdot\frac{\partial^2 p(o_{u,i} = 1, \boldsymbol{g}|x_{u,i})}{\partial\boldsymbol{g}^2}\frac{h^2}{2}\cdot\mathbb{E}\{\delta_{u,i}(\boldsymbol{g})|x_{u,i}\}\right] + o(h^2)$$

$$= \mathbb{E}\left[\mathbb{E}\{\delta_{u,i}(\boldsymbol{g})|x_{u,i}\}\right] + \frac{1}{2}\mu_2\mathbb{E}\left[\frac{\partial^2 p(o_{u,i} = 1, \boldsymbol{g}|x_{u,i})}{\partial\boldsymbol{g}^2}\cdot\delta_{u,i}(\boldsymbol{g})\right]h^2 + o(h^2)$$

$$= \mathbb{E}\left[\delta_{u,i}(\boldsymbol{g})\right] + \frac{1}{2}\mu_2\mathbb{E}\left[\frac{\partial^2 p(o_{u,i} = 1, \boldsymbol{g}|x_{u,i})}{\partial\boldsymbol{g}^2}\cdot\delta_{u,i}(\boldsymbol{g})\right]h^2 + o(h^2)$$

$$= \mathcal{L}_{\text{ideal}}^{\text{N}}(\hat{\mathbf{R}}|\boldsymbol{g}) + \frac{1}{2}\mu_2\mathbb{E}\left[\frac{\partial^2 p(o_{u,i} = 1, \boldsymbol{g}|x_{u,i})}{\partial\boldsymbol{g}^2}\cdot\delta_{u,i}(\boldsymbol{g})\right]h^2 + o(h^2),$$

where the third equation is a Taylor expansion of $p(o_{u,i} = 1, \boldsymbol{g} + ht|x_{u,i})$ under Assumption 4(a). Thus, the bias of $\mathcal{L}_{\text{IPS}}^{\text{N}}(\hat{\mathbf{R}})$ is

$$\mathbb{E}[\mathcal{L}_{\text{IPS}}^{\text{N}}(\hat{\mathbf{R}})] - \mathcal{L}_{\text{ideal}}^{\text{N}}(\hat{\mathbf{R}})$$

$$= \mathbb{E}\left[\int_{\boldsymbol{g}}\left\{\mathcal{L}_{\text{IPS}}^{\text{N}}(\hat{\mathbf{R}}|\boldsymbol{g}) - \mathcal{L}_{\text{ideal}}^{\text{N}}(\hat{\mathbf{R}}|\boldsymbol{g})\right\}\pi(\boldsymbol{g})d\boldsymbol{g}\right]$$

$$= \int_{\boldsymbol{g}}\mathbb{E}\left\{\mathcal{L}_{\text{IPS}}^{\text{N}}(\hat{\mathbf{R}}|\boldsymbol{g}) - \mathcal{L}_{\text{ideal}}^{\text{N}}(\hat{\mathbf{R}}|\boldsymbol{g})\right\}\pi(\boldsymbol{g})d\boldsymbol{g}$$

$$= \frac{1}{2}\mu_2\int\mathbb{E}\left[\frac{\partial^2 p(o_{u,i} = 1, \boldsymbol{g}|x_{u,i})}{\partial\boldsymbol{g}^2}\cdot\delta_{u,i}(\boldsymbol{g})\right]\pi(\boldsymbol{g})d\boldsymbol{g}\cdot h^2 + o(h^2).$$

Likewise, for a given $\boldsymbol{g}$ and $\hat{\delta}_{u,i}(\boldsymbol{g})$, by a similar argument of proof of the bias of N-IPS estimator,

$$\mathbb{E}[\mathcal{L}_{\text{DR}}^{\text{N}}(\hat{\mathbf{R}}|\boldsymbol{g})]$$

$$= \mathbb{E}\left[\delta_{u,i}(\boldsymbol{g}) + \frac{\mathbb{I}(o_{u,i} = 1)\cdot K((\boldsymbol{g}_{u,i} - \boldsymbol{g})/h)\cdot\{\delta_{u,i}(\boldsymbol{g}) - \hat{\delta}_{u,i}(\boldsymbol{g})\}}{h\cdot p_{u,i}(\boldsymbol{g})}\right]$$

$$= \mathcal{L}_{\text{ideal}}^{\text{N}}(\hat{\mathbf{R}}|\boldsymbol{g}) + \mathbb{E}\left[\frac{1}{p_{u,i}(\boldsymbol{g})}\mathbb{E}\left\{o_{u,i}\cdot\frac{1}{h}K\left(\frac{\boldsymbol{g}_{u,i} - \boldsymbol{g}}{h}\right)\Big|x_{u,i}\right\}\cdot\mathbb{E}\{\delta_{u,i}(\boldsymbol{g}) - \hat{\delta}_{u,i}(\boldsymbol{g})|x_{u,i}\}\right]$$

$$= \mathcal{L}_{\text{ideal}}^{\text{N}}(\hat{\mathbf{R}}|\boldsymbol{g}) + \frac{1}{2}\mu_2\mathbb{E}\left[\frac{\partial^2 p(o_{u,i} = 1, \boldsymbol{g}|x_{u,i})}{\partial\boldsymbol{g}^2}\cdot\{\delta_{u,i}(\boldsymbol{g}) - \hat{\delta}_{u,i}(\boldsymbol{g})\}\right]h^2 + o(h^2).$$

Thus, the bias of $\mathcal{L}_{\text{DR}}^{\text{N}}(\hat{\mathbf{R}})$ is given as

$$\frac{1}{2}\mu_2\int\mathbb{E}\left[\frac{\partial^2 p(o_{u,i} = 1, \boldsymbol{g}|x_{u,i})}{\partial\boldsymbol{g}^2}\cdot\{\delta_{u,i}(\boldsymbol{g}) - \hat{\delta}_{u,i}(\boldsymbol{g})\}\right]\pi(\boldsymbol{g})d\boldsymbol{g}\cdot h^2 + o(h^2).$$

(b) By definition, the variance of $\mathcal{L}_{\mathrm{IPS}}^{\mathrm{N}}(\hat{\mathbf{R}})$ can be represented as

$$
\begin{aligned}
&\mathrm{Var}(\mathcal{L}_{\mathrm{IPS}}^{\mathrm{N}}(\hat{\mathbf{R}})) \\
&= \mathrm{Var}\left[\int \mathcal{L}_{\mathrm{IPS}}^{\mathrm{N}}(\hat{\mathbf{R}}|\boldsymbol{g})\pi(\boldsymbol{g})d\boldsymbol{g}\right] \\
&= \mathrm{Var}\left[\frac{1}{|\mathcal{D}|}\sum_{(u,i)\in\mathcal{D}}\int\frac{\mathbb{I}(o_{u,i}=1)}{p_{u,i}(\boldsymbol{g})}\cdot\frac{1}{h}K\left(\frac{\boldsymbol{g}_{u,i}-\boldsymbol{g}}{h}\right)\cdot\delta_{u,i}(\boldsymbol{g})\pi(\boldsymbol{g})d\boldsymbol{g}\right] \\
&= \frac{1}{|\mathcal{D}|}\mathrm{Var}\left[\int\frac{\mathbb{I}(o_{u,i}=1)}{p_{u,i}(\boldsymbol{g})}\cdot\frac{1}{h}K\left(\frac{\boldsymbol{g}_{u,i}-\boldsymbol{g}}{h}\right)\cdot\delta_{u,i}(\boldsymbol{g})\pi(\boldsymbol{g})d\boldsymbol{g}\right] \\
&= \frac{1}{|\mathcal{D}|}\left[\mathbb{E}\left\{\left(\int\frac{\mathbb{I}(o_{u,i}=1)}{p_{u,i}(\boldsymbol{g})}\cdot\frac{1}{h}K\left(\frac{\boldsymbol{g}_{u,i}-\boldsymbol{g}}{h}\right)\cdot\delta_{u,i}(\boldsymbol{g})\pi(\boldsymbol{g})d\boldsymbol{g}\right)^2\right\}\right. \\
&\qquad\left. -\left\{\mathbb{E}\left(\int\frac{\mathbb{I}(o_{u,i}=1)}{p_{u,i}(\boldsymbol{g})}\cdot\frac{1}{h}K\left(\frac{\boldsymbol{g}_{u,i}-\boldsymbol{g}}{h}\right)\cdot\delta_{u,i}(\boldsymbol{g})\pi(\boldsymbol{g})d\boldsymbol{g}\right)\right\}^2\right]. \quad\text{(A.1)}
\end{aligned}
$$

According to the above result of the bias of the N-IPS estimator, we have

$$
\begin{aligned}
&\left\{\mathbb{E}\left(\int\frac{\mathbb{I}(o_{u,i}=1)}{p_{u,i}(\boldsymbol{g})}\cdot\frac{1}{h}K\left(\frac{\boldsymbol{g}_{u,i}-\boldsymbol{g}}{h}\right)\cdot\delta_{u,i}(\boldsymbol{g})\pi(\boldsymbol{g})d\boldsymbol{g}\right)\right\}^2 \\
&= \left\{\mathcal{L}_{\mathrm{ideal}}^{\mathrm{N}}(\hat{\mathbf{R}})+\frac{1}{2}\mu_2\int\mathbb{E}\left[\frac{\partial^2 p(o_{u,i}=1,\boldsymbol{g}|x_{u,i})}{\partial\boldsymbol{g}^2}\cdot\delta_{u,i}(\boldsymbol{g})\right]\pi(\boldsymbol{g})d\boldsymbol{g}\cdot h^2+o(h^2)\right\}^2 \\
&= [\mathcal{L}_{\mathrm{ideal}}^{\mathrm{N}}(\hat{\mathbf{R}})]^2+O(h^2). \quad\text{(A.2)}
\end{aligned}
$$

Then, we focus on analyzing the following term

$$
\mathbb{E}\left\{\left(\int\frac{\mathbb{I}(o_{u,i}=1)}{p_{u,i}(\boldsymbol{g})}\cdot\frac{1}{h}K\left(\frac{\boldsymbol{g}_{u,i}-\boldsymbol{g}}{h}\right)\cdot\delta_{u,i}(\boldsymbol{g})\pi(\boldsymbol{g})d\boldsymbol{g}\right)^2\right\}.
$$

We can observe that

$$
\begin{aligned}
&\left(\int\frac{\mathbb{I}(o_{u,i}=1)}{p_{u,i}(\boldsymbol{g})}\cdot\frac{1}{h}K\left(\frac{\boldsymbol{g}_{u,i}-\boldsymbol{g}}{h}\right)\cdot\delta_{u,i}(\boldsymbol{g})\pi(\boldsymbol{g})d\boldsymbol{g}\right)^2 \\
&= \left(\int\frac{\mathbb{I}(o_{u,i}=1)}{p_{u,i}(\boldsymbol{g})}\cdot\frac{1}{h}K\left(\frac{\boldsymbol{g}_{u,i}-\boldsymbol{g}}{h}\right)\cdot\delta_{u,i}(\boldsymbol{g})\pi(\boldsymbol{g})d\boldsymbol{g}\right) \\
&\quad\cdot\left(\int\frac{\mathbb{I}(o_{u,i}=1)}{p_{u,i}(\boldsymbol{g'})}\cdot\frac{1}{h}K\left(\frac{\boldsymbol{g}_{u,i}-\boldsymbol{g'}}{h}\right)\cdot\delta_{u,i}(\boldsymbol{g'})\pi(\boldsymbol{g'})d\boldsymbol{g'}\right) \\
&= \int\int\frac{\mathbb{I}(o_{u,i}=1)}{p_{u,i}(\boldsymbol{g})p_{u,i}(\boldsymbol{g'})}\cdot\frac{1}{h^2}K\left(\frac{\boldsymbol{g}_{u,i}-\boldsymbol{g}}{h}\right)K\left(\frac{\boldsymbol{g}_{u,i}-\boldsymbol{g'}}{h}\right)\cdot\delta_{u,i}(\boldsymbol{g})\delta_{u,i}(\boldsymbol{g'})\pi(\boldsymbol{g})\pi(\boldsymbol{g'})d\boldsymbol{g}d\boldsymbol{g'},
\end{aligned}
$$

then, we swap the order of integration and expectation, which leads to that

$$
\begin{aligned}
&\mathbb{E}\left\{\left(\int\frac{\mathbb{I}(o_{u,i}=1)}{p_{u,i}(\boldsymbol{g})}\cdot\frac{1}{h}K\left(\frac{\boldsymbol{g}_{u,i}-\boldsymbol{g}}{h}\right)\cdot\delta_{u,i}(\boldsymbol{g})\pi(\boldsymbol{g})d\boldsymbol{g}\right)^2\right\} \\
&= \int\int\mathbb{E}\left[\frac{\mathbb{I}(o_{u,i}=1)}{p_{u,i}(\boldsymbol{g})p_{u,i}(\boldsymbol{g'})}\cdot\frac{1}{h^2}K\left(\frac{\boldsymbol{g}_{u,i}-\boldsymbol{g}}{h}\right)K\left(\frac{\boldsymbol{g}_{u,i}-\boldsymbol{g'}}{h}\right)\cdot\delta_{u,i}(\boldsymbol{g})\delta_{u,i}(\boldsymbol{g'})\right]\pi(\boldsymbol{g})\pi(\boldsymbol{g'})d\boldsymbol{g}d\boldsymbol{g'}.
\end{aligned}
$$

Let $\boldsymbol{g}_{u,i} = \boldsymbol{g} + ht$, then $\boldsymbol{g}_{u,i} - \boldsymbol{g}' = (\boldsymbol{g} - \boldsymbol{g}') + ht$. We have

$$\mathbb{E}\left[\frac{\mathbb{I}(o_{u,i}=1)}{p_{u,i}(\boldsymbol{g})p_{u,i}(\boldsymbol{g}')} \cdot \frac{1}{h^2} K\left(\frac{\boldsymbol{g}_{u,i}-\boldsymbol{g}}{h}\right) K\left(\frac{\boldsymbol{g}_{u,i}-\boldsymbol{g}'}{h}\right) \cdot \delta_{u,i}(\boldsymbol{g})\delta_{u,i}(\boldsymbol{g}')\right]$$

$$= \mathbb{E}\left[\frac{1}{p_{u,i}(\boldsymbol{g})p_{u,i}(\boldsymbol{g}')} \cdot \mathbb{E}\left\{\mathbb{I}(o_{u,i}=1)\frac{1}{h^2} K\left(\frac{\boldsymbol{g}_{u,i}-\boldsymbol{g}}{h}\right) K\left(\frac{\boldsymbol{g}_{u,i}-\boldsymbol{g}'}{h}\right)\Big|x_{u,i}\right\} \cdot \mathbb{E}\left\{\delta_{u,i}(\boldsymbol{g})\delta_{u,i}(\boldsymbol{g}')\Big|x_{u,i}\right\}\right]$$

$$= \mathbb{E}\left[\frac{1}{p_{u,i}(\boldsymbol{g})p_{u,i}(\boldsymbol{g}')} \cdot \int\left\{\frac{1}{h}K\left(t\right) K\left(\frac{\boldsymbol{g}-\boldsymbol{g}'}{h}+t\right) p(o_{u,i}=1,\boldsymbol{g}+ht|x_{u,i})\right\} dt \cdot \mathbb{E}\left\{\delta_{u,i}(\boldsymbol{g})\delta_{u,i}(\boldsymbol{g}')\Big|x_{u,i}\right\}\right]$$

$$= \mathbb{E}\left[\frac{1}{p_{u,i}(\boldsymbol{g})p_{u,i}(\boldsymbol{g}')} \cdot \int\left\{\frac{1}{h}K\left(t\right) K\left(\frac{\boldsymbol{g}-\boldsymbol{g}'}{h}+t\right) p(o_{u,i}=1,\boldsymbol{g}|x_{u,i})+O(h)t\right\} dt \cdot \mathbb{E}\left\{\delta_{u,i}(\boldsymbol{g})\delta_{u,i}(\boldsymbol{g}')\Big|x_{u,i}\right\}\right]$$

$$= \mathbb{E}\left[\frac{1}{p_{u,i}(\boldsymbol{g}')} \cdot \int\frac{1}{h}K\left(t\right) K\left(\frac{\boldsymbol{g}-\boldsymbol{g}'}{h}+t\right) dt \cdot \mathbb{E}\left\{\delta_{u,i}(\boldsymbol{g})\delta_{u,i}(\boldsymbol{g}')\Big|x_{u,i}\right\}\right] \cdot \{1+O(h)\}$$

$$= \mathbb{E}\left[\frac{1}{p_{u,i}(\boldsymbol{g}')} \cdot \int\frac{1}{h}K\left(t\right) K\left(\frac{\boldsymbol{g}-\boldsymbol{g}'}{h}+t\right) dt \cdot \delta_{u,i}(\boldsymbol{g})\delta_{u,i}(\boldsymbol{g}')\right] \cdot \{1+O(h)\}.$$

Denote $\int K\left(t\right) K\left(\frac{\boldsymbol{g}-\boldsymbol{g}'}{h}+t\right) dt = \bar{K}(\frac{\boldsymbol{g}-\boldsymbol{g}'}{h})$, then

$$\mathbb{E}\left\{\left(\int\frac{\mathbb{I}(o_{u,i}=1)}{p_{u,i}(\boldsymbol{g})} \cdot \frac{1}{h}K\left(\frac{\boldsymbol{g}_{u,i}-\boldsymbol{g}}{h}\right) \cdot \delta_{u,i}(\boldsymbol{g})\pi(\boldsymbol{g})d\boldsymbol{g}\right)^2\right\}$$

$$= \int\int\mathbb{E}\left[\frac{1}{p_{u,i}(\boldsymbol{g}')} \cdot \frac{1}{h}\bar{K}(\frac{\boldsymbol{g}-\boldsymbol{g}'}{h}) \cdot \delta_{u,i}(\boldsymbol{g})\delta_{u,i}(\boldsymbol{g}')\right] \cdot \{1+O(h)\}.\pi(\boldsymbol{g})\pi(\boldsymbol{g}')d\boldsymbol{g}d\boldsymbol{g}'$$

$$= \int\mathbb{E}\left[\int\frac{1}{p_{u,i}(\boldsymbol{g}')} \cdot \frac{1}{h}\bar{K}(\frac{\boldsymbol{g}-\boldsymbol{g}'}{h}) \cdot \delta_{u,i}(\boldsymbol{g})\delta_{u,i}(\boldsymbol{g}')\pi(\boldsymbol{g}')d\boldsymbol{g}'\right] \cdot \{1+O(h)\}.\pi(\boldsymbol{g})d\boldsymbol{g}$$

$$\triangleq \int\varphi(\boldsymbol{g})\pi(\boldsymbol{g})d\boldsymbol{g}\frac{1}{h} + O(1), \tag{A.3}$$

where

$$\varphi(\boldsymbol{g}) = \int\frac{1}{p_{u,i}(\boldsymbol{g}')} \cdot \bar{K}(\frac{\boldsymbol{g}-\boldsymbol{g}'}{h}) \cdot \delta_{u,i}(\boldsymbol{g})\delta_{u,i}(\boldsymbol{g}')\pi(\boldsymbol{g}')d\boldsymbol{g}'$$

is a bounded function of $\boldsymbol{g}$.

Combing equations (A.1), (A.2), and (A.3) gives that

$$\text{Var}(\mathcal{L}_{\text{IPS}}^{\text{N}}(\hat{\mathbf{R}})) = \frac{1}{|\mathcal{D}|}\left[\int\varphi(\boldsymbol{g})\pi(\boldsymbol{g})d\boldsymbol{g}\frac{1}{h} + O(1) - [\mathcal{L}_{\text{ideal}}^{\text{N}}(\hat{\mathbf{R}})]^2 + O(h^2)\right]$$

$$= \frac{1}{|\mathcal{D}|h}\int\varphi(\boldsymbol{g})\pi(\boldsymbol{g})d\boldsymbol{g} + o(\frac{1}{|\mathcal{D}|h}).$$

Similarly, the variance of the N-DR estimator is given by

$$\text{Var}(\mathcal{L}_{\text{DR}}^{\text{N}}(\hat{\mathbf{R}})) = \frac{1}{|\mathcal{D}|h}\int\psi(\boldsymbol{g})\pi(\boldsymbol{g})d\boldsymbol{g} + o(\frac{1}{|\mathcal{D}|h}),$$

where

$$\psi(\boldsymbol{g}) = \int\frac{1}{p_{u,i}(\boldsymbol{g}')} \cdot \bar{K}(\frac{\boldsymbol{g}-\boldsymbol{g}'}{h}) \cdot \{\delta_{u,i}(\boldsymbol{g})-\hat{\delta}_{u,i}(\boldsymbol{g})\}\{\delta_{u,i}(\boldsymbol{g}')-\hat{\delta}_{u,i}(\boldsymbol{g}')\}\pi(\boldsymbol{g}')d\boldsymbol{g}'$$

is a bounded function of $\boldsymbol{g}$.

$\square$

### B.3 PROOF OF THEOREM 4

**Theorem 4** (Optimal Bandwidth of N-IPS and N-DR). *Under Assumptions 1–4,*

*(a) the optimal bandwidth for the N-IPS estimator in terms of the asymptotic mean-squared error is*

$$h^*_{\text{N-IPS}} = \left[ \frac{\int \varphi(\boldsymbol{g})\pi(\boldsymbol{g})d\boldsymbol{g}}{4|\mathcal{D}| \left( \frac{1}{2}\mu_2 \int \mathbb{E}\left[ \frac{\partial^2 p(o_{u,i}=1,\boldsymbol{g}_{u,i}=\boldsymbol{g}|x_{u,i})}{\partial \boldsymbol{g}^2} \cdot \delta_{u,i}(\boldsymbol{g}) \right] \pi(\boldsymbol{g})d\boldsymbol{g} \right)^2} \right]^{1/5},$$

*where $\varphi(\boldsymbol{g})$ is defined in Theorem 3;*

*(b) the optimal bandwidth for the N-DR estimator in terms of the asymptotic mean-squared error is*

$$h^*_{\text{N-DR}} = \left[ \frac{\int \psi(\boldsymbol{g})\pi(\boldsymbol{g})d\boldsymbol{g}}{4|\mathcal{D}| \left( \frac{1}{2}\mu_2 \int \mathbb{E}\left[ \frac{\partial^2 p(o_{u,i}=1,\boldsymbol{g}|x_{u,i})}{\partial \boldsymbol{g}^2} \cdot \{\delta_{u,i}(\boldsymbol{g}) - \hat{\delta}_{u,i}(\boldsymbol{g})\} \right] \pi(\boldsymbol{g})d\boldsymbol{g} \right)^2} \right]^{1/5}.$$

*Proof of Theorem 4.* Recall that

$$\text{Bias}(\mathcal{L}^{\text{N}}_{\text{IPS}}(\hat{\mathbf{R}})) = \frac{1}{2}\mu_2 \int \mathbb{E}\left[ \frac{\partial^2 p(o_{u,i}=1,\boldsymbol{g}|x_{u,i})}{\partial \boldsymbol{g}^2} \cdot \delta_{u,i}(\boldsymbol{g}) \right]\pi(\boldsymbol{g})d\boldsymbol{g} \cdot h^2 + o(h^2),$$

$$\text{Var}(\mathcal{L}^{\text{N}}_{\text{IPS}}(\hat{\mathbf{R}})) = \frac{1}{|\mathcal{D}|h} \int \varphi(\boldsymbol{g})\pi(\boldsymbol{g})d\boldsymbol{g} + o(\frac{1}{|\mathcal{D}|h}).$$

The mean-squared error of the N-IPS estimator is given as

$$\mathbb{E}\left[ \left( \mathcal{L}^{\text{N}}_{\text{IPS}}(\hat{\mathbf{R}}) - \mathcal{L}^{\text{N}}_{\text{ideal}}(\hat{\mathbf{R}}) \right)^2 \right]$$

$$= (\text{Bias}(\mathcal{L}^{\text{N}}_{\text{IPS}}(\hat{\mathbf{R}})))^2 + \text{Var}(\mathcal{L}^{\text{N}}_{\text{IPS}}(\hat{\mathbf{R}}))$$

$$= \left( \frac{1}{2}\mu_2 \int \mathbb{E}\left[ \frac{\partial^2 p(o_{u,i}=1,\boldsymbol{g}|x_{u,i})}{\partial \boldsymbol{g}^2} \cdot \delta_{u,i}(\boldsymbol{g}) \right]\pi(\boldsymbol{g})d\boldsymbol{g} \right)^2 \cdot h^4 + o(h^4)$$

$$+ \frac{1}{|\mathcal{D}|h} \int \varphi(\boldsymbol{g})\pi(\boldsymbol{g})d\boldsymbol{g} + o(\frac{1}{|\mathcal{D}|h}).$$

Minimizing the leading terms of the above mean-squared error with respect to $h$ leads to that

$$h^*_{\text{N-IPS}} = \left[ \frac{\int \varphi(\boldsymbol{g})\pi(\boldsymbol{g})d\boldsymbol{g}}{4|\mathcal{D}| \left( \frac{1}{2}\mu_2 \int \mathbb{E}\left[ \frac{\partial^2 p(o_{u,i}=1,\boldsymbol{g}_{u,i}=\boldsymbol{g}|x_{u,i})}{\partial \boldsymbol{g}^2} \cdot \delta_{u,i}(\boldsymbol{g}) \right] \pi(\boldsymbol{g})d\boldsymbol{g} \right)^2} \right]^{1/5} = O(|\mathcal{D}|^{-1/5}).$$

Similarly, the optimal bandwidth for the N-DR estimator in terms of the asymptotic mean-squared error can be obtained.

$\square$

## B.4 PROOF OF THEOREM 5

**Lemma 1** (McDiarmid's Inequality). *Let $X_1, ..., X_m \in \mathcal{X}^m$ be a set of $m \geq 1$ independent random variables and assume that there exist $c_1, ..., c_m > 0$ such that $f : \mathcal{X}^m \to \mathbb{R}$ satisfies the following conditions:*

$$|f(x_1, ..., x_i, ..., x_m) - f(x_1, ..., x'_i, ..., x_m)| \leq c_i,$$

*for all $i \in \{1, 2, ..., m\}$ and any points $x_1, ..., x_m, x'_i \in \mathcal{X}$. Let $f(S)$ denote $f(X_1, ..., X_m)$, then for all $\epsilon > 0$, the following inequalities hold:*

$$\mathbb{P}[f(S) - \mathbb{E}\{f(S)\} \geq \epsilon] \leq \exp\left( -\frac{2\epsilon^2}{\sum_{i=1}^m c_i^2} \right)$$

$$\mathbb{P}[f(S) - \mathbb{E}\{f(S)\} \leq -\epsilon] \leq \exp\left( -\frac{2\epsilon^2}{\sum_{i=1}^m c_i^2} \right).$$

*Proof of Lemma 1.* The proof can be found in Appendix D.3 of Mohri et al. (2018). $\square$

**Lemma 2.** *Under the conditions in Lemma 1, we have with probability at least $1 - \eta$,*

$$|f(S) - \mathbb{E}[f(S)]| \leq \sqrt{\frac{\sum_{i=1}^{m} c_i^2}{2} \log(\frac{2}{\eta})}.$$

*In particular, if $c_i \leq c$ for all $i \in \{1, 2, ..., m\}$,*

$$|f(S) - \mathbb{E}[f(S)]| \leq c\sqrt{\frac{m}{2} \log(\frac{2}{\eta})}.$$

*Proof of Lemma 2.* This conclusion follows by letting $\eta = 2\exp\left(-\frac{2\epsilon^2}{\sum_{i=1}^{m} c_i^2}\right)$ in Lemma 1. $\square$

**Lemma 3** (Rademacher Comparison Lemma). *Let $X \in \mathcal{X}$ be a random variable with distribution $\mathbb{P}$, $X_1, ..., X_m$ be a set of independent copies of $X$, $\mathcal{F}$ be a class of real-valued functions on $\mathcal{X}$. Then we have*

$$\mathbb{E}\sup_{f \in \mathcal{G}}\left|\frac{1}{m}\sum_{i=1}^{m} f(X_i) - \mathbb{E}(f(X_i))\right| \leq 2\mathbb{E}\left[\mathbb{E}_{\boldsymbol{\sigma} \sim \{-1,+1\}^m}\sup_{f \in \mathcal{G}}\frac{1}{m}\sum_{i=1}^{m} f(X_i)\sigma_i\right],$$

*where $\boldsymbol{\sigma} = (\sigma_1, ..., \sigma_m)$ is a Rademacher sequence.*

*Proof of Lemma 3.* See Lemma 26.2 of Shalev-Shwartz and Ben-David (2014). $\square$

Recall that $\mathcal{F}$ is the hypothesis space of prediction matrices $\hat{\mathbf{R}}$ (or prediction model $f_\theta$), the Rademacher complexity is

$$\mathcal{R}(\mathcal{F}) = \mathbb{E}_{\boldsymbol{\sigma} \sim \{-1,+1\}^{|\mathcal{D}|}}\sup_{f_\theta \in \mathcal{F}}\left[\frac{1}{|\mathcal{D}|}\sum_{(u,i) \in \mathcal{D}}\sigma_{u,i}\delta_{u,i}(\boldsymbol{g})\right].$$

**Lemma 4** (Uniform Tail Bound of N-IPS and N-DR). *Under Assumptions 1–5 and suppose that $K(t) \leq M_K$, then we have with probability at least $1 - \eta$,*

*(a)*

$$\sup_{\hat{\mathbf{R}} \in \mathcal{F}}\left|\mathcal{L}_{\text{IPS}}^{\text{N}}(\hat{\mathbf{R}}) - \mathbb{E}[\mathcal{L}_{\text{IPS}}^{\text{N}}(\hat{\mathbf{R}})]\right| \leq \frac{2M_p M_K}{h}\mathcal{R}(\mathcal{F}) + \frac{5}{2}\frac{M_p M_K M_\delta}{h}\sqrt{\frac{2}{|\mathcal{D}|}\log(\frac{4}{\eta})};$$

*(b)*

$$\sup_{\hat{\mathbf{R}} \in \mathcal{F}}\left|\mathcal{L}_{\text{DR}}^{\text{N}}(\hat{\mathbf{R}}) - \mathbb{E}[\mathcal{L}_{\text{DR}}^{\text{N}}(\hat{\mathbf{R}})]\right| \leq \frac{2M_p M_K}{h}\mathcal{R}(\mathcal{F}) + \frac{5}{2}\frac{M_p M_K M_{|\delta-\hat{\delta}|}}{h}\sqrt{\frac{2}{|\mathcal{D}|}\log(\frac{4}{\eta})}.$$

*Proof of Lemma 4.* We first discuss the uniform tail bound of $\mathcal{L}_{\text{IPS}}^{\text{N}}(\hat{\mathbf{R}})$, that is, we want to show the upper bound of $\sup_{\hat{\mathbf{R}} \in \mathcal{F}}\left|\mathcal{L}_{\text{IPS}}^{\text{N}}(\hat{\mathbf{R}}) - \mathbb{E}\{\mathcal{L}_{\text{IPS}}^{\text{N}}(\hat{\mathbf{R}})\}\right|$.

Note that

$$\sup_{\hat{\mathbf{R}} \in \mathcal{F}}\left|\mathcal{L}_{\text{IPS}}^{\text{N}}(\hat{\mathbf{R}}) - \mathbb{E}\{\mathcal{L}_{\text{IPS}}^{\text{N}}(\hat{\mathbf{R}})\}\right|$$

$$= \sup_{\hat{\mathbf{R}} \in \mathcal{F}}\left|\int \pi(\boldsymbol{g})\left[\frac{1}{|\mathcal{D}|}\sum_{(u,i) \in \mathcal{D}}\left\{\frac{\mathbb{I}(o_{u,i} = 1) \cdot K\left((\boldsymbol{g}_{u,i} - \boldsymbol{g})/h\right) \cdot \delta_{u,i}(\boldsymbol{g})}{h \cdot \hat{p}_{u,i}(\boldsymbol{g})}\right\}\right.\right.$$

$$\left.\left. - \mathbb{E}\left\{\frac{\mathbb{I}(o_{u,i} = 1) \cdot K\left((\boldsymbol{g}_{u,i} - \boldsymbol{g})/h\right) \cdot \delta_{u,i}(\boldsymbol{g})}{h \cdot \hat{p}_{u,i}(\boldsymbol{g})}\right\}\right]d\boldsymbol{g}\right|$$

$$\leq \int \pi(\boldsymbol{g})\sup_{\hat{\mathbf{R}} \in \mathcal{F}}\left|\frac{1}{|\mathcal{D}|}\sum_{(u,i) \in \mathcal{D}}\left\{\frac{\mathbb{I}(o_{u,i} = 1) \cdot K\left((\boldsymbol{g}_{u,i} - \boldsymbol{g})/h\right) \cdot \delta_{u,i}(\boldsymbol{g})}{h \cdot \hat{p}_{u,i}(\boldsymbol{g})}\right\}\right.$$

$$\left. - \mathbb{E}\left\{\frac{\mathbb{I}(o_{u,i} = 1) \cdot K\left((\boldsymbol{g}_{u,i} - \boldsymbol{g})/h\right) \cdot \delta_{u,i}(\boldsymbol{g})}{h \cdot \hat{p}_{u,i}(\boldsymbol{g})}\right\}\right|d\boldsymbol{g}.$$

For all prediction model $\hat{\mathbf{R}} \in \mathcal{F}$ and $\boldsymbol{g}$, we have

$$\frac{1}{|\mathcal{D}|} \left| \frac{\mathbb{I}(o_{u,i} = 1) \cdot K\left((\boldsymbol{g}_{u,i} - \boldsymbol{g})/h\right) \cdot \delta_{u,i}(\boldsymbol{g})}{h \cdot \hat{p}_{u,i}(\boldsymbol{g})} - \frac{\mathbb{I}(o_{u',i'} = 1) \cdot K\left((\boldsymbol{g}_{u',i'} - \boldsymbol{g})/h\right) \cdot \delta_{u',i'}(\boldsymbol{g})}{h \cdot \hat{p}_{u',i'}(\boldsymbol{g})} \right| \leq \frac{M_p M_\delta M_K}{h|\mathcal{D}|},$$

then applying Lemma 2 yields that with probability at least $1 - \eta/2$

$$\sup_{\hat{\mathbf{R}} \in \mathcal{F}} \left| \mathbb{E}\left\{ \frac{\mathbb{I}(o_{u,i} = 1) \cdot K\left((\boldsymbol{g}_{u,i} - \boldsymbol{g})/h\right) \cdot \delta_{u,i}(\boldsymbol{g})}{h \cdot \hat{p}_{u,i}(\boldsymbol{g})} \right\} \right.$$

$$\left. - \frac{1}{|\mathcal{D}|} \sum_{(u,i) \in \mathcal{D}} \left\{ \frac{\mathbb{I}(o_{u,i} = 1) \cdot K\left((\boldsymbol{g}_{u,i} - \boldsymbol{g})/h\right) \cdot \delta_{u,i}(\boldsymbol{g})}{h \cdot \hat{p}_{u,i}(\boldsymbol{g})} \right\} \right|$$

$$\leq \mathbb{E}\left[ \sup_{\hat{\mathbf{R}} \in \mathcal{F}} \left| \mathbb{E}\left\{ \frac{\mathbb{I}(o_{u,i} = 1) \cdot K\left((\boldsymbol{g}_{u,i} - \boldsymbol{g})/h\right) \cdot \delta_{u,i}(\boldsymbol{g})}{h \cdot \hat{p}_{u,i}(\boldsymbol{g})} \right\} \right. \right.$$

$$\left. \left. - \frac{1}{|\mathcal{D}|} \sum_{(u,i) \in \mathcal{D}} \left\{ \frac{\mathbb{I}(o_{u,i} = 1) \cdot K\left((\boldsymbol{g}_{u,i} - \boldsymbol{g})/h\right) \cdot \delta_{u,i}(\boldsymbol{g})}{h \cdot \hat{p}_{u,i}(\boldsymbol{g})} \right\} \right| \right] + \frac{M_p M_\delta M_K}{2h} \sqrt{\frac{2}{|\mathcal{D}|} \log(\frac{4}{\eta})}$$

$$\leq \frac{M_p M_K}{h} 2\mathbb{E}[\mathcal{R}(\mathcal{F})] + \frac{M_p M_\delta M_K}{2h} \sqrt{\frac{2}{|\mathcal{D}|} \log(\frac{4}{\eta})}, \tag{A.4}$$

where the last inequality holds by Lemma 3 and

$$\frac{\mathbb{I}(o_{u,i} = 1) \cdot K\left((\boldsymbol{g}_{u,i} - \boldsymbol{g})/h\right)}{h \cdot \hat{p}_{u,i}(\boldsymbol{g})} \leq \frac{M_p M_K}{h}.$$

Recall that

$$\mathcal{R}(\mathcal{F}) = \mathbb{E}_{\boldsymbol{\sigma} \sim \{-1, +1\}^{|\mathcal{D}|}} \sup_{\hat{\mathbf{R}} \in \mathcal{F}} \left[ \frac{1}{|\mathcal{D}|} \sum_{(u,i) \in \mathcal{D}} \sigma_{u,i} \delta_{u,i}(\boldsymbol{g}) \right]$$

and let $f(S) = \mathcal{R}(\mathcal{F})$ and $c = 2M_\delta/|\mathcal{D}|$ in Lemmas 1 and 2, by applying Lemma 2 again, we have with probability at least $1 - \eta/2$

$$\mathbb{E}[\mathcal{R}(\mathcal{F})] - \mathcal{R}(\mathcal{F}) \leq M_\delta \sqrt{\frac{2}{|\mathcal{D}|} \log(\frac{4}{\eta})}. \tag{A.5}$$

Combining inequalities (A.4) and (A.5) leads to that with probability at least $1 - \eta$

$$\sup_{\hat{\mathbf{R}} \in \mathcal{F}} \left| \frac{1}{|\mathcal{D}|} \sum_{(u,i) \in \mathcal{D}} \left\{ \frac{\mathbb{I}(o_{u,i} = 1) \cdot K\left((\boldsymbol{g}_{u,i} - \boldsymbol{g})/h\right) \cdot \delta_{u,i}(\boldsymbol{g})}{h \cdot \hat{p}_{u,i}(\boldsymbol{g})} \right\} \right.$$

$$\left. - \mathbb{E}\left\{ \frac{\mathbb{I}(o_{u,i} = 1) \cdot K\left((\boldsymbol{g}_{u,i} - \boldsymbol{g})/h\right) \cdot \delta_{u,i}(\boldsymbol{g})}{h \cdot \hat{p}_{u,i}(\boldsymbol{g})} \right\} \right|$$

$$\leq \frac{2M_p M_K}{h} \mathcal{R}(\mathcal{F}) + \frac{5}{2} \frac{M_p M_K M_\delta}{h} \sqrt{\frac{2}{|\mathcal{D}|} \log(\frac{4}{\eta})},$$

which implies Lemma 4(a) by noting that $\int \pi(\boldsymbol{g}) d\boldsymbol{g} = 1$.

Next, we show the uniform tail bound of $\mathcal{L}_{\mathrm{DR}}^{\mathrm{N}}(\hat{\mathbf{R}})$. Note that

$$\mathcal{L}_{\mathrm{DR}}^{\mathrm{N}}(\hat{\mathbf{R}}) = \delta_{u,i}(\boldsymbol{g}) + \frac{\mathbb{I}(o_{u,i} = 1) \cdot K\left((\boldsymbol{g}_{u,i} - \boldsymbol{g})/h\right) \cdot \{\delta_{u,i}(\boldsymbol{g}) - \hat{\delta}_{u,i}(\boldsymbol{g})\}}{h \cdot \hat{p}_{u,i}(\boldsymbol{g})}$$

and

$$\sup_{\hat{\mathbf{R}}\in\mathcal{F}} \left| \mathcal{L}_{\mathrm{DR}}^{\mathrm{N}}(\hat{\mathbf{R}}) - \mathbb{E}\{\mathcal{L}_{\mathrm{DR}}^{\mathrm{N}}(\hat{\mathbf{R}})\} \right|$$

$$= \sup_{\hat{\mathbf{R}}\in\mathcal{F}} \int \pi(\boldsymbol{g}) \left[ \frac{1}{|\mathcal{D}|} \sum_{(u,i)\in\mathcal{D}} \left\{ \frac{\mathbb{I}(o_{u,i}=1)\cdot K\left((\boldsymbol{g}_{u,i}-\boldsymbol{g})/h\right)\cdot\{\delta_{u,i}(\boldsymbol{g})-\hat{\delta}_{u,i}(\boldsymbol{g})\}}{h\cdot\hat{p}_{u,i}(\boldsymbol{g})} \right\} \right.$$

$$\left. - \mathbb{E}\left\{ \frac{\mathbb{I}(o_{u,i}=1)\cdot K\left((\boldsymbol{g}_{u,i}-\boldsymbol{g})/h\right)\cdot\{\delta_{u,i}(\boldsymbol{g})-\hat{\delta}_{u,i}(\boldsymbol{g})\}}{h\cdot\hat{p}_{u,i}(\boldsymbol{g})} \right\} \right] d\boldsymbol{g},$$

which has the same form as $\sup_{\hat{\mathbf{R}}\in\mathcal{F}} \left| \mathcal{L}_{\mathrm{IPS}}^{\mathrm{N}}(\hat{\mathbf{R}}) - \mathbb{E}\{\mathcal{L}_{\mathrm{IPS}}^{\mathrm{N}}(\hat{\mathbf{R}})\} \right|$, except that $\delta_{u,i}(\boldsymbol{g})$ is replaced by $\delta_{u,i}(\boldsymbol{g}) - \hat{\delta}_{u,i}$. By a similar argument of the proof of the N-IPS estimator, we obtain the Lemma 4(b).

$\square$

Let

$$\hat{\mathbf{R}}^{\dagger} = \arg\min_{\hat{\mathbf{R}}\in\mathcal{F}} \mathcal{L}_{\mathrm{DR}}^{\mathrm{N}}(\hat{\mathbf{R}}), \quad \hat{\mathbf{R}}^{\ddagger} = \arg\min_{\hat{\mathbf{R}}\in\mathcal{F}} \mathcal{L}_{\mathrm{IPS}}^{\mathrm{N}}(\hat{\mathbf{R}}).$$

The following Theorem 5 shows the generalization error bounds of the N-IPS and N-DR estimators.

**Theorem 5** (Generalization Error Bounds of N-IPS and N-DR). *Under Assumptions 1–5 and suppose that $K(t) \leq M_K$, we have with probability at least $1 - \eta$,*

*(a)*

$$\mathcal{L}_{\mathrm{ideal}}^{\mathrm{N}}(\hat{\mathbf{R}}^{\dagger}) \leq \min_{\hat{\mathbf{R}}\in\mathcal{F}} \mathcal{L}_{\mathrm{ideal}}^{\mathrm{N}}(\hat{\mathbf{R}}) + \mu_2 M_{|\delta-\hat{\delta}|} \int \mathbb{E}\left[ \frac{\partial^2 p(o_{u,i}=1,\boldsymbol{g}|x_{u,i})}{\partial\boldsymbol{g}^2} \right] \pi(\boldsymbol{g}) d\boldsymbol{g} \cdot h^2 + o(h^2)$$

$$+ \frac{4M_p M_K}{h} \mathcal{R}(\mathcal{F}) + \frac{5M_p M_K M_{|\delta-\hat{\delta}|}}{h} \sqrt{\frac{2}{|\mathcal{D}|}\log(\frac{4}{\eta})};$$

*(b)*

$$\mathcal{L}_{\mathrm{ideal}}^{\mathrm{N}}(\hat{\mathbf{R}}^{\ddagger}) \leq \min_{\hat{\mathbf{R}}\in\mathcal{F}} \mathcal{L}_{\mathrm{ideal}}^{\mathrm{N}}(\hat{\mathbf{R}}) + \mu_2 M_{\delta} \int \mathbb{E}\left[ \frac{\partial^2 p(o_{u,i}=1,\boldsymbol{g}|x_{u,i})}{\partial\boldsymbol{g}^2} \right] \pi(\boldsymbol{g}) d\boldsymbol{g} \cdot h^2 + o(h^2)$$

$$+ \frac{4M_p M_K}{h} \mathcal{R}(\mathcal{F}) + \frac{5M_p M_K M_{\delta}}{h} \sqrt{\frac{2}{|\mathcal{D}|}\log(\frac{4}{\eta})}.$$

*Proof of Theorem 5.* It suffices to show Theorem 5(a), since Theorem 5(b) can be derived from a similar argument. Define

$$\hat{\mathbf{R}}^{*} = \arg\min_{\hat{\mathbf{R}}\in\mathcal{F}} \mathcal{L}_{\mathrm{ideal}}^{\mathrm{N}}(\hat{\mathbf{R}}),$$

then we have

$$\mathcal{L}_{\mathrm{ideal}}^{\mathrm{N}}(\hat{\mathbf{R}}^{\dagger}) - \min_{\hat{\mathbf{R}}\in\mathcal{F}} \mathcal{L}_{\mathrm{ideal}}^{\mathrm{N}}(\hat{\mathbf{R}})$$

$$= \mathcal{L}_{\mathrm{ideal}}^{\mathrm{N}}(\hat{\mathbf{R}}^{\dagger}) - \mathcal{L}_{\mathrm{ideal}}^{\mathrm{N}}(\hat{\mathbf{R}}^{*})$$

$$\leq \mathcal{L}_{\mathrm{ideal}}^{\mathrm{N}}(\hat{\mathbf{R}}^{\dagger}) - \mathcal{L}_{\mathrm{DR}}^{\mathrm{N}}(\hat{\mathbf{R}}^{\dagger}) + \mathcal{L}_{\mathrm{DR}}^{\mathrm{N}}(\hat{\mathbf{R}}^{\dagger}) - \mathcal{L}_{\mathrm{DR}}^{\mathrm{N}}(\hat{\mathbf{R}}^{*}) + \mathcal{L}_{\mathrm{DR}}^{\mathrm{N}}(\hat{\mathbf{R}}^{*}) - \mathcal{L}_{\mathrm{ideal}}^{\mathrm{N}}(\hat{\mathbf{R}}^{*})$$

$$\leq \mathcal{L}_{\mathrm{ideal}}^{\mathrm{N}}(\hat{\mathbf{R}}^{\dagger}) - \mathcal{L}_{\mathrm{DR}}^{\mathrm{N}}(\hat{\mathbf{R}}^{\dagger}) + \mathcal{L}_{\mathrm{DR}}^{\mathrm{N}}(\hat{\mathbf{R}}^{*}) - \mathcal{L}_{\mathrm{ideal}}^{\mathrm{N}}(\hat{\mathbf{R}}^{*})$$

$$:= A + B,$$

where

$$A = \mathcal{L}_{\mathrm{ideal}}^{\mathrm{N}}(\hat{\mathbf{R}}^{\dagger}) - \mathcal{L}_{\mathrm{DR}}^{\mathrm{N}}(\hat{\mathbf{R}}^{\dagger}),$$

$$B = \mathcal{L}_{\mathrm{DR}}^{\mathrm{N}}(\hat{\mathbf{R}}^{*}) - \mathcal{L}_{\mathrm{ideal}}^{\mathrm{N}}(\hat{\mathbf{R}}^{*}).$$

The first term can be decomposed as follows

$$
\begin{aligned}
A &= \mathcal{L}_{\text{ideal}}^{\text{N}}(\hat{\mathbf{R}}^{\dagger}) - \mathbb{E}[\mathcal{L}_{\text{DR}}^{\text{N}}(\hat{\mathbf{R}}^{\dagger})] + \mathbb{E}[\mathcal{L}_{\text{DR}}^{\text{N}}(\hat{\mathbf{R}}^{\dagger})] - \mathcal{L}_{\text{DR}}^{\text{N}}(\hat{\mathbf{R}}^{\dagger}) \\
&= \left| \text{Bias}[\mathcal{L}_{\text{DR}}^{\text{N}}(\hat{\mathbf{R}}^{\dagger})] \right| + \mathbb{E}[\mathcal{L}_{\text{DR}}^{\text{N}}(\hat{\mathbf{R}}^{\dagger})] - \mathcal{L}_{\text{DR}}^{\text{N}}(\hat{\mathbf{R}}^{\dagger}) \\
&\leq \left| \text{Bias}[\mathcal{L}_{\text{DR}}^{\text{N}}(\hat{\mathbf{R}}^{\dagger})] \right| + \sup_{\hat{\mathbf{R}} \in \mathcal{F}} \left[ \mathbb{E}\{\mathcal{L}_{\text{IPS}}^{\text{N}}(\hat{\mathbf{R}})\} - \mathcal{L}_{\text{IPS}}^{\text{N}}(\hat{\mathbf{R}}) \right] \\
&= \frac{1}{2}\mu_2 \left| \int \mathbb{E}\left[ \frac{\partial^2 p(o_{u,i} = 1, \boldsymbol{g}|x_{u,i})}{\partial \boldsymbol{g}^2} \cdot \{\delta_{u,i}(\boldsymbol{g}) - \hat{\delta}_{u,i}(\boldsymbol{g})\} \right] \pi(\boldsymbol{g}) d\boldsymbol{g} \right| \cdot h^2 + o(h^2) \\
&\quad + \sup_{\hat{\mathbf{R}} \in \mathcal{F}} \left[ \mathbb{E}\{\mathcal{L}_{\text{IPS}}^{\text{N}}(\hat{\mathbf{R}})\} - \mathcal{L}_{\text{IPS}}^{\text{N}}(\hat{\mathbf{R}}) \right] \\
&\leq \frac{1}{2}\mu_2 M_{|\delta - \hat{\delta}|} \left| \int \mathbb{E}\left[ \frac{\partial^2 p(o_{u,i} = 1, \boldsymbol{g}|x_{u,i})}{\partial \boldsymbol{g}^2} \right] \pi(\boldsymbol{g}) d\boldsymbol{g} \right| \cdot h^2 + o(h^2) \\
&\quad + \sup_{\hat{\mathbf{R}} \in \mathcal{F}} \left[ \mathbb{E}\{\mathcal{L}_{\text{IPS}}^{\text{N}}(\hat{\mathbf{R}})\} - \mathcal{L}_{\text{IPS}}^{\text{N}}(\hat{\mathbf{R}}) \right],
\end{aligned}
$$

the upper bound does not depend on $\hat{\mathbf{R}}^{\dagger}$. Likewise, the upper bound of the second term is

$$
\begin{aligned}
B &\leq \frac{1}{2}\mu_2 M_{|\delta - \hat{\delta}|} \left| \int \mathbb{E}\left[ \frac{\partial^2 p(o_{u,i} = 1, \boldsymbol{g}|x_{u,i})}{\partial \boldsymbol{g}^2} \right] \pi(\boldsymbol{g}) d\boldsymbol{g} \right| \cdot h^2 + o(h^2) \\
&\quad + \sup_{\hat{\mathbf{R}} \in \mathcal{F}} \left[ \mathbb{E}\{\mathcal{L}_{\text{IPS}}^{\text{N}}(\hat{\mathbf{R}})\} - \mathcal{L}_{\text{IPS}}^{\text{N}}(\hat{\mathbf{R}}) \right].
\end{aligned}
$$

Then by Lemma 4, we have with probability at least $1 - \eta$,

$$
\begin{aligned}
A + B &\leq \mu_2 M_{|\delta - \hat{\delta}|} \left| \int \mathbb{E}\left[ \frac{\partial^2 p(o_{u,i} = 1, \boldsymbol{g}|x_{u,i})}{\partial \boldsymbol{g}^2} \right] \pi(\boldsymbol{g}) d\boldsymbol{g} \right| \cdot h^2 + o(h^2) \\
&\quad + 2 \sup_{\hat{\mathbf{R}} \in \mathcal{F}} \left[ \mathbb{E}\{\mathcal{L}_{\text{IPS}}^{\text{N}}(\hat{\mathbf{R}})\} - \mathcal{L}_{\text{IPS}}^{\text{N}}(\hat{\mathbf{R}}) \right] \\
&\leq \mu_2 M_{|\delta - \hat{\delta}|} \left| \int \mathbb{E}\left[ \frac{\partial^2 p(o_{u,i} = 1, \boldsymbol{g}|x_{u,i})}{\partial \boldsymbol{g}^2} \right] \pi(\boldsymbol{g}) d\boldsymbol{g} \right| \cdot h^2 + o(h^2) \\
&\quad + \frac{4 M_p M_K}{h} \mathcal{R}(\mathcal{F}) + 5 \frac{M_p M_K M_{|\delta - \hat{\delta}|}}{h} \sqrt{\frac{2}{|\mathcal{D}|} \log(\frac{4}{\eta})},
\end{aligned}
$$

which implies the conclusion of Theorem 5(a).

$\square$

## C EXTENSION: MULTI-DIMENSIONAL TREATMENT REPRESENTATION

For ease of presentation, we focus on the case of univariate $\boldsymbol{g}$ in the manuscript. In this section, we extend the univariate case and consider the case of multi-dimensional $\boldsymbol{g}$.

Suppose that $\boldsymbol{g}$ is a $q$-dimensional vector denoted as $\boldsymbol{g} = (g_1, ..., g_q)$. In this case, the bandwidth is $\boldsymbol{h} = (h_1, ..., h_q)$ and the kernel function is $\boldsymbol{K}((\boldsymbol{g}_{u,i} - \boldsymbol{g})/\boldsymbol{h}) = \prod_{s=1}^{q} K((g_{u,i}^s - g_s)/h_s)$, where $\boldsymbol{g}_{u,i} = (g_{u,i}^1, ..., g_{u,i}^q)$ with $g_{u,i}^s$ being its $s$-th element, $K(\cdot)$ is the univariate kernel function such as Epanechnikov kernel $K(t) = \frac{3}{4}(1 - t^2)\mathbb{I}\{|t| \leq 1\}$ and Gaussian kernel $K(t) = \frac{1}{\sqrt{2\pi}} \cdot \exp\{-\frac{t^2}{2}\}$ for $t \in \mathbb{R}$. The N-IPS and N-DR estimators are given as

$$
\mathcal{L}_{\text{IPS}}^{\text{N}}(\hat{\mathbf{R}}) = \int \mathcal{L}_{\text{IPS}}^{\text{N}}(\hat{\mathbf{R}}|\boldsymbol{g})\pi(\boldsymbol{g})d\boldsymbol{g},
$$

$$
\mathcal{L}_{\text{DR}}^{\text{N}}(\hat{\mathbf{R}}) = \int \mathcal{L}_{\text{DR}}^{\text{N}}(\hat{\mathbf{R}}|\boldsymbol{g})\pi(\boldsymbol{g})d\boldsymbol{g},
$$

where

$$\mathcal{L}_{\mathrm{IPS}}^{\mathrm{N}}(\hat{\mathbf{R}}|\boldsymbol{g}) = \frac{1}{|\mathcal{D}|} \sum_{(u,i)\in\mathcal{D}} \frac{\mathbb{I}(o_{u,i}=1)\cdot \boldsymbol{K}\left((\boldsymbol{g}_{u,i}-\boldsymbol{g})/\boldsymbol{h}\right)\cdot \delta_{u,i}(\boldsymbol{g})}{\prod_{s=1}^{q} h_s \cdot p_{u,i}(\boldsymbol{g})},$$

$$\mathcal{L}_{\mathrm{DR}}^{\mathrm{N}}(\hat{\mathbf{R}}|\boldsymbol{g}) = \frac{1}{|\mathcal{D}|} \sum_{(u,i)\in\mathcal{D}} \left[\hat{\delta}_{u,i}(\boldsymbol{g}) + \frac{\mathbb{I}(o_{u,i}=1)\cdot \boldsymbol{K}\left((\boldsymbol{g}_{u,i}-\boldsymbol{g})/\boldsymbol{h}\right)\cdot \{\delta_{u,i}(\boldsymbol{g})-\hat{\delta}_{u,i}(\boldsymbol{g})\}}{\prod_{s=1}^{q} h_s \cdot p_{u,i}(\boldsymbol{g})}\right].$$

We show the theoretical properties of the proposed N-IPS and N-DR estimators, extending the results of Theorems 3–5. First, we present the bias of the N-IPS and N-DR estimators in the setting of multi-dimension treatment representation.

**Assumption 6** (Regularity Conditions for Kernel Smoothing). *(a) For $s = 1,...,q$, $h_s \to 0$ as $|\mathcal{D}| \to \infty$; (b) $|\mathcal{D}| \prod_{s=1}^{q} h_s \to \infty$ as $|\mathcal{D}| \to \infty$; (c) $p(o_{u,i} = 1, \boldsymbol{g}_{u,i} = \boldsymbol{g} \mid x_{u,i})$ is twice differentiable with respect to $\boldsymbol{g} = (g_1,...,g_q)$.*

**Theorem 6** (Bias and Variance of N-IPS and N-DR). *Under Assumptions 1–3 and 6,*

*(a) the bias of the N-IPS estimator is given as*

$$\frac{1}{2}\mu_2 \left[\sum_{s=1}^{q} h_s^2 \int \mathbb{E}\left\{\frac{\partial^2 p(o_{u,i}=1,\boldsymbol{g}_{u,i}=\boldsymbol{g}|x_{u,i})}{\partial g_s^2}\cdot \delta_{u,i}(\boldsymbol{g})\right\}\pi(\boldsymbol{g})d\boldsymbol{g}\right] + o(\sum_{s=1}^{q} h_s^2),$$

*where $\mu_2 = \int K(t)t^2 dt$. The bias of the N-DR estimator is given as*

$$\frac{1}{2}\mu_2 \left[\sum_{s=1}^{q} h_s^2 \int \mathbb{E}\left\{\frac{\partial^2 p(o_{u,i}=1,\boldsymbol{g}_{u,i}=\boldsymbol{g}|x_{u,i})}{\partial g_s^2}\cdot \left(\delta_{u,i}(\boldsymbol{g})-\hat{\delta}_{u,i}(\boldsymbol{g})\right)\right\}\pi(\boldsymbol{g})d\boldsymbol{g}\right] + o(\sum_{s=1}^{q} h_s^2);$$

*(b) the variance of the N-IPS estimator is*

$$Var(\mathcal{L}_{\mathrm{IPS}}^{\mathrm{N}}(\hat{\mathbf{R}})) = \frac{1}{|\mathcal{D}|\prod_{s=1}^{q} h_s} \int \varphi(\boldsymbol{g})\pi(\boldsymbol{g})d\boldsymbol{g} + o(\frac{1}{|\mathcal{D}|\prod_{s=1}^{q} h_s}),$$

*where*

$$\varphi(\boldsymbol{g}) = \int \frac{1}{p_{u,i}(\boldsymbol{g}')}\cdot \bar{K}(\frac{\boldsymbol{g}-\boldsymbol{g}'}{h})\cdot \delta_{u,i}(\boldsymbol{g})\delta_{u,i}(\boldsymbol{g}')\pi(\boldsymbol{g}')d\boldsymbol{g}'$$

*is a bounded function of $\boldsymbol{g}$, $\bar{K}(\frac{\boldsymbol{g}-\boldsymbol{g}'}{\boldsymbol{h}}) = \int \prod_{s=1}^{q} K(t_s) K\left(\frac{g_s-g_s'}{h_s}+t_s\right) dt_1\cdots dt_q$. The variance of the N-DR estimator is*

$$Var(\mathcal{L}_{\mathrm{DR}}^{\mathrm{N}}(\hat{\mathbf{R}})) = \frac{1}{|\mathcal{D}|\prod_{s=1}^{q} h_s} \int \psi(\boldsymbol{g})\pi(\boldsymbol{g})d\boldsymbol{g} + o(\frac{1}{|\mathcal{D}|\prod_{s=1}^{q} h_s}),$$

*where*

$$\psi(\boldsymbol{g}) = \int \frac{1}{p_{u,i}(\boldsymbol{g}')}\cdot \bar{K}(\frac{\boldsymbol{g}-\boldsymbol{g}'}{h})\cdot \{\delta_{u,i}(\boldsymbol{g})-\hat{\delta}_{u,i}(\boldsymbol{g})\}\{\delta_{u,i}(\boldsymbol{g}')-\hat{\delta}_{u,i}(\boldsymbol{g}')\}\pi(\boldsymbol{g}')d\boldsymbol{g}'$$

*is a bounded function of $\boldsymbol{g}$.*

*Proof of Theorem 6.* We show the bias and variance of the N-IPS estimator, and the bias and variance of the N-DR estimator can be obtained similarly.

(a) For a given $\boldsymbol{g}$, by a similar argument of the proof of Theorem 3(a),

$$\mathbb{E}[\mathcal{L}_{\mathrm{IPS}}^{\mathrm{N}}(\hat{\mathbf{R}}|\boldsymbol{g})] = \mathbb{E}\left[\frac{\mathbb{I}(o_{u,i}=1)\cdot \boldsymbol{K}\left((\boldsymbol{g}_{u,i}-\boldsymbol{g})/\boldsymbol{h}\right)\cdot \delta_{u,i}(\boldsymbol{g})}{\prod_{s=1}^{q} h_s \cdot p_{u,i}(\boldsymbol{g})}\right]$$

$$= \mathbb{E}\left[\frac{1}{p_{u,i}(\boldsymbol{g})}\int \prod_{s=1}^{q}\frac{1}{h_s}K(\frac{g_{u,i}^s-g_s}{h_s})p(o_{u,i}=1,\boldsymbol{g}_{u,i}|x_{u,i})d\boldsymbol{g}_{u,i}\cdot \mathbb{E}\{\delta_{u,i}(\boldsymbol{g})|x_{u,i}\}\right].$$

Let $t_s = (g^s_{u,i} - g_s)/h_s$ for $s = 1, ..., q$, then $g^s_{u,i} = g_s + h_s t_s$, $d\boldsymbol{g}_{u,i} = dg^1_{u,i} \cdots dg^q_{u,i} = \prod^q_{s=1} h_s dt_1 \cdots dt_q$. By a Taylor expansion of $p(o_{u,i} = 1, g_1 + h_1 t_1, ..., g_q + h_q t_q | x_{u,i})$, we have

$$
\mathbb{E}\left[ \frac{1}{p_{u,i}(\boldsymbol{g})} \int \prod^q_{s=1} \frac{1}{h_s} K(\frac{g^s_{u,i} - g_s}{h_s}) p(o_{u,i} = 1, \boldsymbol{g}_{u,i} | x_{u,i}) d\boldsymbol{g}_{u,i} \cdot \mathbb{E}\{\delta_{u,i}(\boldsymbol{g}) | x_{u,i}\} \right]
$$

$$
= \mathbb{E}\left[ \frac{1}{p_{u,i}(\boldsymbol{g})} \int \cdots \int \prod^q_{s=1} K(t_s) p(o_{u,i} = 1, g_1 + h_1 t_1, ..., g_q + h_q t_q | x_{u,i}) dt_1 \cdots dt_q \cdot \mathbb{E}\{\delta_{u,i}(\boldsymbol{g}) | x_{u,i}\} \right]
$$

$$
= \mathbb{E}\Big[ \frac{1}{p_{u,i}(\boldsymbol{g})} \int \cdots \int \prod^q_{s=1} K(t_s) \Big\{ p(o_{u,i} = 1, \boldsymbol{g} | x_{u,i}) + \sum^q_{s=1} \frac{\partial p(o_{u,i} = 1, \boldsymbol{g} | x_{u,i})}{\partial g_s} h_s t_s
$$

$$
+ \sum^q_{s=1} \sum^q_{s'=1} \frac{\partial^2 p(o_{u,i} = 1, \boldsymbol{g} | x_{u,i})}{\partial g_s \partial g_{s'}} \frac{h_s h_{s'} t_s t_{s'}}{2} + o(\sum^q_{s=1} h^2_s) \Big\} dt_1 \cdots dt_q \cdot \mathbb{E}\{\delta_{u,i}(\boldsymbol{g}) | x_{u,i}\} \Big]
$$

$$
= \mathbb{E}\left[ \frac{1}{p_{u,i}(\boldsymbol{g})} \cdot \prod^q_{s=1} \int K(t_s) dt_s \cdot p(o_{u,i} = 1, \boldsymbol{g} | x_{u,i}) \cdot \mathbb{E}\{\delta_{u,i}(\boldsymbol{g}) | x_{u,i}\} \right] + 0
$$

$$
+ \mathbb{E}\Big[ \frac{1}{p_{u,i}(\boldsymbol{g})} \sum^q_{s=1} \int K(t_s) t^2_s dt_s \cdot \frac{\partial^2 p(o_{u,i} = 1, \boldsymbol{g} | x_{u,i})}{\partial g^2_s} \frac{h^2_s}{2} \cdot \mathbb{E}\{\delta_{u,i}(\boldsymbol{g}) | x_{u,i}\} \Big] + o(\sum^q_{s=1} h^2_s)
$$

$$
= \mathbb{E}\left[ \mathbb{E}\{\delta_{u,i}(\boldsymbol{g}) | x_{u,i}\} \right] + \frac{1}{2} \mu_2 \mathbb{E}\Big[ \sum^q_{s=1} \frac{\partial^2 p(o_{u,i} = 1, \boldsymbol{g} | x_{u,i})}{\partial g^2_s} \cdot h^2_s \cdot \delta_{u,i}(\boldsymbol{g}) \Big] + o(\sum^q_{s=1} h^2_s)
$$

$$
= \mathcal{L}^N_{\text{ideal}}(\hat{\mathbf{R}} | \boldsymbol{g}) + \frac{1}{2} \mu_2 \mathbb{E}\Big[ \sum^q_{s=1} \frac{\partial^2 p(o_{u,i} = 1, \boldsymbol{g} | x_{u,i})}{\partial g^2_s} \cdot h^2_s \cdot \delta_{u,i}(\boldsymbol{g}) \Big] + o(\sum^q_{s=1} h^2_s).
$$

Thus, the bias of $\mathcal{L}^N_{\text{IPS}}(\hat{\mathbf{R}})$ is

$$
\mathbb{E}[\mathcal{L}^N_{\text{IPS}}(\hat{\mathbf{R}})] - \mathcal{L}^N_{\text{ideal}}(\hat{\mathbf{R}}) = \int_{\boldsymbol{g}} \mathbb{E}\{\mathcal{L}^N_{\text{IPS}}(\hat{\mathbf{R}} | \boldsymbol{g}) - \mathcal{L}^N_{\text{ideal}}(\hat{\mathbf{R}} | \boldsymbol{g})\} \pi(\boldsymbol{g}) d\boldsymbol{g}
$$

$$
= \frac{1}{2} \mu_2 \left[ \sum^q_{s=1} h^2_s \int \mathbb{E}\Big\{ \frac{\partial^2 p(o_{u,i} = 1, \boldsymbol{g} | x_{u,i})}{\partial g^2_s} \cdot \delta_{u,i}(\boldsymbol{g}) \Big\} \pi(\boldsymbol{g}) d\boldsymbol{g} \right] + o(\sum^q_{s=1} h^2_s).
$$

(b) The variance of $\mathcal{L}^N_{\text{IPS}}(\hat{\mathbf{R}})$ can be represented as

$$
\text{Var}\{\mathcal{L}^N_{\text{IPS}}(\hat{\mathbf{R}})\} = \frac{1}{|\mathcal{D}|} \text{Var}\left[ \int \frac{\mathbb{I}(o_{u,i} = 1) \cdot \boldsymbol{K}((\boldsymbol{g}_{u,i} - \boldsymbol{g})/\boldsymbol{h}) \cdot \delta_{u,i}(\boldsymbol{g})}{\prod^q_{s=1} h_s \cdot p_{u,i}(\boldsymbol{g})} \pi(\boldsymbol{g}) d\boldsymbol{g} \right]
$$

$$
= \frac{1}{|\mathcal{D}|} \Bigg[ \mathbb{E}\left\{ \left( \int \frac{\mathbb{I}(o_{u,i} = 1) \cdot \boldsymbol{K}((\boldsymbol{g}_{u,i} - \boldsymbol{g})/\boldsymbol{h}) \cdot \delta_{u,i}(\boldsymbol{g})}{\prod^q_{s=1} h_s \cdot p_{u,i}(\boldsymbol{g})} \pi(\boldsymbol{g}) d\boldsymbol{g} \right)^2 \right\}
$$

$$
- \left\{ \mathbb{E}\left( \int \frac{\mathbb{I}(o_{u,i} = 1) \cdot \boldsymbol{K}((\boldsymbol{g}_{u,i} - \boldsymbol{g})/\boldsymbol{h}) \cdot \delta_{u,i}(\boldsymbol{g})}{\prod^q_{s=1} h_s \cdot p_{u,i}(\boldsymbol{g})} \pi(\boldsymbol{g}) d\boldsymbol{g} \right) \right\}^2 \Bigg].
$$

By the bias of the N-IPS estimator,

$$
\left\{ \mathbb{E}\left( \int \frac{\mathbb{I}(o_{u,i} = 1) \cdot \boldsymbol{K}((\boldsymbol{g}_{u,i} - \boldsymbol{g})/\boldsymbol{h}) \cdot \delta_{u,i}(\boldsymbol{g})}{\prod^q_{s=1} h_s \cdot p_{u,i}(\boldsymbol{g})} \pi(\boldsymbol{g}) d\boldsymbol{g} \right) \right\}^2 = [\mathcal{L}^N_{\text{ideal}}(\hat{\mathbf{R}})]^2 + O(\sum^q_{s=1} h^2_s) = O(1).
$$

On the other hand, note that

$$
\left( \int \frac{\mathbb{I}(o_{u,i} = 1) \cdot \boldsymbol{K}((\boldsymbol{g}_{u,i} - \boldsymbol{g})/\boldsymbol{h}) \cdot \delta_{u,i}(\boldsymbol{g})}{\prod^q_{s=1} h_s \cdot p_{u,i}(\boldsymbol{g})} \pi(\boldsymbol{g}) d\boldsymbol{g} \right)^2
$$

$$
= \int \int \frac{\mathbb{I}(o_{u,i} = 1)}{p_{u,i}(\boldsymbol{g}) p_{u,i}(\boldsymbol{g}')} \cdot \frac{1}{\prod^q_{s=1} h^2_s} \boldsymbol{K}\left( \frac{\boldsymbol{g}_{u,i} - \boldsymbol{g}}{\boldsymbol{h}} \right) \boldsymbol{K}\left( \frac{\boldsymbol{g}_{u,i} - \boldsymbol{g}'}{\boldsymbol{h}} \right) \cdot \delta_{u,i}(\boldsymbol{g}) \delta_{u,i}(\boldsymbol{g}') \pi(\boldsymbol{g}) \pi(\boldsymbol{g}') d\boldsymbol{g} d\boldsymbol{g}',
$$

we swap the order of integration and expectation, which leads to that

$$\mathbb{E}\left\{\left(\int \frac{\mathbb{I}(o_{u,i}=1)\cdot \boldsymbol{K}\left((\boldsymbol{g}_{u,i}-\boldsymbol{g})/\boldsymbol{h}\right)\cdot \delta_{u,i}(\boldsymbol{g})}{\prod_{s=1}^{q}h_s\cdot p_{u,i}(\boldsymbol{g})}\pi(\boldsymbol{g})d\boldsymbol{g}\right)^2\right\}$$

$$=\int\int \mathbb{E}\left[\frac{\mathbb{I}(o_{u,i}=1)}{p_{u,i}(\boldsymbol{g})p_{u,i}(\boldsymbol{g}')}\cdot \frac{1}{\prod_{s=1}^{q}h_s^2}\boldsymbol{K}\left(\frac{\boldsymbol{g}_{u,i}-\boldsymbol{g}}{\boldsymbol{h}}\right)\boldsymbol{K}\left(\frac{\boldsymbol{g}_{u,i}-\boldsymbol{g}'}{\boldsymbol{h}}\right)\cdot \delta_{u,i}(\boldsymbol{g})\delta_{u,i}(\boldsymbol{g}')\right]\pi(\boldsymbol{g})\pi(\boldsymbol{g}')d\boldsymbol{g}d\boldsymbol{g}'.$$

Let $\boldsymbol{g}_{u,i}=\boldsymbol{g}+\boldsymbol{h}\boldsymbol{t}$, i.e., $g_{u,i}^s=g_s+h_s t_s$, then $\boldsymbol{g}_{u,i}-\boldsymbol{g}'=(\boldsymbol{g}-\boldsymbol{g}')+\boldsymbol{h}\boldsymbol{t}$, we have

$$\mathbb{E}\left[\frac{\mathbb{I}(o_{u,i}=1)}{p_{u,i}(\boldsymbol{g})p_{u,i}(\boldsymbol{g}')}\cdot \frac{1}{\prod_{s=1}^{q}h_s^2}\boldsymbol{K}\left(\frac{\boldsymbol{g}_{u,i}-\boldsymbol{g}}{\boldsymbol{h}}\right)\boldsymbol{K}\left(\frac{\boldsymbol{g}_{u,i}-\boldsymbol{g}'}{\boldsymbol{h}}\right)\cdot \delta_{u,i}(\boldsymbol{g})\delta_{u,i}(\boldsymbol{g}')\right]$$

$$=\mathbb{E}\left[\frac{1}{p_{u,i}(\boldsymbol{g})p_{u,i}(\boldsymbol{g}')}\cdot \int\left\{\prod_{s=1}^{q}\frac{1}{h_s^2}K(t_s)K\left(\frac{g_s-g_s'}{h_s}+t_s\right)p(o_{u,i}=1,\boldsymbol{g}+\boldsymbol{h}\boldsymbol{t}|x_{u,i})\right\}dt_1\cdots dt_q\right.$$

$$\left.\cdot\,\mathbb{E}\left\{\delta_{u,i}(\boldsymbol{g})\delta_{u,i}(\boldsymbol{g}')\Big|x_{u,i}\right\}\right]$$

$$=\mathbb{E}\left[\frac{1}{p_{u,i}(\boldsymbol{g})p_{u,i}(\boldsymbol{g}')}\cdot \int\left\{\prod_{s=1}^{q}\frac{1}{h_s^2}K(t_s)K\left(\frac{g_s-g_s'}{h_s}+t_s\right)p(o_{u,i}=1,\boldsymbol{g}|x_{u,i})+O(\sum_{s=1}^{q}h_s)\right\}dt_1\cdots dt_q\right.$$

$$\left.\cdot\,\mathbb{E}\left\{\delta_{u,i}(\boldsymbol{g})\delta_{u,i}(\boldsymbol{g}')\Big|x_{u,i}\right\}\right]$$

$$=\mathbb{E}\left[\frac{1}{p_{u,i}(\boldsymbol{g}')}\cdot \int\prod_{s=1}^{q}\frac{1}{h_s}K(t_s)K\left(\frac{g_s-g_s'}{h_s}+t_s\right)dt_1\cdots dt_q\cdot \mathbb{E}\left\{\delta_{u,i}(\boldsymbol{g})\delta_{u,i}(\boldsymbol{g}')\Big|x_{u,i}\right\}\right]\cdot\left\{1+O(\sum_{s=1}^{q}h_s)\right\}$$

$$=\mathbb{E}\left[\frac{1}{p_{u,i}(\boldsymbol{g}')}\cdot \int\prod_{s=1}^{q}\frac{1}{h_s}K(t_s)K\left(\frac{g_s-g_s'}{h_s}+t_s\right)dt_1\cdots dt_q\cdot \delta_{u,i}(\boldsymbol{g})\delta_{u,i}(\boldsymbol{g}')\right]\cdot\left\{1+O(\sum_{s=1}^{q}h_s)\right\}.$$

Thus, define $\bar{K}(\frac{\boldsymbol{g}-\boldsymbol{g}'}{\boldsymbol{h}})=\int\prod_{s=1}^{q}K(t_s)K\left(\frac{g_s-g_s'}{h_s}+t_s\right)dt_1\cdots dt_q$, then

$$\mathbb{E}\left\{\left(\int \frac{\mathbb{I}(o_{u,i}=1)\cdot \boldsymbol{K}\left((\boldsymbol{g}_{u,i}-\boldsymbol{g})/\boldsymbol{h}\right)\cdot \delta_{u,i}(\boldsymbol{g})}{\prod_{s=1}^{q}h_s\cdot p_{u,i}(\boldsymbol{g})}\pi(\boldsymbol{g})d\boldsymbol{g}\right)^2\right\}$$

$$=\int \mathbb{E}\left[\int \frac{1}{p_{u,i}(\boldsymbol{g}')}\cdot \frac{1}{\prod_{s=1}^{q}h_s}\bar{K}(\frac{\boldsymbol{g}-\boldsymbol{g}'}{\boldsymbol{h}})\cdot \delta_{u,i}(\boldsymbol{g})\delta_{u,i}(\boldsymbol{g}')\pi(\boldsymbol{g}')d\boldsymbol{g}'\right]\cdot\left\{1+O(\sum_{s=1}^{q}h_s)\right\}.\pi(\boldsymbol{g})d\boldsymbol{g}$$

$$\triangleq \int \varphi(\boldsymbol{g})\pi(\boldsymbol{g})d\boldsymbol{g}\frac{1}{\prod_{s=1}^{q}h_s}(1+o(1)),$$

where

$$\varphi(\boldsymbol{g})=\int \frac{1}{p_{u,i}(\boldsymbol{g}')}\cdot \bar{K}(\frac{\boldsymbol{g}-\boldsymbol{g}'}{\boldsymbol{h}})\cdot \delta_{u,i}(\boldsymbol{g})\delta_{u,i}(\boldsymbol{g}')\pi(\boldsymbol{g}')d\boldsymbol{g}'$$

is a bounded function of $\boldsymbol{g}$. Therefore, we have

$$\text{Var}(\mathcal{L}_{\text{IPS}}^{\text{N}}(\hat{\mathbf{R}}))=\frac{1}{|\mathcal{D}|\prod_{s=1}^{q}h_s}\int \varphi(\boldsymbol{g})\pi(\boldsymbol{g})d\boldsymbol{g}+o(\frac{1}{|\mathcal{D}|\prod_{s=1}^{q}h_s}).$$

$\square$

**Theorem 7** (Optimal Bandwidth of N-IPS and N-DR). *Under Assumptions 1–3 and 6, and assume that $h=h_1=\cdots=h_q$, then*

*(a) the optimal bandwidth for the N-IPS estimator in terms of the asymptotic mean-squared error is*

$$h_{\text{N-IPS}}^*=\left[\frac{\int \varphi(\boldsymbol{g})\pi(\boldsymbol{g})d\boldsymbol{g}}{4|\mathcal{D}|\left(\frac{1}{2}\mu_2\sum_{s=1}^{q}\int \mathbb{E}\left[\frac{\partial^2 p(o_{u,i}=1,\boldsymbol{g}_{u,i}=\boldsymbol{g}|x_{u,i})}{\partial g_s^2}\cdot \delta_{u,i}(\boldsymbol{g})\right]\cdot \pi(\boldsymbol{g})d\boldsymbol{g}\right)^2}\right]^{1/(4+q)},$$

*where $\varphi(\boldsymbol{g})$ is defined in Theorem 6;*

*(b) the optimal bandwidth for the N-DR estimator in terms of the asymptotic mean-squared error is*

$$h_{\text{N-DR}}^* = \left[ \frac{\int \psi(\boldsymbol{g})\pi(\boldsymbol{g})d\boldsymbol{g}}{4|\mathcal{D}|\left(\frac{1}{2}\mu_2 \sum_{s=1}^q \int \mathbb{E}\left[\frac{\partial^2 p(o_{u,i}=1, \boldsymbol{g}_{u,i}=\boldsymbol{g}|x_{u,i})}{\partial g_s^2} \cdot \{\delta_{u,i}(\boldsymbol{g}) - \hat{\delta}_{u,i}(\boldsymbol{g})\}\right] \cdot \pi(\boldsymbol{g})d\boldsymbol{g}\right)^2} \right]^{1/(4+q)}.$$

*Proof of Theorem 7.* This conclusion can be derived similarly from the proof of Theorem 4.

$\square$

Then, we obtain the uniform tail bound of the N-IPS and N-DR estimators with multi-dimensional treatment representation.

**Lemma 5** (Uniform Tail Bound of N-IPS and N-DR). *Under Assumptions 1–3, 5, and 6, and suppose that $K(t) \leq M_K$, then we have with probability at least $1 - \eta$,*

*(a)*

$$\sup_{\hat{\mathbf{R}} \in \mathcal{F}} \left| \mathbb{E}[\mathcal{L}_{\text{IPS}}^{\text{N}}(\hat{\mathbf{R}})] - \mathcal{L}_{\text{IPS}}^{\text{N}}(\hat{\mathbf{R}}) \right| \leq \frac{2M_p(M_K)^q}{\prod_{s=1}^q h_s} \mathcal{R}(\mathcal{F}) + \frac{5M_p M_\delta(M_K)^q}{2\prod_{s=1}^q h_s} \sqrt{\frac{2}{|\mathcal{D}|}\log(\frac{4}{\eta})};$$

*(b)*

$$\sup_{\hat{\mathbf{R}} \in \mathcal{F}} \left| \mathbb{E}[\mathcal{L}_{\text{DR}}^{\text{N}}(\hat{\mathbf{R}})] - \mathcal{L}_{\text{DR}}^{\text{N}}(\hat{\mathbf{R}}) \right| \leq \frac{2M_p(M_K)^q}{\prod_{s=1}^q h_s} \mathcal{R}(\mathcal{F}) + \frac{5M_p M_{|\delta-\hat{\delta}|}(M_K)^q}{2\prod_{s=1}^q h_s} \sqrt{\frac{2}{|\mathcal{D}|}\log(\frac{4}{\eta})}.$$

*Proof of Lemma 5.* It is sufficient to prove the uniform tail bound of $\mathcal{L}_{\text{IPS}}^{\text{N}}(\hat{\mathbf{R}})$, and the result for $\mathcal{L}_{\text{DR}}^{\text{N}}(\hat{\mathbf{R}})$ can be derived by a similar argument.

Note that

$$\sup_{\hat{\mathbf{R}} \in \mathcal{F}} \left| \mathcal{L}_{\text{IPS}}^{\text{N}}(\hat{\mathbf{R}}) - \mathbb{E}\{\mathcal{L}_{\text{IPS}}^{\text{N}}(\hat{\mathbf{R}})\} \right|$$

$$\leq \int \pi(\boldsymbol{g}) \sup_{\hat{\mathbf{R}} \in \mathcal{F}} \left| \frac{1}{|\mathcal{D}|} \sum_{(u,i) \in \mathcal{D}} \left\{ \frac{\mathbb{I}(o_{u,i}=1) \cdot \boldsymbol{K}((\boldsymbol{g}_{u,i}-\boldsymbol{g})/\boldsymbol{h}) \cdot \delta_{u,i}(\boldsymbol{g})}{\prod_{s=1}^q h_s \cdot \hat{p}_{u,i}(\boldsymbol{g})} \right\} \right.$$

$$\left. - \mathbb{E}\left\{ \frac{\mathbb{I}(o_{u,i}=1) \cdot \boldsymbol{K}((\boldsymbol{g}_{u,i}-\boldsymbol{g})/\boldsymbol{h}) \cdot \delta_{u,i}(\boldsymbol{g})}{\prod_{s=1}^q h_s \cdot \hat{p}_{u,i}(\boldsymbol{g})} \right\} \right| d\boldsymbol{g}.$$

For all prediction model $\hat{\mathbf{R}} \in \mathcal{F}$ and $\boldsymbol{g}$, we have

$$\frac{1}{|\mathcal{D}|} \left| \frac{\mathbb{I}(o_{u,i}=1) \cdot \boldsymbol{K}((\boldsymbol{g}_{u,i}-\boldsymbol{g})/\boldsymbol{h}) \cdot \delta_{u,i}(\boldsymbol{g})}{\prod_{s=1}^q h_s \cdot \hat{p}_{u,i}(\boldsymbol{g})} - \frac{\mathbb{I}(o_{u',i'}=1) \cdot \boldsymbol{K}((\boldsymbol{g}_{u',i'}-\boldsymbol{g})/\boldsymbol{h}) \cdot \delta_{u',i'}(f)}{\prod_{s=1}^q h_s \cdot \hat{p}_{u',i'}(\boldsymbol{g})} \right| \leq \frac{M_p M_\delta(M_K)^q}{\prod_{s=1}^q h_s |\mathcal{D}|},$$

then applying Lemma 2 yields that with probability at least $1 - \eta/2$

$$\sup_{\hat{\mathbf{R}} \in \mathcal{F}} \left| \mathbb{E}\left\{ \frac{\mathbb{I}(o_{u,i}=1) \cdot \boldsymbol{K}((\boldsymbol{g}_{u,i}-\boldsymbol{g})/\boldsymbol{h}) \cdot \delta_{u,i}(\boldsymbol{g})}{\prod_{s=1}^q h_s \cdot \hat{p}_{u,i}(\boldsymbol{g})} \right\} - \frac{1}{|\mathcal{D}|} \sum_{(u,i) \in \mathcal{D}} \left\{ \frac{\mathbb{I}(o_{u,i}=1) \cdot \boldsymbol{K}((\boldsymbol{g}_{u,i}-\boldsymbol{g})/\boldsymbol{h}) \cdot \delta_{u,i}(\boldsymbol{g})}{\prod_{s=1}^q h_s \cdot \hat{p}_{u,i}(\boldsymbol{g})} \right\} \right|$$

$$\leq \mathbb{E}\left[ \sup_{\hat{\mathbf{R}} \in \mathcal{F}} \left| \mathbb{E}\left\{ \frac{\mathbb{I}(o_{u,i}=1) \cdot \boldsymbol{K}((\boldsymbol{g}_{u,i}-\boldsymbol{g})/\boldsymbol{h}) \cdot \delta_{u,i}(\boldsymbol{g})}{\prod_{s=1}^q h_s \cdot \hat{p}_{u,i}(\boldsymbol{g})} \right\} - \frac{1}{|\mathcal{D}|} \sum_{(u,i) \in \mathcal{D}} \left\{ \frac{\mathbb{I}(o_{u,i}=1) \cdot \boldsymbol{K}((\boldsymbol{g}_{u,i}-\boldsymbol{g})/\boldsymbol{h}) \cdot \delta_{u,i}(\boldsymbol{g})}{\prod_{s=1}^q h_s \cdot \hat{p}_{u,i}(\boldsymbol{g})} \right\} \right| \right]$$

$$+ \frac{M_p M_\delta(M_K)^q}{2\prod_{s=1}^q h_s} \sqrt{\frac{2}{|\mathcal{D}|}\log(\frac{4}{\eta})}$$

$$\leq \frac{M_p(M_K)^q}{\prod_{s=1}^q h_s} 2\mathbb{E}[\mathcal{R}(\mathcal{F})] + \frac{M_p M_\delta(M_K)^q}{2\prod_{s=1}^q h_s} \sqrt{\frac{2}{|\mathcal{D}|}\log(\frac{4}{\eta})},$$

where the last inequality holds by Lemma 3 and

$$\frac{\mathbb{I}(o_{u,i}=1)\cdot \boldsymbol{K}\left((\boldsymbol{g}_{u,i}-\boldsymbol{g})/\boldsymbol{h}\right)}{\prod_{s=1}^{q}h_s\cdot \hat{p}_{u,i}(\boldsymbol{g})}\leq \frac{M_p(M_K)^q}{\prod_{s=1}^{q}h_s}.$$

Applying (A.5) yields that with probability at least $1-\eta$

$$\sup_{\hat{\mathbf{R}}\in\mathcal{F}}\left|\mathbb{E}\left\{\frac{\mathbb{I}(o_{u,i}=1)\cdot \boldsymbol{K}\left((\boldsymbol{g}_{u,i}-\boldsymbol{g})/\boldsymbol{h}\right)\cdot \delta_{u,i}(\boldsymbol{g})}{\prod_{s=1}^{q}h_s\cdot \hat{p}_{u,i}(\boldsymbol{g})}\right\}-\frac{1}{|\mathcal{D}|}\sum_{(u,i)\in\mathcal{D}}\left\{\frac{\mathbb{I}(o_{u,i}=1)\cdot \boldsymbol{K}\left((\boldsymbol{g}_{u,i}-\boldsymbol{g})/\boldsymbol{h}\right)\cdot \delta_{u,i}(\boldsymbol{g})}{\prod_{s=1}^{q}h_s\cdot \hat{p}_{u,i}(\boldsymbol{g})}\right\}\right|$$

$$\leq \frac{2M_p(M_K)^q}{\prod_{s=1}^{q}h_s}\mathcal{R}(\mathcal{F})+\frac{5M_pM_\delta(M_K)^q}{2\prod_{s=1}^{q}h_s}\sqrt{\frac{2}{|\mathcal{D}|}\log(\frac{4}{\eta})},$$

which implies Lemma 5(a) by noting that $\int \pi(\boldsymbol{g})d\boldsymbol{g}=1$.

$\square$

Let

$$\mathbf{R}^{\dagger}=\arg\min_{\hat{\mathbf{R}}\in\mathcal{F}}\mathcal{L}^{\mathrm{N}}_{\mathrm{IPS}}(\hat{\mathbf{R}}),\quad \mathbf{R}^{\ddagger}=\arg\min_{\hat{\mathbf{R}}\in\mathcal{F}}\mathcal{L}^{\mathrm{N}}_{\mathrm{DR}}(\hat{\mathbf{R}}).$$

The following Theorem 8 shows the generalization error bounds of the N-IPS and N-DR estimators.

**Theorem 8** (Generalization Error Bounds of N-IPS and N-DR). *Under Assumptions 1–3, 5, and 6, and suppose that $K(t)\leq M_K$, then we have with probability at least $1-\eta$,*

*(a)*

$$\mathcal{L}^{\mathrm{N}}_{\mathrm{ideal}}(\hat{\mathbf{R}}^{\dagger})\leq \min_{\hat{\mathbf{R}}\in\mathcal{F}}\mathcal{L}^{\mathrm{N}}_{\mathrm{ideal}}(\hat{\mathbf{R}})+\mu_2\left[\sum_{s=1}^{q}h_s^2\int \mathbb{E}\left\{\frac{\partial^2 p(o_{u,i}=1,\boldsymbol{g}_{u,i}=\boldsymbol{g}|x_{u,i})}{\partial g_s^2}\cdot \delta_{u,i}(\boldsymbol{g})\right\}\pi(\boldsymbol{g})d\boldsymbol{g}\right]$$

$$+\frac{4M_p(M_K)^q}{\prod_{s=1}^{q}h_s}\mathcal{R}(\mathcal{F})+\frac{5M_pM_\delta(M_K)^q}{\prod_{s=1}^{q}h_s}\sqrt{\frac{2}{|\mathcal{D}|}\log(\frac{4}{\eta})}+o(\sum_{s=1}^{q}h_s^2);$$

*(b)*

$$\mathcal{L}^{\mathrm{N}}_{\mathrm{ideal}}(\hat{\mathbf{R}}^{\ddagger})\leq \min_{\hat{\mathbf{R}}\in\mathcal{F}}\mathcal{L}^{\mathrm{N}}_{\mathrm{ideal}}(\hat{\mathbf{R}})+\mu_2\left[\sum_{s=1}^{q}h_s^2\int \mathbb{E}\left\{\frac{\partial^2 p(o_{u,i}=1,\boldsymbol{g}_{u,i}=\boldsymbol{g}|x_{u,i})}{\partial g_s^2}\cdot \left(\delta_{u,i}(\boldsymbol{g})-\hat{\delta}_{u,i}(\boldsymbol{g})\right)\right\}\pi(\boldsymbol{g})d\boldsymbol{g}\right]$$

$$+\frac{4M_p(M_K)^q}{\prod_{s=1}^{q}h_s}\mathcal{R}(\mathcal{F})+\frac{5M_pM_{|\delta-\hat{\delta}|}(M_K)^q}{\prod_{s=1}^{q}h_s}\sqrt{\frac{2}{|\mathcal{D}|}\log(\frac{4}{\eta})}+o(\sum_{s=1}^{q}h_s^2).$$

*Proof of Theorem 8.* This conclusion can be derived similarly from the proof of Theorem 5. $\square$

# D PSEUDO-CODES FOR PROPENSITY LEARNING, N-IPS, N-DR-JL AND N-MRDR-JL

To learn the propensity model $p_{u,i}(\boldsymbol{g})$, we have the following holds that

$$\frac{1}{p_{u,i}(\boldsymbol{g})}=\frac{c}{\mathbb{P}(o=1\mid \boldsymbol{x})}\cdot \frac{\mathbb{P}(L=1)}{\mathbb{P}(L=0)}\cdot \frac{\mathbb{P}(L=0\mid \boldsymbol{x},\boldsymbol{g})}{\mathbb{P}(L=1\mid \boldsymbol{x},\boldsymbol{g})}\propto \frac{1}{\mathbb{P}(o=1\mid \boldsymbol{x})}\cdot \frac{\mathbb{P}(L=0\mid \boldsymbol{x},\boldsymbol{g})}{\mathbb{P}(L=1\mid \boldsymbol{x},\boldsymbol{g})}.$$

The constant $c$ and $\mathbb{P}(L=1)/\mathbb{P}(L=0)$ can be ignored and $\mathbb{P}(o=1\mid \boldsymbol{x})$ can be estimated by using the existing methods such as naive Bayes or logistic regression with or without a few unbiased ratings, respectively. Therefore, we need to train the model that estimates $\mathbb{P}(L=1\mid \boldsymbol{x},\boldsymbol{g})$, as shown in Algorithm 1. In addition, the learning algorithms for N-IPS, N-DR-JL and N-MRDR-JL are shown in Algorithms 2-4. Specifically, for the N-MRDR-JL learning algorithm, the imputation model parameters are optimized by minimizing the following loss

$$\mathcal{L}_e^{\mathrm{N-MRDR}}(\hat{\mathbf{R}})=\int |\mathcal{D}|^{-1}\sum_{(u,i)\in\mathcal{D}}\frac{\mathbb{I}(o_{u,i}=1)\cdot (1-p_{u,i}(\boldsymbol{g}))\cdot K\left((\boldsymbol{g}_{u,i}-\boldsymbol{g})/h\right)\cdot (\delta_{u,i}(\boldsymbol{g})-\hat{\delta}_{u,i}(\boldsymbol{g}))^2}{h\cdot p_{u,i}^2(\boldsymbol{g})}\pi(\boldsymbol{g})d\boldsymbol{g}.$$

Compared to the $\mathcal{L}_e^{\mathrm{N-DR}}(\hat{\mathbf{R}})$, the $\mathcal{L}_e^{\mathrm{N-MRDR}}(\hat{\mathbf{R}})$ loss has variance reduction property.

---

**Algorithm 1:** The Proposed Propensity Learning Algorithm

---

**Input:** the observation matrix $\mathbf{O}$ and the representation space $\mathcal{G}$.

1 Using previous methods such as logistic regression to train a model to estimate $\mathbb{P}(o = 1 \mid \boldsymbol{x})$;

2 **while** *stopping criteria is not satisfied* **do**

3      Sample a batch of user-item pairs $\{(u_j, i_j)\}_{j=1}^J$ with $o_{u,i} = 1$ to generate samples $\{(\boldsymbol{x}_{u_j, i_j}, \boldsymbol{g}_{u_j, i_j})\}$ with positive label $(L = 1)$;

4      Uniformly sample a batch of treatments $\{\boldsymbol{g}'_{u_k, i_k}\}_{k=1}^K \subset \mathcal{G}$ to generate samples $\{(\boldsymbol{x}_{u_j, i_j}, \boldsymbol{g}'_{u_k, i_k})\}$ with negative label $(L = 0)$;

5      Using gradient descent to train a logistic regression model that estimates $\mathbb{P}(L = 1 \mid \boldsymbol{x}, \boldsymbol{g})$ using the positive samples and negative samples;

6 **end**

**Output:** the propensity model that estimates $\mathbb{P}(L = 1 \mid \boldsymbol{x}, \boldsymbol{g})$.

---

**Algorithm 2:** The Proposed N-IPS Learning Algorithm

---

**Input:** the observed ratings $\mathbf{Y}^o$, the representation space $\mathcal{G}$, the pre-specified $\pi(\boldsymbol{g})$, the pre-specified kernel function $K(\cdot)$ and the propensity model.

1 **while** *stopping criteria is not satisfied* **do**

2      Sample a batch of user-item pairs $\{(u_j, i_j)\}_{j=1}^J$ from $\mathcal{O}$;

3      Calculate $\hat{p}_{u_j, i_j}(\boldsymbol{g})$ using the propensity model;

4      Update $\theta$ by descending along the gradient $\nabla_\theta \mathcal{L}_{\text{IPS}}^{\text{N}}(\hat{\mathbf{R}})$;

5 **end**

**Output:** the debiased prediction model $f_\theta(x)$.

---

## E    Semi-synthetic Experiment Details

Following the previous studies (Schnabel et al., 2016; Wang et al., 2019; Guo et al., 2021), we set propensity $p_{u,i} = p\alpha^{\max(0, 4 - r_{u,i})}$ for each user-item pair to introduce selection bias. Meanwhile, to investigate the effect of the neighborhood effect, we randomly block off some user rows and item columns to get the mask matrix. Specifically, we let $m_u \sim \text{Bern}(p_u), \forall u \in \mathcal{U}$, $m_i \sim \text{Bern}(p_i), \forall i \in \mathcal{I}$ and $m_{u,i} = m_u \cdot m_i$, where $\text{Bern}(\cdot)$ denotes the Bernoulli distribution and $p_u$ and $p_i$ are the mask ratio for user and item. We set $p_u$ and $p_i$ equal to 1 in **RQ1**, and set $p_u = (|\mathcal{U}| - n_u)/|\mathcal{U}|$ and $p_i = (|\mathcal{I}| - n_i)/|\mathcal{I}|$ in **RQ2**, where $|\mathcal{U}|$ and $|\mathcal{I}|$ are the total user and item numbers, and $n_u$ and $n_i$ are the user and item mask numbers, respectively. We tune the $n_u \in \{50, 150, 250, 350\}$ and let $n_i = n_u \cdot |\mathcal{I}|/|\mathcal{U}|$. Then we obtain the propensities with $p_{u,i} \leftarrow p_{u,i} \cdot m_{u,i} = p\alpha^{\max(0, 4 - r_{u,i})} \cdot m_{u,i}$. For different mask numbers, we adjust $p$ for unmasked user-item pairs to ensure the total observed sample is 5% of the entire matrix (Schnabel et al., 2016). The different $n_u$ and $n_i$ correspond to the different strengths of the neighborhood effect. Next, we follow Wang et al. (2019); Guo et al. (2021) to add a uniform distributed variable to introduce noise to obtain the estimated propensities $\frac{1}{\hat{p}_{u,i}} = \frac{\beta}{p_{u,i}} + \frac{1 - \beta}{p_o}$, where $\beta$ is from a uniform distribution $U(0, 1)$ and $p_o = \frac{1}{|\mathcal{D}|} \sum_{(u,i) \in \mathcal{D}} o_{u,i}$.

## F    Real-World Experiment Details

**Dataset.** We verify the effectiveness of the proposed estimators on three real-world datasets. **Coat** [3] contains 6,960 MNAR ratings and 4,640 missing-at-random (MAR) ratings. Both MNAR and MAR ratings are from 290 users and 300 items. **Yahoo! R3** [4] contains 311,704 MNAR ratings and 54,000 MAR ratings. The MNAR ratings are from 15,400 users and 1,000 items, and the MAR ratings are from the first 5,400 users and 1,000 items. For both datasets, ratings are binarized to 1 if $r_{u,i} \geq 3$, and 0 otherwise. **KuaiRec** [5] (Gao et al., 2022) is a public large-scale industrial dataset, which contains

---

[3] https://www.cs.cornell.edu/~schnabts/mnar/

[4] http://webscope.sandbox.yahoo.com/

[5] https://github.com/chongminggao/KuaiRec

---

**Algorithm 3:** The Proposed N-DR-JL Learning Algorithm

---

**Input:** the observed ratings $\mathbf{Y}^o$, the observation matrix $\mathbf{O}$, the representation space $\mathcal{G}$, the pre-specified $\pi(\boldsymbol{g})$, the pre-specified kernel function $K(\cdot)$ and the propensity model.

**while** *stopping criteria is not satisfied* **do**

    **for** *number of steps for training the imputation model* **do**

        Sample a batch of user-item pairs $\{(u_j, i_j)\}_{j=1}^J$ from $\mathcal{O}$;

        Calculate $\hat{p}_{u_j, i_j}(\boldsymbol{g})$ using the propensity model;

        Update $\phi_{\boldsymbol{g}}$ by descending along the gradient $\nabla_{\phi_{\boldsymbol{g}}} \mathcal{L}_e^{\mathrm{N-DR}}(\hat{\mathbf{R}})$;

    **end**

    **for** *number of steps for training the prediction model* **do**

        Sample a batch of user-item pairs $\{(u_k, i_k)\}_{k=1}^K$ from $\mathcal{D}$;

        Calculate $\hat{p}_{u_k, i_k}(\boldsymbol{g})$ using the propensity model for user-item pair with $o_{u_k, i_k} = 1$;

        Update $\theta$ by descending along the gradient $\nabla_{\theta} \mathcal{L}_{\mathrm{DR}}^{\mathrm{N}}(\hat{\mathbf{R}})$;

    **end**

**end**

**Output:** the debiased prediction model $f_{\theta}(x)$.

---

**Algorithm 4:** The Proposed N-MRDR-JL Learning Algorithm

---

**Input:** the observed ratings $\mathbf{Y}^o$, the observation matrix $\mathbf{O}$, the representation space $\mathcal{G}$, the pre-specified $\pi(\boldsymbol{g})$, the pre-specified kernel function $K(\cdot)$ and the propensity model.

**while** *stopping criteria is not satisfied* **do**

    **for** *number of steps for training the imputation model* **do**

        Sample a batch of user-item pairs $\{(u_j, i_j)\}_{j=1}^J$ from $\mathcal{O}$;

        Calculate $\hat{p}_{u_j, i_j}(\boldsymbol{g})$ using the propensity model;

        Update $\phi_{\boldsymbol{g}}$ by descending along the gradient $\nabla_{\phi_{\boldsymbol{g}}} \mathcal{L}_e^{\mathrm{N-MRDR}}(\hat{\mathbf{R}})$;

    **end**

    **for** *number of steps for training the prediction model* **do**

        Sample a batch of user-item pairs $\{(u_k, i_k)\}_{k=1}^K$ from $\mathcal{D}$;

        Calculate $\hat{p}_{u_k, i_k}(\boldsymbol{g})$ using the propensity model for user-item pair with $o_{u_k, i_k} = 1$;

        Update $\theta$ by descending along the gradient $\nabla_{\theta} \mathcal{L}_{\mathrm{DR}}^{\mathrm{N}}(\hat{\mathbf{R}})$;

    **end**

**end**

**Output:** the debiased prediction model $f_{\theta}(x)$.

---

4,676,570 video watching ratio records from 1,411 users for 3,327 videos. The records less than 2 are set to 0 and otherwise are set to 1.

**Experimental Details.** All the experiments are implemented on PyTorch with Adam as the optimizer. For all experiments, we use NVIDIA GeForce RTX 3090 as the computing resource. We tune the learning rate in $\{0.005, 0.01, 0.05, 0.1\}$ and weight decay in $[1e-6, 1e-2]$. We tune bandwidth value in $\{40, 45, 50, 55, 60\}$ for **Coat**, $\{1000, 1500, 2000, 2500, 3000\}$ for **Yahoo! R3** and $\{100, 150, 200, 250, 300\}$ for **KuaiRec**.

