# OpenReview forum: "Be Aware of the Neighborhood Effect: Modeling Selection Bias under Interference"
_ICLR.cc/2024/Conference — ICLR 2024 poster_

### Official Review · Reviewer_1Wkk · 2023-10-22

**Soundness:** 3 good
**Presentation:** 3 good
**Contribution:** 3 good
**Rating:** 8
**Confidence:** 3

**Summary:**

In this paper, the authors introduce a new debias problem under the causal inference framework for handling selection bias in recommendation systems in the presence of the neighboring effect: when the potential outcome for one user-item pair does vary with the treatments assigned to other user-item pairs. The potential outcome, treatment and a new ideal loss are defined so as to include both the selection bias AND the neighborhood interference effect. Follow two new estimators to estimate the newly designed ideal loss: neighborhood inverse propensity score (N-IPS) and neighborhood doubly robust (N-DR).

**Strengths:**

The paper is well written and quite enjoyable to read.

The newly proposed ideal loss and estimators are well theoretically investigated:

-The difference between the two losses with and without neighborhood effect (Theorem 2) is studied.

-The new ideal loss is shown as identifiable (Theorem 1).

-The first proposed estimator tackles the case when $\bold{g}_{u,i}$, the treatment representation vector, is a continuous probability density via a smoothing symmetric kernel function (ex: Epanechnikov or Gaussian kernels). The N-DR estimator is derived similarly.  -Bias and variance of both estimators are computed with tail and generalization error bounds provided (Theorem 5).

Then, it is shown how to estimate the propensity score for the joint effect of the treatment and the treatment representation vector for the neighboring effect.

The approach is accompanied first with experiments on semi-synthetic data (based on MovieLens 100K) to:
1) assess whether the proposed estimators provide a more accurate estimation for the ideal loss compared to the state-of-the-art methods when neighboring interference is observed.
2) measure the influence of the neighborhood effect strength on the estimation accuracy.
On the semi-synthetic dataset, for all interference strengths, N versions of DR or MRDR are giving better accuracy (lower relative error) than DR and MRDR but also are less harmed by the interference strength for 6 different methods of predicting the ratings.

Real-world experimentation is done on Coat, Yahoo! R3 and KuaiRec for which MSE, AUC and NDCG are evaluated N-* are usually in the 3 best results.

**Weaknesses:**

I would develop some explanations for the experimental part even in the appendix. Cf. questions.

Minor, typos:

-p.2: “In addition, we introduces”

-p.3: “... both … leads”

-p.9: “For the methods require propensity”, “the three choice”, “Guassian”, “on the a prior”

-p.10: “Early literature focus”

Also, when Figure 1 is first introduced in Introduction section, we don’t yet have the preliminaries content and all elements are not fully defined such as $\bold{g}_{u,i}$ which can make it difficult to understand at first sight.

**Questions:**

Q1: In practice, how do you specify the probability density function of $\bold{g}$?

Q2: In practice, what is the best choice between Epanechnikov and Gaussian kernels? Experiments seem to have been done only with the Gaussian kernel.

Q3: Can you please detail again for the semi-synthetic experiments how do you define the set of $\mathcal{N)_{u,i}$ as the set of historical user and item interactions for the neighbors of (u,i) who do have an influence on user u?

Q4: How does $p_{u,i} = p \alpha^{max(0,4-r_{u, i})}$ account for the neighboring effect too?

Q5: c is chosen to be the median of all g_u,i according to p. 7 but g_u,i is also defined depending on c. Can you please explain?

Q6: Does KuaiRec specify the MAR and MAR watching ratio records? If not, how to measure the neighboring effect?

Q7: How p is defined in $p_{u,i} = p \alpha^{max(0,4-r_{u, i})}$ is explained in p. 28. Can you please elaborate: “we adjust p to ensure the total observed sample is 5% of the entire matrix”?

=== AFTER REBUTTAL ===

I thank the authors for taking the time to answer my questions that are now addressed. Many thanks for the added experiments that show the results are not tight to the choice of the kernel.
Hence, I upgrade my score to Accept.

---

> ### Author Response · Authors · 2023-11-23
> **Please kindly find our concise and clear rebuttal below for addressing your current concerns [Q1-Q4]**
>
> We sincerely appreciate the reviewer’s great efforts and insightful comments to improve our manuscript. In below, we address these concerns point by point and try our best to update the manuscript accordingly.
>
> > **[Q1] In practice, how do you specify the probability density function of $\boldsymbol{g}$.**
>
> **Response:** We thank the reviewer for the comment. In our experiments, we choose the one with the better debiasing performance among **a uniform distribution or a Dirichlet distribution** for $\boldsymbol{g}. In practice, some other distributions can also be chosen.
>
> > **[Q2] In practice, what is the best choice between Epanechnikov and Gaussian kernels? Experiments seem to have been done only with the Gaussian kernel.**
>
> **Response:** We thank the reviewer for the question. Yes, in our original manuscript, the experiments are only done with Gaussian kernel. For exploring and comparing the Epanechnikov and Gaussian kernels, we add the experiments that our methods adopting Epanechnikov kernel on all three dataset (Page 8, Table 2) in our revised manuscript. The results are shown below, the best two results are bolded in IPS based, DR-JL based and MRDR-JL based methods, respectively.
>
> |Dataset| |Coat| || |Yahoo! R3| || |KuaiRec| |
> |:--|:--:|:--:|:--:|:--:|:--:|:--:|:--:|:--:|:--:|:--:|:--:|
> |Method|MSE $\downarrow$|AUC $\uparrow$|NDCG@5 $\uparrow$||MSE  $\downarrow$|AUC $\uparrow$|NDCG@5 $\uparrow$||MSE  $\downarrow$|AUC $\uparrow$|NDCG@5 $\uparrow$|
> |N-IPS [LR, Gaussian]|0.212|0.742|**0.678**||0.226|0.693|**0.664**||0.092|**0.796**|**0.585**|
> |N-IPS [LR, Epanechnikov]|0.224|**0.746**|0.645||0.242|**0.703**|**0.673**||0.094|**0.794**|**0.582**|
> |N-IPS [NB, Gaussian]|**0.206**|0.744|**0.648**||**0.196**|**0.693**|0.658||**0.049**|0.785|0.579|
> |N-IPS [NB, Epanechnikov]|**0.210**|**0.753**|0.646||**0.197**|0.685|0.653||**0.047**|0.755|0.562|
> ||
> |N-DR-JL [LR, Gaussian]|0.231|0.731|**0.651**||0.247|**0.698**|**0.664**||0.113|0.779|0.537|
> |N-DR-JL [LR, Epanechnikov]|0.235|0.741|**0.655**||0.251|**0.693**|**0.663**||0.108|**0.784**|0.552|
> |N-DR-JL [NB, Gaussian]|**0.204**|**0.748**|0.650||**0.198**|0.691|0.653||**0.049**|0.778|**0.574**|
> |N-DR-JL [NB, Epanechnikov]|**0.209**|**0.744**|0.648||**0.191**|0.681|0.637||**0.046**|**0.786**|**0.570**|
> ||
> |N-MRDR-JL [LR, Gaussian]|0.217|0.728|**0.662**||0.252|**0.697**|**0.666**||0.107|0.785|0.539|
> |N-MRDR-JL [LR, Epanechnikov]|0.233|0.734|**0.656**||0.253|**0.695**|**0.666**||0.097|0.791|0.560|
> |N-MRDR-JL [NB, Gaussian]|**0.208**|**0.742**|0.651||**0.206**|0.694|0.663||**0.045**|**0.793**|**0.583**|
> |N-MRDR-JL [NB, Epanechnikov]|**0.207**|**0.756**|0.635||**0.194**|0.690|0.644||**0.044**|**0.802**|**0.587**|
> ||
>
> Adopting either the Gaussian kernel or the Epanechnikov kernel in our methods is able to stably outperform the baseline methods in all metrics and these two kernels yield similar performance of our methods.
>
> > **[Q3] Can you please detail again for the semi-synthetic experiments how do you define the set of $\mathcal{N}_{u,i}$ as the set of historical user and item interactions for the neighbors of (u,i) who do have an influence on user u?**
>
> **Response:** We thank the reviewer for pointing out this issue. In the semi-synthetic experiment, **we suppose that all the $o_{u, i'}$ which $i' \neq i$ affects $r_{u, i}$.** The intuition behind is that users who have seen better movies than the current movie are more likely to have a low rating towards the current movie than the average rating. It is intuitive and is verified in the case study (Appendix I) in our revised manuscript. **Meanwhile, we suppose that all the $o_{u', i}$ which $u' \neq u$ affects $r_{u, i}$** because users may subjectively believe that movies that have been watched more times are of better quality. So we suppose $\mathcal{N}_{u, i}$ is {$({u'}, i')\neq (u, i) \mid  u'=u~ \text{or}~ i'=i$} in the semi-synthetic experiment.
>
> > **[Q4] How does $p_{u, i}=p \alpha^{\max \left(0, 4-r_{u, i}\right)}$ account for the neighboring effect too?**
>
> **Response:** We thank the reviewer for the question. In fact, $p_{u, i}=p \alpha^{\max \left(0, 4-r_{u, i}\right)}$ only taking MNAR effect into account. Neighborhood effect is considered by splitting $r_{u, i}$ into $r_{u, i}(1, g)$, introducing $\pi_{g}$ and setting the ideal loss to $\tilde{\mathcal{L}}_\mathrm{ideal}(\hat{\mathbf{R}})$.

---

> ### Author Response · Authors · 2023-11-23
> **Please kindly find our concise and clear rebuttal below for addressing your current concerns [Q5-Q7, W1]**
>
> > **[Q5] c is chosen to be the median of all g_u,i according to p. 7 but g_u,i is also defined depending on c.**
>
> **Response:** We thank the reviewer for pointing out this issue and we apologize for the **typo** here. Actually, $c$ is chosen to be the median of all
> $ \sum_{ (u',i') \in N_{(u,i)} } o_{u',i'} $. We have fixed this typo in our revised manuscript.
>
> > **[Q6] Does KuaiRec specify the MAR and MAR watching ratio records? If not, how to measure the neighboring effect?**
>
> **Response:** We thank the reviewer for the useful question. We manually split the MNAR and MAR set in KuaiRec dataset. Because KuaiRec is a fully exposed dataset, we uniformly sample 5% watching ratio records for each user to generate the MAR set. Meanwhile, we adopt the same technic as semi-synthetic experiments to manually set a propensity for each user-item pair to sample the MNAR set.
>
> > **[Q7] Can you please elaborate: “we adjust p to ensure the total observed sample is 5% of the entire matrix”?**
>
> **Response:** We thank the reviewer for the detailed question. As reviewer mentioned, the definition of $p_{u, i}$ is $p_{u, i} = \textcolor{red}{p} \alpha^{\max \left(0, 4-r_{u, i}\right)} \cdot m_{u, i}$, where $\textcolor{red}{p}$ is a pre-specified constant. The expected observation number of user-item pair is $\sum_{(u,i) \in \mathcal{D}} p_{u, i}$. Following the previous studies [1-2], we adjust $\textcolor{red}{p}$ to ensure $\sum_{(u,i) \in \mathcal{D}} p_{u, i} = 0.05 * |\mathcal{D}|$ for different mask number $ m_{u, i}$.
>
> > **[W1] Some typos and unclear Figure 1.**
>
> **Response:** We thank the reviewer for pointing out some typos and unclear figure in our original manuscript. In our revised manuscript, we fixed those typos and add a notation table in Figure 1 to improve the readability.
>
> ***
> **We hope the above discussion will fully address your concerns about our work.** We really appreciate your insightful and constructive comments to further help us improve the quality of our manuscript. Thank you!
> ***
>
> **Reference**
>
> [1] Tobias Schnabel et al. Recommendations as Treatments: Debiasing Learning and Evaluation. ICML 16.
>
> [2] Xiaojie Wang et al. Doubly Robust Joint Learning for Recommendation on Data Missing Not at Random. ICML 19.

---

### Official Review · Reviewer_TJ54 · 2023-10-24

**Soundness:** 2 fair
**Presentation:** 3 good
**Contribution:** 2 fair
**Rating:** 5
**Confidence:** 4

**Summary:**

This paper addresses the combined impact of selection bias and neighborhood effects in recommender systems.\
It introduces a novel approach to represent neighborhood effects as interference, alongside a treatment representation.\
The paper establishes a theoretical connection with existing methods, showing that their approach achieves unbiased learning in the presence of both selection bias and neighborhood effects.\
Experimental validation is conducted on semi-synthetic and real-world datasets to demonstrate the effectiveness of the proposed methods.

**Strengths:**

1. The paper is comprehensive and provides a theoretical analysis.
- the paper provides a robust theoretical foundation for its proposed methods. It derives unbiased estimators for the ideal loss, establishes a connection to prior methods that do not account for neighborhood effects, and includes analyses of tail bounds and generalization error bounds for the proposed estimators.

2. The experiment is thorough.
- the paper substantiates its claims with empirical results from experiments conducted on both semi-synthetic and real-world datasets. These experiments demonstrate that the proposed estimators outperform previous methods when neighborhood effects are present, underscoring the practical utility and effectiveness of the proposed approach.

**Weaknesses:**

1. Motivation is weak
- why do we need to eliminate the neighborhood effect?
- for example, existing recommenders can consider the neighborhood effect in the training phase and make recommendations with the neighborhood effect (e.g., similar users have similar embedding and thus get similar recommendations).
- Therefore, the neighborhood effect can be a rich information source for model training.

2. Assumptions are not realistic.
- why $r_{u,i}$ is affected by $o_{u,i}$? In my opinion, $o_{u,i}$ is just a treatment to observe $r_{u,i}$, and does not affect the 'value' of  $r_{u,i}$. (i.e., the value of $r$ is affected only by $x$ and observed only when $o=1$).
- If $r_{u,i}$ is affected by $o_{u,i}$, i think the assumption 3 is not hold.
- In the paper, g is a scalar (a continuous variable), not a representation vector.

3. Minor concerns
- In the real-world experiment, the authors use 5% MAR test ratings for the propensity estimation. This process is unrealistic.
- In the semi-synthetic experiment, the definition of neighborhood effect is the number of neighbor pairs with $o >= c$. what does it mean? Since $o \in {0,1}$, i cannot understand the c is chosen to be the median of all $g$.

**Questions:**

Please refer to the weaknesses.

---

> ### Author Response · Authors · 2023-11-23
> **Please kindly find our concise and clear rebuttal below for addressing your current concern [W1]**
>
> We sincerely appreciate the reviewer’s great efforts and insightful comments to improve our manuscript. In below, we address these concerns point by point and try our best to update the manuscript accordingly.
>
> **1. Motivation**
> > **Why do we need to eliminate the neighborhood effect?** For example, existing recommenders can consider the neighborhood effect in the training phase and make recommendations with the neighborhood effect (e.g., similar users have similar embedding and thus get similar recommendations). Therefore, the neighborhood effect can be a rich information source for model training.
>
> **Response:** The reviewer has raised an important point. We hope the following clarification addresses your concerns about the motivation.
>
> -	We agree with the reviewer that modern recommender systems consider the neighborhood effect in the training phase and make recommendations with the neighborhood effect.
> -	Nonetheless, we know that **users tend to behave similarly to the others in a group, even if doing so goes against their own judgment**, making the feedback in the observed data not reflect the true preferences of the users [1].
> -	We also notice another related work [2] referenced by **Reviewer qJ35,** in which the authors discuss the presence of interference in debiasing user feedback for the learning-to-rank (LTR) task.
>
> Moreover, **we also add a real-world example using the KuaiRec dataset demonstrating the necessity of eliminating the neighborhood effect for debiased recommendation** in Appendix I, for a clearer comprehension of the motivation in this paper.
>
> -	For the same item interacted with different users, we use the user social network information to compute and compare the feedback similarity of friends and non-friends. The results are shown in Figure 3(a). **Compared with the non-friend user pairs, it is clear that there is a higher similarity in the ratings of friends for the same item.**
>
> -	For a given user with different items, we use timestamps to first select the $K$ most recent items this user has interacted with, then compute the average video viewing time among all users on the fully exposed dataset. **As shown in Table 3 and Figure 3(b), we found that the lower the average viewing time of the items the user recently interacted with, the better the user's feedback on the current item will be, and vice versa.** Such a feedback mechanism is reasonable: when users have previously observed more low-quality videos, they will provide better feedback on the current items.

---

> ### Author Response · Authors · 2023-11-23
> **Please kindly find our concise and clear rebuttal below for addressing your current concerns [W2-W3]**
>
> **2. Assumptions**
> > **[W2.1] why $r_{u, i}$ is affected by $o_{u, i}$ ? In my opinion, $o_{u, i}$ is just a treatment to observe $r_{u, i}$ and does not affect the 'value' of $r_{u, i}$. (i.e., the value of $r$ is affected only by $x$ and observed only when $o=1$).**
>
> **Response:** In fact, $o_{u, i}$ is **not** just a treatment to observe $r_{u, i}$ and **does** affect the 'value' of $r_{u, i}$. **We further clarify the definition of $o_{u, i}$ in Section 2 and add a notation table in Figure 1 to make the presentation clearer.**
>
> -	Specifically, $x_{u,i}$, $o_{u,i}$, and $r_{u,i}$ are the feature, treatment (e.g., exposure), and feedback (e.g., conversion)} of the user-item pair $(u,i)$. **These broad concepts can have different meanings in different recommendation scenarios.**
> -	For instance, one may consider $o_{u,i}$ equals 1 or 0 represents whether the item $i$ is exposed to user $u$ or not, and $r_{u,i}$ is the conversion indicator. **Thus it is plausible that the exposure of the item affects the conversion.**
>
> > **[W2.2] If $r_{u, i}$ is affected by $o_{u, i}$, I think the assumption 3 is not hold.**
>
> **Response:** We would like to distinguish here between **the observed feedback $r_{u, i}$** and **the potential feedback $r_{u, i}(1)$**.
>
> -	We **agree** with the reviewer that $r_{u, i}$ is affected by $o_{u, i}$, e.g., the exposure of the item affects the conversion.
>
> -	Nonetheless, what assumption 3 states is that the potential feedback $r_{u, i}(1)$ is independent with the exposure indicator $o_{u, i}$.
>
> -	Since the definition of $r_{u, i}(1)$ is the potential feedback that would be observed if item $i$ had been exposed to user $u$  (i.e., $o_{u,i}$ had been set to 1). Thus it is plausible that $r_{u, i}(1)$ is independent with $o_{u, i}$.
>
> > **[W2.3] In the paper, $g$ is a scalar (a continuous variable), not a representation vector.**
>
> **Response:** We thank the reviewer for pointing out this issue.
>
> -	On a **theoretical** level, our approach allows $g$ to be a vector. Noting that **there is a significant difference in the statistical theory when $g$ is a vector and a scalar, we discuss in detail the new proofs and the theorems in main text for multi-dimensional $g$ in Appendix G.**
>
> -	On a **experimental** level, **as the reviewer captured, we only considered the cases when $g$ is a scalar (a continuous variable), but such implementation has already achieved significant performance improvement**. In the future, we will conduct more experiments to explore the use of multi-dimensional $g$ to implement the proposed methods.
>
> **3. Minor concerns**
> - In the real-world experiment, the authors use 5\% MAR test ratings for the propensity estimation. This process is unrealistic.
>
> **Response:** Indeed, in our original manuscript, **we implement the proposed methods using both Logistic Regression (LR) and Naive Bayes (NB) for estimating the propensities. LR does not require any MAR test ratings for the propensity estimation**, whereas NB requires 5\% MAR test ratings for the propensity estimation. **We carefully revised the statement about propensity estimation in Section 6, as well as highlighting the results of the experiments using LR and NB in Table 2.**
>
> > **[W3.2] In the semi-synthetic experiment, the definition of neighborhood effect is the number of neighbor pairs with $o \geq c$. what does it mean? Since $o \in $ {$0$, $1$}, I cannot understand the $c$ is chosen to be the median of all $\boldsymbol{g}$.**
>
> **Response:** We thank the reviewer for the careful reading and apologize for the **typo** here. Actually, $c$ is chosen to be the median of all $ \sum_{ (u',i') \in N_{(u,i)} } o_{u',i'} $. **We have fixed this typo in our revised manuscript.**
>
> ***
>
> Finally, we would like to kindly remind the reviewer that he/she **might have mistakenly flagged "Flag For Ethics Review"** due to checking both "No ethics review needed" and "Yes, Responsible research practice (e.g., human subjects, data release)".
>
> **We hope the above discussion will fully address your concerns about our work.** We really appreciate your insightful and constructive comments to further help us improve the quality of our manuscript. Thank you!
>
> ***
>
> **References**
>
> [1] Yu Zheng et al. Disentangling User Interest and Conformity for Recommendation with Causal Embedding, WWW 2021.
>
> [2] Mouxiang Chen et al. Adapting Interactional Observation Embedding for Counterfactual Learning to Rank. SIGIR 2021.

---

### Official Review · Reviewer_g9eU · 2023-10-30

**Soundness:** 3 good
**Presentation:** 3 good
**Contribution:** 3 good
**Rating:** 6
**Confidence:** 2

**Summary:**

Selection bias in recommender systems arises from the filtering process and user interactions, with most studies focusing on addressing it for unbiased prediction models. However, these studies often overlook the neighborhood effect, which is the variation in potential outcomes due to treatments assigned to other user-item pairs. This paper formulates the neighborhood effect as an interference problem and proposes a novel ideal loss to deal with selection bias in the presence of this effect. Two new estimators are developed, which are shown to achieve unbiased learning when both selection bias and neighborhood effects are present, unlike existing methods. Extensive experiments confirm the effectiveness of these proposed methods.

**Strengths:**

1. The studied topic is practical and interesting.
2. The experiments are very detailed for reproducing.

**Weaknesses:**

1. Too many assumptions made the manuscript hard to follow.

**Questions:**

n/a

---

> ### Author Response · Authors · 2023-11-23
> **Please kindly find our concise and clear rebuttal below for addressing your current concern [W1]**
>
> We sincerely appreciate the reviewer’s great efforts and insightful comments to improve our manuscript. In below, we try our best to address these concern and update the manuscript accordingly.
>
> > **[W1] Too many assumptions made the manuscript hard to follow.**
>
> **Response:** We thank for the reviewer for raising such concern. We would like to kindly remind the reviewer that all causal conclusions are based on a set of assumptions [1,2]. In this paper, we have adhered to common assumptions widely employed in causal inference.
>
> **It's important to highlight that the primary assumption underpinning our approach is Assumption 1 (neighborhood treatment representation). The remaining assumptions we utilize are standard in causal inference and kernel-smoothing estimation.** To elaborate:
> - Assumptions 2-3 are common assumptions (also called backdoor adjustment criterion in structural causal model, or unconfoundedness assumption in potential outcome framework) in causal inference to ensure the identifiability of the ideal loss (eq. (2)).
> - Assumption 4 aligns with the standard assumption in kernel-smoothing estimation [3, 4, 5].
> - Assumption 5 merely assumes the boundedness of propensities and imputed errors, which is a trivial assumption.
>
> ***
> **We hope the above discussion will fully address your concerns about our work.** We really appreciate your insightful and constructive comments to further help us improve the quality of our manuscript. Thank you!
> ***
>
> **Reference**
>
> [1] Miguel A. Hernán et al. Causal Inference: What If. 2020.
>
> [2] Guido W. Imbens et al. Causal Inference For Statistics Social and Biomedical Science. 2015.
>
> [3] Jianqing Fan et al. Local Polynomial Modelling and Its Applications. 1996.
>
> [4] Qi Li et al. Nonparametric econometrics. 2007.
>
> [5] Wolfgang Härdle et al. Nonparametric and Semiparametric Models. 2004.

---

### Official Review · Reviewer_qJ35 · 2023-10-31

**Soundness:** 3 good
**Presentation:** 3 good
**Contribution:** 3 good
**Rating:** 8
**Confidence:** 3

**Summary:**

This research investigates the influence of other user-item interactions on their ratings in Recommender Systems (RS). While prior studies concentrated on reducing selection bias, neglecting the neighborhood effect can lead to distorted estimates and subpar predictive model performance. This study introduces a treatment representation to capture the neighborhood effect and suggests a new loss function and estimators to tackle both selection bias and neighborhood effects, resulting in unbiased learning compared to current approaches. The effectiveness of these methods is demonstrated through theoretical assurances and comprehensive experiments.

**Strengths:**

- The introduction of the neighborhood effect in mitigating bias in Recommender Systems (RS) is innovative. The research addresses a significant issue, and its rationale is evident.

- The paper is well-structured and the method is substantiated by robust theoretical foundations.

**Weaknesses:**

- The breach of SUTVA and the presence of interference in debiasing user feedback were previously discussed in [1], where they also examined the interactions between propensity and implicit feedbacks on other items. I would like to see a discussion on it in this manuscript.

- A case study or a real-world example demonstrating the neighborhood effect would be valuable for a clearer comprehension of the underlying motivation.

- To enhance the clarity, it is advisable for the authors to furnish pseudocodes delineating the procedural steps of the proposed estimator and the propensity estimation process.

[1] Mouxiang Chen, Chenghao Liu, Jianling Sun, and Steven C.H. Hoi. 2021. Adapting Interactional Observation Embedding for Counterfactual Learning to Rank. In Proceedings of the 44th International ACM SIGIR Conference on Research and Development in Information Retrieval (SIGIR '21). Association for Computing Machinery, New York, NY, USA, 285–294. https://doi.org/10.1145/3404835.3462901

**Questions:**

Please see the Weaknesses section.

---

> ### Author Response · Authors · 2023-11-23
> **Please kindly find our concise and clear rebuttal below for addressing your current concerns [W1-W3]**
>
> We sincerely appreciate the reviewer’s great efforts and insightful comments to improve our manuscript. In below, we address these concerns point by point and try our best to update the manuscript accordingly.
>
> > **[W1] The breach of SUTVA and the presence of interference in debiasing user feedback were previously discussed in [1], where they also examined the interactions between propensity and implicit feedback on other items. I would like to see a discussion on it in this manuscript.**
>
> **Response:** We thank the reviewer for bringing up the relevant literature [1], and **we have added a detailed discussion in the Related Work Section regarding the relation and difference between our paper and [1]**, and **have added a clear citation to [1] as a key motivation of our paper** in the Introduction Section.
>
> -	Despite **both our paper and [1] exploring the influence induced by "other user-item interactions"**, it appears to be the sole similarity between the two studies.
> -	Firstly, for the **problem settings**, [1] focuses on the task of learning to rank (LTR), addressing position bias using implicit feedback data. In contrast, our paper focuses on eliminating selection bias in the rating prediction task using explicit feedback data.
> -	Secondly, for the **basic ideas**, [1] considers "other user-item interactions" as "confounders" (refer to the third paragraph in the Introduction Section). In contrast, our paper regards "other user-item interactions" as a new "treatment" from the perspective of interference in causal inference.
> -	Thirdly, for the **proposed methods**, [1] uses embedding as a proxy confounder to capture the influence of "other user-item interactions". In contrast, our paper formally formulates the influence of "other user-item interactions" as an interference problem in causal inference, and introduces a treatment representation to capture the influence. On this basis, we propose a novel ideal loss that can be used to deal with selection bias in the presence of interference. We also provide comprehensive **theoretical guarantees**.
>
> > **[W2] A case study or a real-world example demonstrating the neighborhood effect would be valuable for a clearer comprehension of the underlying motivation.**
>
> **Response:** Thank you for the kind advice. As suggested by the reviewer, we add a real-world example using the KuaiRec dataset demonstrating the presence of the neighborhood effect in Appendix I, for a clearer comprehension of the motivation in this paper.
>
> -	For the same item interacted with different users, we use the user social network information to compute and compare the feedback similarity of friends and non-friends. The results are shown in Figure 3(a). **Compared with the non-friend user pairs, it is clear that there is a higher similarity in the ratings of friends for the same item.**
>
> -	For a given user with different items, we use timestamps to first select the $K$ most recent items this user has interacted with, then compute the average video viewing time among all users on the fully exposed dataset. **As shown in Table 3 and Figure 3(b), we found that the lower the average viewing time of the items the user recently interacted with, the better the user's feedback on the current item will be, and vice versa.** Such a feedback mechanism is reasonable: when users have previously observed more low-quality videos, they will provide better feedback on the current items.
>
> > **[W3] To enhance the clarity, it is advisable for the authors to furnish pseudocodes delineating the procedural steps of the proposed estimator and the propensity estimation process.**
>
> **Response:** We thank the reviewer for pointing out this issue. As suggested by the reviewer, **we add pseudocodes delineating the propensity estimation process in Alg. 1, as well as the procedural steps of the proposed N-IPS, N-DR, and N-MRDR in Alg. 2-4, respectively.** Please kindly refer to Appendix H for the added 4 algorithmic details using pseudocodes.
>
> ***
> **We hope the above discussion will fully address your concerns about our work.** We really appreciate your insightful and constructive comments to further help us improve the quality of our manuscript. Thank you!
> ***
>
> **Reference**
>
> [1] Mouxiang Chen et al. Adapting Interactional Observation Embedding for Counterfactual Learning to Rank. SIGIR 2021.

---

> > ### Comment · Reviewer_qJ35 · 2023-11-23
> > **Thank you for your responses**
> >
> > Dear authors,
> >
> > Thanks for the responses. The responses address the most of my concerns. I decide to maintain my score.

---

> > > ### Author Response · Authors · 2023-11-23
> > > **Thank you for your timely responses and positive recommendation of our manuscript!**
> > >
> > > We are glad to know that your concerns have been effectively addressed. We are very grateful for your constructive comments and questions, which helped improve the clarity and quality of our paper. Thanks again!

---

### Author Response · Authors · 2023-11-23
**General responses and manuscript revision summary**

Dear reviewers and AC,

We sincerely thank all reviewers and AC for their great effort and constructive comments on our manuscript. We know that we are now approaching the end of the author-reviewer discussion and apologize for our late rebuttal. During the rebuttal period, we have been focusing on these beneficial suggestions from the reviewers and doing our best to add several experiments and revise our manuscript. We believe our current carefully revised manuscript can address all the reviewers’ concerns.

As reviewers highlighted, we believe our paper tackles an important and relevant problem **(Reviewer g9eU, Reviewer qJ35)**, introduces a novel interesting and perspective idea **(Reviewer g9eU, Reviewer qJ35)** and is well-written **(Reviewer 1Wkk)**. We also appreciate that the reviewers found the proposed methods intriguing and offers inspiration **(Reviewer g9eU)** with sound theoretical analysis **(Reviewer qJ35, Reviewer TJ54, Reviewer 1Wkk)**. In addition, **all the reviewers** mentioned that our paper has a solid and convincing experiments.

Moreover, we thank the reviewers for pointing out the concerns regarding the motivation and clarity problems **(Reviewer TJ54)**, as well as for the suggestions for investigating both types of kernel functions and explaining more about the detail in semi-synthetic experiments **(Reviewer 1Wkk)**, and for the suggestions for discussing the relation and different between our methods and previous methods and providing pseudocodes for easy reading **(Reviewer qJ35)**. In response to these comments, we have carefully revised and enhanced our manuscript with the following important changes with the added experiments:

-	[Reviewer TJ54, Reviewer qJ35] For clarifying our motivation, **we conduct a case study on the KuaiRec dataset demonstrating the necessity of eliminating the neighborhood effect for debiased recommendation** (in Appendix I).

-    [Reviewer TJ54] **We further clarify the definition of $o_{u, i}$ in Section 2 and add a notation table in Figure 1 to make the presentation and assumption 3 clearer.**

-   [Reviewer TJ54] We implement the proposed methods **using both Logistic Regression (LR) and Naive Bayes (NB) for estimating the propensities**, where **LR does not require any MAR test ratings for the propensity estimation**.

-	[Reviewer 1Wkk] We **add experiments using both Gaussian kernel and Epanechnikov kernel** in proposed methods in Table 2.

-	[Reviewer qJ35] We **add the discussion in terms of the relation and difference between our paper and [Chen et al., SIGIR 21]** in the Related Work Session, and **add a clear citation to [Chen et al., SIGIR 21] as a key motivation of our paper** in the Introduction Section.

-	[Reviewer qJ35] We **add pseudo-codes for propensity learning, N-IPS, N-DR-JL and N-MRDR-JL** (in Appendix H).

These updates are temporarily highlighted in "$\textcolor{red}{red}$" for facilitating checking.

We hope our response and revision could address all the reviewers' concerns, and are more than eager to have further discussions with the reviewers in response to these revisions.

Thanks,
Submission8755 Authors.

---

### Meta-Review · Area_Chair_ToHs · 2023-12-14

**Metareview:**

The paper addresses neighborhood effect for unbiased predictions in recommendations, in addition to the commonly studied selection bias. Two new estimators are developed, which are shown to achieve unbiased learning when both selection bias and neighborhood effects are present.

Strength: the method is substantiated by robust theoretical foundations, and the authors provided extensive experiments to validate the effectiveness of the proposed method.
Weakness: various reviewers raised questions on assumptions made, and the motivation on neighborhood effect is not fully justified.

**Justification For Why Not Higher Score:**

The topic might be less applicable to this specific community. The motivation in eliminating neighborhood effect is weak. The analyses on KuaiRec dataset is compounded by the fact that friends tend to share similar interests and preferences.

**Justification For Why Not Lower Score:**

The topic

---

### Decision · Program_Chairs · 2024-01-16

Accept (poster)